# Generalized-Smooth Bilevel Optimization with Nonconvex Lower-Level

**Siqi Zhang** [1]  **Xing Huang** [1]  **Feihu Huang** [1,2]

## Abstract

Bilevel optimization is widely applied in many machine learning tasks such as hyper-parameter learning and meta learning. Recently, many algorithms have been proposed to solve these bilevel optimization problems, which rely on the smoothness condition of objective functions of the bilevel optimization. In fact, some machine learning tasks such as learning language model do not satisfy the smoothness condition of objective functions. More recently, some methods have begun to study generalized smooth bilevel optimization. However, these proposed methods for generalized smooth bilevel optimization only focus on the (strongly) convex lower objective function. Meanwhile, these methods only consider the generalized-smooth upper-level objective, but still require the standard smooth lower-level objective in the bilevel optimization. To fill this gap, in the paper, thus we study the generalized-smooth bilevel optimization with the nonconvex lower-level objective function, where both upper-level and lower-level objectives are generalized-smooth. We propose an efficient single-loop Hessian/Jacobian-free penalty normalized gradient (i.e., PNGBiO) method. Moreover, we prove that our PNGBiO obtains a fast convergence rate of $O(\frac{1}{T^{1/4}})$ for finding a stationary solution, where $T$ denotes the iteration number. Meanwhile, we also propose a stochastic version of our PNGBiO (i.e., S-PNGBiO) method to solve stochastic bilevel problems, and prove that our S-PNGBiO has a fast convergence rate of $O(\frac{1}{T^{1/6}})$. Some experimental results on hyper-parameter learning and meta learning demonstrate efficiency of our proposed methods.

[1]College of Computer Science and Technology, Nanjing University of Aeronautics and Astronautics, Nanjing, China. [2]MIIT Key Laboratory of Pattern Analysis and Machine Intelligence, Nanjing, China. Correspondence to: Feihu Huang <huangfeihu2018@gmail.com>.

*Proceedings of the 42$^{nd}$ International Conference on Machine Learning*, Vancouver, Canada. PMLR 267, 2025. Copyright 2025 by the author(s).

## 1. Introduction

In this work, we focus on studying the following nonsmooth nonconvex bilevel optimization, defined as

$$\min_{x\in\mathbb{R}^d, y\in y^*(x)} f(x,y), \qquad \text{(UL)} \qquad (1)$$

$$\text{s.t.} \quad y^*(x) \triangleq \arg\min_{y\in\mathbb{R}^p} g(x,y), \qquad \text{(LL)}$$

where the upper-level (UL) function $f(x,y) : \mathbb{R}^d \times \mathbb{R}^p \to \mathbb{R}$ is *generalized-smooth*, and nonconvex with respect to (w.r.t) the variables $x$ and $y$, and the lower-level (LL) function $g(x,y) : \mathbb{R}^d \times \mathbb{R}^p \to \mathbb{R}$ is *generalized-smooth* and weakly convex w.r.t the LL variable $y$. Problem (1) frequently appears many machine learning tasks such as meta learning (Hao et al., 2024; Gong et al., 2024b) and deep neural network pruning (Gao et al., 2024). Recently, bilevel optimization has widely received attention in machine learning community, due to its ability of capturing the hierarchical structures in many machine learning tasks such as meta learning (Ji et al., 2021; Hao et al., 2024) and reinforcement learning (Hong et al., 2023). Meanwhile, many algorithms have been developed to solve the bilevel optimization problems.

When the lower-level objective function $g(x,y)$ on the variable $y$ is strongly convex and twice differential, fortunately, we can get a closed-form gradient of the upper-level objective function $F(x) = f(x, y^*(x))$, defined as

$$\nabla F(x) = \nabla_1 f(x, y^*(x)) \qquad (2)$$
$$- \nabla_{12}^2 g(x, y^*(x))\nabla_{22}^2 g(x, y^*(x))^{-1}\nabla_2 f(x, y^*(x)).$$

Recently, based on the above closed-form gradient (2), some effective approximated gradient algorithms (Ghadimi & Wang, 2018; Ji et al., 2021; Hong et al., 2023) have been proposed to solve bilevel optimization by using the following approximated gradient:

$$\hat{\nabla} f(x, \hat{y}) = \nabla_1 f(x, \hat{y}) \qquad (3)$$
$$- \nabla_{12}^2 g(x, \hat{y})\nabla_{22}^2 g(x, \hat{y})^{-1}\nabla_2 f(x, \hat{y}),$$

where $\hat{y}$ is an approximate of the solution $y^*(x) = \arg\min_y g(x,y)$. For example, (Ghadimi & Wang, 2018) proposed a class of approximated gradient methods based on approximate implicit differentiation. (Ji et al., 2021) proposed an effective approximated gradient methods based on

*Table 1.* Comparison of our method and the existing methods for **generalized-smooth bilevel optimization**. Here **S** denotes the standard smoothness condition. **GS** denotes the generalized-smoothness condition. **SC** denotes the strongly convex condition. **NC** denotes the non-convex condition. **H.J.F.** stands for Hessian/Jacobian-Free. Here $f(\cdot,\cdot)$ and $g(\cdot,\cdot)$ denote the upper-level and lower-level objective functions, respectively. $g(x,\cdot)$ denotes function on the second variable $y$ with fixing variable $x$.

| Algorithm | Reference | $g(\cdot,\cdot)$ | $f(\cdot,\cdot)$ | $g(x,\cdot)$ | Single-Loop | H.J.F. |
|---|---|---|---|---|---|---|
| BO-REP | (Hao et al., 2024) | **S** | **GS** | **SC** | | |
| SLIP | (Gong et al., 2024b) | **S** | **GS** | **SC** | $\checkmark$ | |
| ACCBO | (Gong et al., 2024a) | **S** | **GS** | **SC** | | |
| PNGBiO/S-PNGBiO | Ours | **GS** | **GS** | **NC** | $\checkmark$ | $\checkmark$ |

the iterative differentiation. More recently, (Huang, 2023; 2024) studied the nonconvex bilevel optimization where the lower-level objective function $g(x,y)$ on the variable $y$ is *local* strongly convex based on the following projection-aided approximated function, defined as

$$\tilde{\nabla} f(x,\hat{y}) = \nabla_1 f(x,\hat{y}) \qquad (4)$$
$$- \nabla_{12}^2 g(x,\hat{y})\big(\mathcal{S}_{[\mu,L_g]}\big[\nabla_{22}^2 g(x,\hat{y})\big]\big)^{-1}\nabla_2 f(x,\hat{y}),$$

where $\mathcal{S}_{[\mu,L_g]}[\cdot]$ denotes a projection on the set $\{X \in \mathbb{R}^{d\times d} : \mu \leq \varrho(X) \leq L_g\}$, and $\varrho(\cdot)$ denotes the eigenvalue function. In particular, (Huang, 2024) proposed an optimal Hessian/Jacobian-free method by using finite-difference estimator to approximate the projection-aided function (4).

When the lower-level objective function $g(x,y)$ on the variable $y$ is *not* (local) strongly convex, unfortunately, we can not get a closed-form gradient defined in (2). Thus, most of the existing methods (Liu et al., 2021; 2022; Kwon et al., 2023a; Liu et al., 2024) solve the following single-level constrained optimization problem instead of directly solving the bilevel optimization problem, defined as

$$\min_{x\in\mathbb{R}^d,y\in\mathbb{R}^p} f(x,y) \quad \text{s.t. } g(x,y) - \min_{y\in\mathbb{R}^p} g(x,y) \leq 0. \quad (5)$$

For example, (Liu et al., 2021) proposed a value-function-based interior-point method for nonconvex bilevel optimization. Meanwhile, (Liu et al., 2022) designed an effective dynamic barrier gradient descent method for bilevel optimization. More recently, (Liu et al., 2024) solve a variant of the single-level constrained optimization problem (5), where uses its Moreau envelope instead of the problem $\min_{y\in\mathbb{R}^p} g(x,y)$ in the problem (5), defined as

$$\min_{x\in\mathbb{R}^d,y\in\mathbb{R}^p} f(x,y) \quad \text{s.t. } g(x,y) - v_\gamma(x,y) \leq 0, \quad (6)$$

$$v_\gamma(x,y) = \min_{\theta\in\mathbb{R}^p} g(x,\theta) + \frac{1}{2\gamma}\|\theta - y\|^2.$$

So far, the above methods for bilevel optimization rely on the smoothness of its objective functions. In fact, some machine learning tasks such as learning language model (Zhang et al., 2019) and distributionally robust optimization (Chen et al., 2023) do not satisfy the smoothness condition of objective

functions. More recently, thus, some methods (Hao et al., 2024; Gong et al., 2024b) have begun to study generalized smooth bilevel optimization. However, these proposed methods for generalized smooth bilevel optimization only focus on the (strongly) convex lower objective function. Meanwhile, these methods only consider the generalized-smooth upper-level objective, but still require the standard smooth lower-level objective in the bilevel optimization.

In this paper, to fill this gap, we study the generalized-smooth bilevel optimization problem (1), where both upper-level and lower-level objectives are generalized-smooth, and its lower-level is nonconvex. We propose an efficient Hessian/Jacobian-free penalty normalized gradient (i.e., PNGBiO) method to solve the deterministic problem (1). Meanwhile, we also propose a stochastic PNGBiO (i.e., S-PNGBiO) method to solve the stochastic version of problem (1), defined as

$$\min_{x\in\mathbb{R}^d,y\in y^*(x)} \mathbb{E}_{\xi\sim\mathcal{D}}[F(x,y;\xi)], \qquad \text{(UL)} \quad (7)$$

$$\text{s.t.} \quad y^*(x) \triangleq \arg\min_{y\in\mathbb{R}^p} \mathbb{E}_{\zeta\sim\mathcal{O}}[G(x,y;\zeta)], \quad \text{(LL)}$$

where $f(x,y) \equiv \mathbb{E}_{\xi\sim\mathcal{D}}[F(x,y;\xi)]$ and $g(x,y) \equiv \mathbb{E}_{\zeta\sim\mathcal{O}}[G(x,y;\zeta)]$. Here $\xi$ and $\zeta$ are random variables.

In summary, our contributions are as follows:

(1) We propose an efficient single-loop PNGBiO method to solve the generalized-smooth nonconvex bilevel problem (1). Meanwhile, we also present a stochastic version of PNGBiO (i.e., S-PNGBiO) method to solve problem (7). In particular, our methods only use the first-order gradients instead of high computational Hessian/Jacobian matrices, so it has a lower computation at each iteration.

(2) We present a solid convergence analysis for our methods. Under some mild conditions, we prove that our PNGBiO method obtains a fast convergence rate of $O(\frac{1}{T^{1/4}})$ for finding a stationary solution of the problem (1). Meanwhile, we also prove that our S-PNGBiO method has a fast convergence rate of $O(\frac{1}{T^{1/6}})$ for finding a stationary solution of the problem (7).

(3) We provide some numerical experiments on data hyper-cleaning, hyper-parameter learning and meta learning to demonstrate efficiency of our methods.

## 2. Related Work

In this section, we review the generalized smoothness condition in optimization algorithms and the algorithms of bilevel optimization with nonconvex lower-level.

### 2.1. Generalized smoothness condition

The generalized smoothness condition firstly was studied in (Zhang et al., 2019), and was applied to the gradient clipping and normalized gradient methods. Subsequently, (Zhang et al., 2020) studied the momentum-based gradient clipping method under the generalized smoothness condition. Meanwhile, (Qian et al., 2021; Zhao et al., 2021) studied the incremental gradient clipping and stochastic normalized gradient methods under the generalized smoothness condition, respectively. Recently, (Chen et al., 2023) proposed a new symmetric generalized smoothness, which extends the standard $(L_0, L_1)$-generalized smoothness, and studied the variance reduction under this symmetric generalized smoothness condition. Subsequently, (Li et al., 2023) presented a more generalized smoothness condition and studied various gradient-based methods under this condition. More recently, (Hao et al., 2024; Gong et al., 2024b) studied the bilevel optimization under the generalized-smoothness condition. Meanwhile, (Xian et al., 2024) also studied the nonconvex minimax optimization under the generalized-smoothness condition.

### 2.2. Bilevel optimization with nonconvex Lower-Level

In recent years, numerous methods have been proposed to solve the bilevel optimization with nonconvex lower-level. For example, (Liu et al., 2022) proposed a first-order method for nonconvex-PL bilevel optimization with nonconvex LL satisfying the PL condition. (Shen & Chen, 2023) designed a penalty-based gradient method for constrained nonconvex-PL cases. Subsequently, (Huang, 2023) proposed momentum-based gradient methods for nonconvex-PL bilevel optimization. Meanwhile, (Kwon et al., 2023a) studied nonconvex bilevel optimization with LL meeting the proximal error-bound (EB) condition similar to PL. (Kwon et al., 2023b) proposed the F$^2$BA method for nonconvex-PL bilevel optimization, achieving the $O(\epsilon^{-1})$ gradient complexity when finding $\epsilon$-stationary solutions. However, this method requires stricter conditions such as the Lipschitz Hessian of the upper-level function $f(x, y)$. Although achieving certain results, they both rely on computationally expensive projected Hessian/Jacobian matrices. To address this issue, (Huang, 2024) proposed the HJFBiO method, using a finite-difference estimator and a new pro-jection operator to replace high-cost matrices with low-cost first-order gradients. On the other hand, (Liu et al., 2024) proposed the MEHA method for general bilevel optimization with nonconvex and possibly non-smooth LL objective functions, avoiding Hessian-related approximations and enabling single-loop implementation.

## Notation

For notational simplicity, let $\nabla_1 f(x, y)$ and $\nabla_2 f(x, y)$ denote the partial differentiation of function $f(x, y)$ on the first variable $x$ and the second variable $y$, respectively. $\|\cdot\|$ denotes the $\ell_2$ norm for vectors and spectral norm for matrices. $\langle x, y \rangle$ denotes the inner product of two vectors $x$ and $y$. $\mathcal{F}_t := \sigma(x^0, y^0, \theta^0, \mathcal{S}_f^0, \mathcal{S}_g^0, \hat{\mathcal{S}}_g^0, \cdots, \mathcal{S}_f^t, \mathcal{S}_g^t, \hat{\mathcal{S}}_g^t)$ is a $\sigma$-algebras. $\mathbb{E}[\cdot|\mathcal{F}_t]$ is the conditional probability given the $t$-th iteration. And $\mathbb{E}[\cdot]$ is the expectation operator w.r.t. stochastic variables. Let $\{x^t\}$ denote a sequence.

## 3. Preliminaries

In the problem (1), since lower-level objective $g(x, y)$ is nonconvex on variable $y$, we can not easily get the minimum of lower-level problem $\min_{y \in \mathbb{R}^p} g(x, y)$. Thus, we reformulate the lower-level problem by using a value function defined as:

$$v(x) \triangleq \min_{y \in \mathbb{R}^p} g(x, y). \tag{8}$$

Then, the problem (1) is equivalent to the following nonlinear programming problem

$$\min_{(x,y) \in \mathbb{R}^d \times \mathbb{R}^p} f(x, y), \quad \text{s.t. } g(x, y) - v(x) \leq 0, \tag{9}$$

which was initially introduced by (Outrata, 1990). Unfortunately, the local solutions of the reformulated problem (9) do not necessarily correspond to the stationary points of the problem (1) (Alcantara & Takeda, 2024). Therefore, we consider a Moreau envelope problem of problem (9) as in (Liu et al., 2024), which also can be seen as an approximate problem:

$$\min_{(x,y) \in \mathbb{R}^d \times \mathbb{R}^p} f(x, y), \quad \text{s.t. } g(x, y) - v_\gamma(x, y) \leq 0, \tag{10}$$

$$v_\gamma(x, y) = \min_{\theta \in \mathbb{R}^p} g(x, \theta) + \frac{1}{2\gamma} \|\theta - y\|^2,$$

where $\gamma > 0$ is regularization tuning parameter. Here we solve the following Lagrange function ( i.e., penalized function) instead of directly solving the problem (10), defined as:

$$\min_{x \in \mathbb{R}^d, y \in \mathbb{R}^p} f(x, y) + c_t \big(g(x, y) - v_\gamma(x, y)\big), \tag{11}$$

where $c_t > 0$ is a penalty parameter.

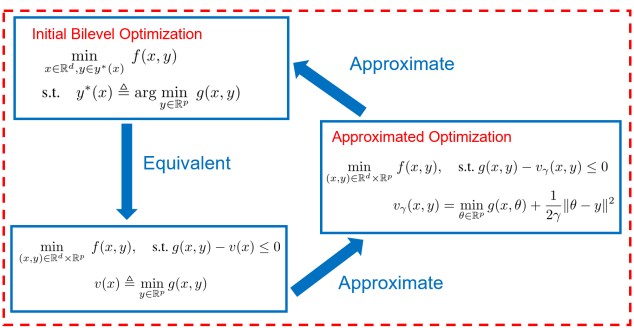

*Figure 1.* Relationship of the bilevel optimization problem (1) and the approximated optimzation problem (10).

For the problem (11), we consider the following residual function, defined as

$$R_t(x, y) := \text{dist}\big(0, \nabla f(x, y) + c_t(\nabla g(x, y) - \nabla v_\gamma(x, y))\big).$$

This residual function is a stationary measure for the problem (11). When $(x, y)$ is the stationary point to the problem (11), it follows that $\nabla f(x, y) + c_t\big(\nabla g(x, y) - \nabla v_\gamma(x, y)\big) = 0$. Based on Theorem A.4 of (Liu et al., 2024), it is known that any limit point $(\bar{x}, \bar{y})$ of sequence $(x^t, y^t)$ is a solution to the problem (10).

## 4. Penalty Normalized Gradient Methods

In this section, we propose an effective single-loop penalty normalized gradient (PNGBiO) method to solve the deterministic bilevel optimization problem (1). Meanwhile, we present a stochastic PNGBiO (i.e., S-PNGBiO) method to solve the stochastic bilevel optimization problem (7).

The PNGBiO algorithm is provided in Algorithm 1. From the above section, our PNGBiO method solve the bilevel optimization problem (1) by directly solving the approximated problem (10). Figure 1 shows that the relationship of these problems. Here we use the penalty method to solve the approximated problem (10) by directly solving the problem (11).

In Algorithm 1, given the current $\{x^t, y^t, \theta^t\}$, we begin with updating the variable $\theta$, defined as

$$\theta^{t+1} = \theta^t - \eta_t\big(\nabla_2 g(x^t, \theta^t) + \frac{\theta^t - y^t}{\gamma}\big), \quad (12)$$

where $\eta_t > 0$. Then we give two gradients:

$$d_x^t = \frac{1}{c_t}\nabla_1 f(x^t, y^t) + \nabla_1 g(x^t, y^t) - \nabla_1 g(x^t, \theta^{t+1}), \quad (13)$$

$$d_y^t = \frac{1}{c_t}\nabla_2 f(x^{t+1}, y^t) + \nabla_2 g(x^{t+1}, y^t) + \frac{\theta^{t+1} - y^t}{\gamma}, \quad (14)$$

---

**Algorithm 1** PNGBiO Algorithm

**Input**: Iteration number $T$, initialization $x^0, y^0, \theta^0$, learning rates $\eta_t, \alpha_t, \beta_t$, proximal parameter $\gamma > 0$, penalty parameter $c_t > 0$;

**Output**: $x^T, y^T$

1: **for** $t = 0, 1, \cdots, T - 1$ **do**
2:    $\theta^{t+1} = \theta^t - \eta_t\big(\nabla_2 g(x^t, \theta^t) + \frac{1}{\gamma}(\theta^t - y^t)\big)$;
3:    Compute $d_x^t = \frac{1}{c_t}\nabla_1 f(x^t, y^t) + \nabla_1 g(x^t, y^t) - \nabla_1 g(x^t, \theta^{t+1})$;
4:    Update $x^{t+1} = x^t - \alpha_t \frac{d_x^t}{\|d_x^t\|}$;
5:    Compute $d_y^t = \frac{1}{c_t}\nabla_2 f(x^{t+1}, y^t) + \nabla_2 g(x^{t+1}, y^t) + \frac{1}{\gamma}(\theta^{t+1} - y^t)$;
6:    Update $y^{t+1} = y^t - \beta_t \frac{d_y^t}{\|d_y^t\|}$.
7: **end for**

---

to update the variables $x$ and $y$, respectively. At the lines 4 and 6 of Algorithm 1, we use the normalized gradient descent to update the variables $x$ and $y$. Compared with the MEHA algorithm of (Liu et al., 2024), our PNGBiO algorithm uses the normalized gradient descent iteration, which unify the feature scales to make the gradient update more stable and efficient, and facilitating the rapid convergence of algorithm.

Algorithm 2 provides an algorithmic framework for our S-PNGBiO method, which extends Algorithm 1 to a stochastic setting. At the line 2 of Algorithm 2, we randomly draw three independent minibatch samples $\mathcal{S}_f^t = \{\xi_i^t\}_{i=1}^{B_t} \sim \mathcal{D}$, $\mathcal{S}_g^t = \{\zeta_i^t\}_{i=1}^{B_t} \sim \mathcal{O}$, and $\hat{\mathcal{S}}_g^t = \{\hat{\zeta}_i^t\}_{i=1}^{B_t} \sim \mathcal{O}$. Then at its line 3, we update the variable $\theta$ as follows,

$$\theta^{t+1} = \theta^t - \eta_t\left(\frac{1}{B_t}\sum_{i=1}^{B_t}\nabla_2 G(x^t, \theta^t; \hat{\zeta}_i^t) + \frac{1}{\gamma}(\theta^t - y^t)\right). \quad (15)$$

The gradients for updating the variables $x$ and $y$ are given,

$$\tilde{d}_x^t = \frac{1}{B_t}\left(\frac{1}{c_t}\sum_{i=1}^{B_t}\nabla_1 F(x^t, y^t; \xi_i^t) \right. \quad (16)$$
$$\left. + \sum_{i=1}^{B_t}(\nabla_1 G(x^t, y^t; \zeta_i^t) - \nabla_1 G(x^t, \theta^{t+1}; \zeta_i^t))\right),$$

$$\tilde{d}_y^t = \frac{1}{B_t}\left(\frac{1}{c_t}\sum_{i=1}^{B_t}\nabla_2 F(x^{t+1}, y^t; \xi_i^t) \right. \quad (17)$$
$$\left. + \sum_{i=1}^{B_t}\nabla_2 G(x^{t+1}, y^t; \zeta_i^t)\right) + \frac{\theta^{t+1} - y^t}{\gamma}.$$

**Algorithm 2** S-PNGBiO Algorithm

**Input**: Iteration number $T$, initialization $x^0, y^0, \theta^0$, learning rates $\eta_t, \alpha_t, \beta_t$, proximal parameter $\gamma > 0$, penalty parameter $c_t > 0$ and mini-batch size $B_t \geq 1$;

**Output**: $x^T, y^T$

1: **for** $t = 0, 1, \cdots, T-1$ **do**
2:     Randomly drawn three independent minibatch samples $\mathcal{S}_f^t = \{\xi_i^t\}_{i=1}^{B_t} \sim \mathcal{D}$, $\mathcal{S}_g^t = \{\zeta_i^t\}_{i=1}^{B_t} \sim \mathcal{O}$, $\hat{\mathcal{S}}_g^t = \{\hat{\zeta}_i^t\}_{i=1}^{B_t} \sim \mathcal{O}$;
3:     Update $\theta^{t+1} = \theta^t - \eta_t \big( \frac{1}{B_t} \sum_{i=1}^{B_t} \nabla_2 G(x^t, \theta^t; \hat{\zeta}_i^t) + \frac{1}{\gamma}(\theta^t - y^t) \big)$;
4:     Compute $\tilde{d}_x^t = \frac{1}{B_t} \sum_{i=1}^{B_t} \big( \frac{1}{c_t} \nabla_1 F(x^t, y^t; \xi_i^t) + \nabla_1 G(x^t, y^t; \zeta_i^t) - \nabla_1 G(x^t, \theta^{t+1}; \zeta_i^t) \big)$;
5:     Update $x^{t+1} = x^t - \alpha_t \frac{\tilde{d}_x^t}{\|\tilde{d}_x^t\|}$;
6:     Compute $\tilde{d}_y^t = \frac{1}{B_t} \sum_{i=1}^{B_t} \big( \frac{1}{c_t} \nabla_2 F(x^{t+1}, y^t; \xi_i^t) + \nabla_2 G(x^{t+1}, y^t; \zeta_i^t) \big) + \frac{\theta^{t+1} - y^t}{\gamma}$;
7:     Update $y^{t+1} = y^t - \beta_t \frac{\tilde{d}_y^t}{\|\tilde{d}_y^t\|}$.
8: **end for**

## 5. Convergence Analysis

In this section, we study the convergence properties of our PNGBiO and S-PNGBiO method under some mild assumptions. All related proofs are provided in the following Appendix A. We first give some standard assumptions on the lower-level objective $f(x, y)$ and the upper-level objective $g(x, y)$.

For notational simplicity, let $u = (x, y)$, $u' = (x', y')$, $\mu \in (0, 1)$ and $u_\mu = \mu u' + (1 - \mu)u$. Further let $\mu \theta + (1 - \mu)\theta_\gamma^*(x, y)$ and $M = \max_{\mu \in [0,1]} \|\nabla_1 g(x, \theta_\mu)\|$, where

$$\theta_\gamma^*(x, y) = \arg\min_{\theta \in \mathbb{R}^p} g(x, \theta) + \frac{1}{2\gamma} \|\theta - y\|^2.$$

**Assumption 5.1.** The upper-level objective function $f(u)$ is bounded below, i.e., $f^* = \inf_{u \in \mathbb{R}^d \times \mathbb{R}^p} f(u) > -\infty$.

**Assumption 5.2.** (**Generalized Smoothness of UL Function**) There exists $L_{fx,0}$, $L_{fx,1}$, $L_{fy,0}$ and $L_{fy,1}$ such that $\|\nabla_1 f(u) - \nabla_1 f(u')\| \leq (L_{fx,0} + L_{fx,1} \max_{\mu \in [0,1]} \|\nabla_1 f(u_\mu)\|^\rho) \|u - u'\|$ and $\|\nabla_2 f(u) - \nabla_2 f(u')\| \leq (L_{fy,0} + L_{fy,1} \max_{\mu \in [0,1]} \|\nabla_2 f(u_\mu)\|^\rho) \|u - u'\|$, where $\rho \in [0, 1]$ is a constant.

**Assumption 5.3.** (**Generalized Smoothness of LL Function**) There exists $L_{gx,0}$, $L_{gx,1}$, $L_{gy,0}$ and $L_{gy,1} > 0$ such that $\|\nabla_1 g(u) - \nabla_1 g(u')\| \leq (L_{gx,0} + L_{gx,1} \max_{\mu \in [0,1]} \|\nabla_1 g(u_\mu)\|^\rho) \|u - u'\|$ and $\|\nabla_2 g(u) - \nabla_2 g(u')\| \leq (L_{gy,0} + L_{gy,1} \max_{\mu \in [0,1]} \|\nabla_2 g(u_\mu)\|^\rho) \|u - u'\|$, where $\rho \in [0, 1]$ is a constant.

**Assumption 5.4.** The lower-level objective function $g(\cdot, \cdot)$ is $(\kappa_{g_1}, \kappa_{g_2})$-weakly convex on $\mathbb{R}^d \times \mathbb{R}^p$, i.e., $g(x, y) + \frac{\kappa_{g_1}}{2}\|x\|^2 + \frac{\kappa_{g_2}}{2}\|y\|^2$ is convex on $\mathbb{R}^d \times \mathbb{R}^p$.

Assumption 5.1 gives a lower bound of the upper-level objective function, which ensures the feasibility of the problem (1). Assumption 5.2 shows the generalized smoothness condition of the upper-level objective as in (Chen et al., 2023), which is milder than the generalized smoothness condition used in (Hao et al., 2024; Gong et al., 2024b). Moreover, we also give the generalized smoothness condition of the lower-level objective in Assumption 5.3, which instead of the standard smoothness condition of lower-level objective used in (Hao et al., 2024; Gong et al., 2024b). Assumption 5.4 shows that the lower-level objective function is weakly convex as in (Liu et al., 2024). In fact, this nonconvex lower-level objective condition used in our paper is also milder than the strongly-convex lower-level objective condition used in (Liu et al., 2024).

### 5.1. Deterministic Setting

**Theorem 5.5.** *Under Assumptions 5.1, 5.2, 5.3 and 5.4, given* $\gamma \in (0, \frac{1}{2\kappa_{g_2}})$, $c_t = \underline{c}(t+1)^{1/4}$ *with* $\underline{c} > 0$, *and when* $\rho \in [0, 1)$ *let* $0 < \alpha_t \leq \frac{\|\frac{1}{c_t}\nabla_1 f(x^t, y^t)\| + \|\nabla_1 g(x^t, y^t)\|}{4(K_0 + \kappa + K_1 + 2K_2) + 8(1 + \frac{1}{\eta_t \kappa_{g_2}})L_\theta^2 C + 6\|\nabla_1 g(x^t, \theta^{t+1})\|}$, $0 < \beta_t \leq \frac{\|\frac{1}{c_t}\nabla_2 f(x^t, y^t)\| + \|\nabla_2 g(x^t, y^t)\|}{4(K_3 + \kappa + K_4 + 2K_5) + 8(1 + \frac{1}{\eta_t \kappa_{g_2}})L_\theta^2 C}$, *when* $\rho = 1$ *let* $0 < \alpha_t \leq \frac{\|\frac{1}{c_t}\nabla_1 f(x^t, y^t)\| + \|\nabla_1 g(x^t, y^t)\|}{4(L_0 + \kappa + L_1) + 8(1 + \frac{1}{\eta_t \kappa_{g_2}})L_\theta^2 C + 6\|\nabla_1 g(x^t, \theta^{t+1})\|}$, $0 < \beta_t \leq \frac{\|\frac{1}{c_t}\nabla_2 f(x^t, y^t)\| + \|\nabla_2 g(x^t, y^t)\|}{4(L_2 + \kappa + L_3) + 8(1 + \frac{1}{\eta_t \kappa_{g_2}})L_\theta^2 C}$, *and*

$$\eta_t \in \left( \frac{1 + \sqrt{1 + 4C\left((L_{gx,0} + L_{gx,1}M)\frac{\alpha_t}{\|\tilde{d}_x^t\|} + \frac{\beta_t}{\gamma^2\|\tilde{d}_y^t\|}\right)}}{2\kappa_{g_2}C}, \right.$$

$$\left. \frac{1/\gamma - \kappa_{g_2}}{(1/\gamma + L_{gy,0} + L_{gy,1}\max_{\mu \in (0,1)} \|\nabla_2 g(x, \theta_\gamma^*(x,y)_\mu)\|^\rho)^2} \right), \quad where$$

$C = L_{gx,0} + L_{gx,1}M^\rho + \frac{2}{\gamma^2}$. *The the sequence of* $\{x^t, y^t, \theta^t\}_{t=0}^T$ *generated by Algorithm 1 satisfies*

$$\min_{0 \leq t \leq T} \|\theta^t - \theta_\gamma^*(x^t, y^t)\| = O\left(\frac{1}{T^{1/2}}\right),$$

$$\min_{0 \leq t \leq T} R_t(x^{t+1}, y^{t+1}) = O\left(\frac{1}{T^{\frac{1}{4}}}\right),$$

$$g(x^T, y^T) - v_\gamma(x^T, y^T) = O\left(\frac{1}{T^{\frac{1}{4}}}\right).$$

*Remark* 5.6. From the above Theorem 5.5, the two terms $\min_{0 \leq t \leq T} \|\theta^t - \theta_\gamma^*(x^t, y^t)\| = O\left(\frac{1}{T^{1/2}}\right)$ and $\min_{0 \leq t \leq T} R_t(x^{t+1}, y^{t+1}) = O\left(\frac{1}{T^{\frac{1}{4}}}\right)$ shows that our PNG-BiO algorithm has a convergence rate $O\left(\frac{1}{T^{\frac{1}{4}}}\right)$ to obtain the stationary solution of the problem (10). Adding the term $g(x^T, y^T) - v_\gamma(x^T, y^T) = O\left(\frac{1}{T^{\frac{1}{4}}}\right)$ ensures that the sequence $\{g(x^t, y^t)\}$ approaches the sequence $\{v_\gamma(x^t, y^t)\}$,

so the above three terms show that our PNGBiO algorithm has a convergence rate $O\left(\frac{1}{T^{\frac{1}{4}}}\right)$ to obtain the stationary solution of the problem (1). Thus, our PNGBiO method can obtain a gradient complexity of $O(\epsilon^{-4})$ for finding an $\epsilon$-stationary solution of the problem (1).

## 5.2. Stochastic Setting

**Assumption 5.7.** The stochastic gradient is unbiased, i.e., $\mathbb{E}_{\xi \sim \mathcal{D}}[\nabla F(x, y; \xi)] = \nabla f(x, y)$, $\mathbb{E}_{\zeta \sim \mathcal{O}}[\nabla G(x, y; \zeta)] = \nabla g(x, y)$. The variances of stochastic gradient estimators are bounded:

$$\mathbb{E}_{\xi \sim \mathcal{D}}[\|\nabla F(x, y; \xi) - \nabla f(x, y)\|^2] \leq \delta_f^2,$$
$$\mathbb{E}_{\zeta \sim \mathcal{O}}[\|\nabla G(x, y; \zeta) - \nabla g(x, y)\|^2] \leq \delta_g^2.$$

Assumption 5.7 is the classical assumption for stochastic algorithms (Hong et al., 2023; Kwon et al., 2023b). By Assumption 5.7, we have the variance $\|d_x^t - \tilde{d}_x^t\|^2 \leq \frac{\frac{1}{c_t}\delta_f^2 + 2\sigma_g^2}{B_t}$ and $\|d_y^t - \tilde{d}_y^t\|^2 \leq \frac{\frac{1}{c_t}\delta_f^2 + \sigma_g^2}{B_t}$.

**Theorem 5.8.** *Under Assumptions 5.1, 5.2, 5.3, 5.4 and 5.7, given $\gamma \in (0, \frac{1}{2\kappa_{g_2}})$, $c_t = \underline{c}(t+1)^{1/4}$ with $\underline{c} > 0$ and $B_t = O(T^{1/2})$, and when $\rho \in [0, 1)$ let $0 < \alpha_t \leq \frac{\|\frac{1}{c_t}\nabla_1 f(x^t, y^t)\| + \|\nabla_1 g(x^t, y^t)\|}{4(K_0 + \kappa + K_1 + 2K_2 + 2) + 8(1 + \frac{1}{\eta_t \kappa_{g_2}})L_\theta^2 C + 6\|\nabla_1 g(x^t, \theta^{t+1})\|}$, $0 < \beta_t \leq \frac{\|\frac{1}{c_t}\nabla_2 f(x^t, y^t)\| + \|\nabla_2 g(x^t, y^t)\|}{4(K_3 + \kappa + K_4 + 2K_5 + 2) + 8(1 + \frac{1}{\eta_t \kappa_{g_2}})L_\theta^2 C}$, when $\rho = 1$ let $0 < \alpha_t \leq \frac{\|\frac{1}{c_t}\nabla_1 f(x^t, y^t)\| + \|\nabla_1 g(x^t, y^t)\|}{4(L_0 + \kappa + L_1 + 2) + 8(1 + \frac{1}{\eta_t \kappa_{g_2}})L_\theta^2 C + 6\|\nabla_1 g(x^t, \theta^{t+1})\|}$, $0 < \beta_t \leq \frac{\|\frac{1}{c_t}\nabla_2 f(x^t, y^t)\| + \|\nabla_2 g(x^t, y^t)\|}{4(L_2 + \kappa + L_3 + 2) + 8(1 + \frac{1}{\eta_t \kappa_{g_2}})L_\theta^2 C}$, and*

$$\eta_t \in \left(\frac{1 + \sqrt{1 + 4C\left((L_{gx,0} + L_{gx,1}M)\frac{\alpha_t}{\|\tilde{d}_x^t\|} + \frac{\beta_t}{\gamma^2 \|\tilde{d}_y^t\|}\right)}}{2\kappa_{g_2}C},\right.$$

$$\left.\frac{1/\gamma - \kappa_{g_2}}{(1/\gamma + L_{gy,0} + L_{gy,1} \max_{\mu \in (0,1)} \|\nabla_2 g(x, \theta_\gamma^*(x,y)_\mu)\|^\rho)^2}\right), \quad where$$

$C = L_{gx,0} + L_{gx,1}M^\rho + \frac{2}{\gamma^2}$. *Then the sequence of $(x^t, y^t, \theta^t)$ generated by Algorithm 2 satisfies*

$$\min_{0 \leq t \leq T} \mathbb{E}[\|\theta^t - \theta_\gamma^*(x^t, y^t)\|] = O\left(\frac{1}{T^{1/2}}\right),$$
$$\min_{0 \leq t \leq T} \mathbb{E}[R_t(x^{t+1}, y^{t+1})] = O\left(\frac{1}{T^{1/6}}\right),$$
$$\mathbb{E}[g(x^T, y^T) - v_\gamma(x^T, y^T)] = O\left(\frac{1}{T^{1/4}}\right).$$

*Remark* 5.9. From the above Theorem 5.8, the two terms $\min_{0 \leq t \leq T} \mathbb{E}\|\theta^t - \theta_\gamma^*(x^t, y^t)\| = O\left(\frac{1}{T^{1/2}}\right)$ and $\min_{0 \leq t \leq T} \mathbb{E}[R_t(x^{t+1}, y^{t+1})] = O\left(\frac{1}{T^{\frac{1}{6}}}\right)$ shows that our S-PNGBiO algorithm has a convergence rate $O\left(\frac{1}{T^{\frac{1}{6}}}\right)$ to obtain the stationary solution of the problem (10). Adding the term

$\mathbb{E}[g(x^T, y^T) - v_\gamma(x^T, y^T)] = O\left(\frac{1}{T^{\frac{1}{4}}}\right)$ ensures that the sequence $\{g(x^t, y^t)\}$ approaches the sequence $\{v_\gamma(x^t, y^t)\}$, so the above three terms show that our S-PNGBiO algorithm has a convergence rate $O\left(\frac{1}{T^{\frac{1}{6}}}\right)$ to obtain the stationary solution of the problem (7). Thus, our S-PNGBiO method can obtain a gradient complexity of $\sum_{t=1}^T B_t = TB_t = O(\epsilon^{-9})$ for finding an $\epsilon$-stationary solution of the problem (7).

## 6. Numerical Experiments

In this section, we conduct some experiments on data hyper-cleaning, hyper-parament learning and meta learning to verify efficiency of our proposed methods. We evaluate the performance of our PNGBiO and S-PNGBiO algorithms in terms of loss and test accuracy by comparing it with several competitive baseline algorithms, including BOME (Liu et al., 2022), F²SA (Kwon et al., 2023a), PBGD (Shen & Chen, 2023), MEHA (Liu et al., 2024) and SLIP (Gong et al., 2024b). Due to a high computational cost of computing Hessian and Jacobia matrices required in the SLIP algorithm, we omit it on conducting the meta learning task at theCIFAR10 dataset (Krizhevsky et al., 2009).

### 6.1. Data Hyper-Cleaning

In this experiment, we conduct the data hyper-cleaning task (Franceschi et al., 2017; Shen & Chen, 2023) on the on MNIST (Deng, 2012) and FashionMNIST (Xiao et al., 2017) datasets, respectively. Specifically, we solve the following bilevel optimization problem

$$\min_{\lambda} \frac{1}{|S_{val}|} \sum_{i \in S_{val}} \mathcal{L}_{val}(x_i, y_i; \theta^*(\lambda))$$

$$\text{s.t. } \theta^*(\lambda) = \arg\min_{\theta} \frac{1}{|S_{tr}|} \sum_{i \in S_{tr}} \lambda_i \mathcal{L}_{tr}(x_i, y_i; \theta),$$

where $S_{val}$ and $S_{tr}$ denote validation and training datasets, respectively, and $\theta = (w, b)$ denotes the parameter of logistic regression.

In the experiment, the dataset is partitioned into a training set, a validation set, and a test set at a ratio of 1:1:2. For our algorithm, we set $\eta_t = 0.01$, $\alpha_t = \frac{0.013}{(t+1)^{0.8}}$, $\beta_t = \frac{0.011}{(t+1)^{0.8}}$ in MNIST dataset and $\eta_t = 0.01$, $\alpha_t = \frac{0.013}{(t+1)^{0.5}}$, $\beta_t = \frac{0.011}{(t+1)^{0.5}}$ in FashionMNIST dataset. The learning rate settings of other algorithms are shown in the following Table 2 and Table 3.

From Figures 2 and 3, we can find that our PNGBiO algorithm achieves a higher accuracy and a lower loss value on the test set. However, during the experiment, the convergence speed of our algorithm is slower compared to that of the F²SA. This disparity can be attributed to the fact that the F²SA employs a momentum acceleration technique, which is absent in our algorithm.

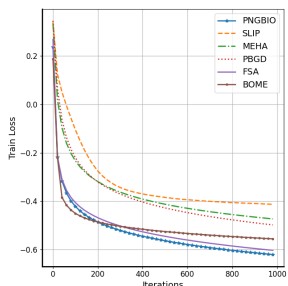 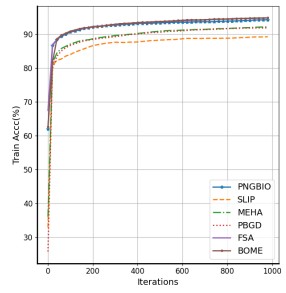 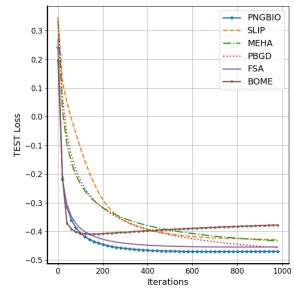 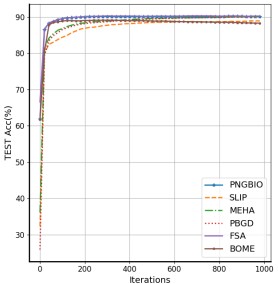

*Figure 2.* Experimental results of data hyper-cleaning on MNIST dataset.

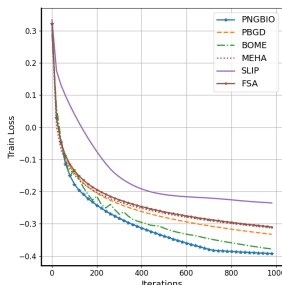 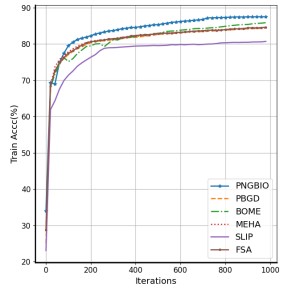 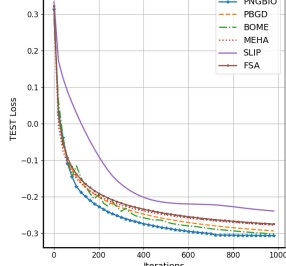 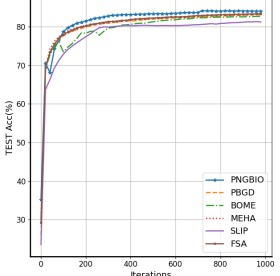

*Figure 3.* Experimental results of data hyper-cleaning on FashionMNIST dataset.

*Table 2.* Data hyper-cleaning learning rates of other algorithms on MNIST dataset.

| | BOME | F²SA | PBGD | MEHA | SLIP |
|---|---|---|---|---|---|
| $\alpha_t$ | 1.05 | 0.05 | 0.1 | 0.01 | 1.3 |
| $\beta_t$ | 1.05 | 0.05 | 0.1 | 0.03 | 0.5 |
| $\eta_t$ | 0.01 | 0.01 | 0.01 | 0.01 | - |

*Table 3.* Data hyper-cleaning learning rates of other algorithms on FashionMNIST dataset.

| | BOME | F²SA | PBGD | MEHA | SLIP |
|---|---|---|---|---|---|
| $\alpha_t$ | $\frac{0.001}{(t+1)^{1.5}}$ | $\frac{0.001}{(t+1)^{1.1}}$ | 0.1 | $\frac{0.0005}{(t+1)^{1.8}}$ | $\frac{0.001}{t+1}$ |
| $\beta_t$ | $\frac{0.01}{t+1}$ | $\frac{0.001}{(t+1)^{1.1}}$ | 0.1 | 0.001 | $\frac{0.001}{t+1}$ |
| $\eta_t$ | 0.01 | 0.01 | 0.01 | 0.01 | - |

## 6.2. Hyper-Parameter Learning

In this experiment, we conduct the hyper-parameter learning task (Franceschi et al., 2018) on MNIST (Deng, 2012) and FashionMNIST (Xiao et al., 2017) datesets, respectively. Specifically, we solve the following bilevel optimization problem

$$\min_\lambda \mathcal{L}_{S_{val}}(\lambda) = \frac{1}{|S_{val}|} \sum_{i \in S_{val}} \mathcal{L}(x_i, y_i; \omega^*(\lambda))$$

$$\text{s.t. } \omega^*(\lambda) = \arg\min_\omega \frac{1}{|S_{tr}|} \sum_{i \in S_{tr}} (\mathcal{L}(x_i, y_i; \omega) + \mathcal{R}_{\omega,\lambda}),$$

where $\omega$ denotes parameters of model, and $\lambda$ denotes parameters of regularization. Here $S_{val}$ and $S_{tr}$ are validation and training data, $\mathcal{L}$ is the cross-entropy loss, and $\mathcal{R}_{\omega,\lambda}$ is a regularizer. In the experiment, we use logistic regression. For our algorithm, we set $\eta_t = 0.01$, $\alpha_t = \frac{1.3}{(t+1)^{0.75}}$, $\beta_t = \frac{0.75}{(t+1)^{0.74}}$ in MNIST dataset and $\eta_t = 0.01$, $\alpha_t = \frac{1.5}{(t+1)^{0.45}}$, $\beta_t = \frac{1.75}{(t+1)^{0.44}}$ in FashionMNIST dataset. The learning rate settings of other algorithms are shown in the following Table 4.

Figures 4 and 5 also show that our PNGBiO algorithm achieves a higher accuracy and a lower loss value on the test set. Similarly, the convergence speed of our algorithm

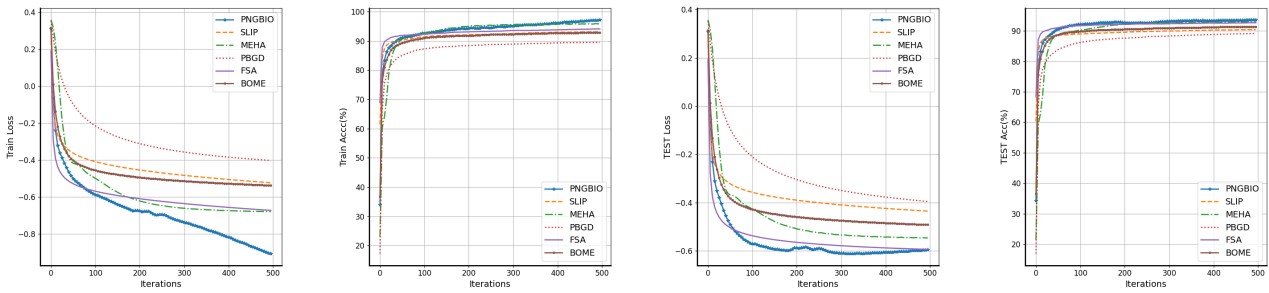

*Figure 4.* Experimental results of hyper-parameter learning on MNIST dataset.

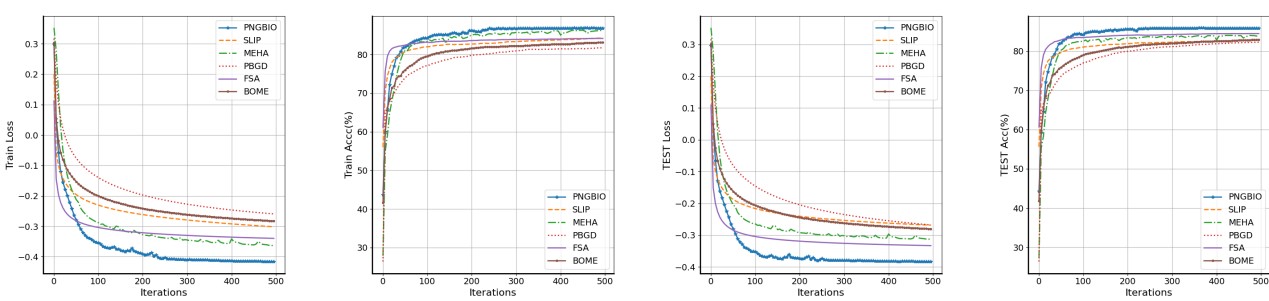

*Figure 5.* Experimental results of hyper-parameter learning on FashionMNIST dataset.

*Table 4.* Hyper-parameter learning rates of other algorithms on MNIST(FashionMNIST) dataset.

|  | BOME | F$^2$SA | PBGD | MEHA | SLIP |
|---|---|---|---|---|---|
| $\alpha_t$ | 1.05 | 0.05 | 0.1 | 0.01 | 1.3 |
| $\beta_t$ | 1.05 | 0.05 | 0.1 | 0.03 | 0.5 |
| $\eta_t$ | 0.01 | 0.01 | 0.01 | 0.01 | - |

*Table 5.* Meta learning rates of other algorithms on CIFAR10 dataset.

|  | BOME | F$^2$SA | MEHA | PBGD |
|---|---|---|---|---|
| $\alpha_t$ | 0.03 | 0.01 | 0.05 | 0.05 |
| $\beta_t$ | 0.03 | 0.02 | 0.01 | 0.01 |
| $\eta_t$ | 0.03 | 0.04 | 0.03 | 0.01 |

is slower compared to that of the F$^2$SA.

### 6.3. Meta Learning

In this experiment, we conduct the meta learning task (Franceschi et al., 2018), which can be represented by a bilevel optimization problem. Specifically, given $K$ tasks and training sets $\{\mathcal{D}_i^{tr}|i = 1, \cdots, K\}$ and validation sets $\{\mathcal{D}_i^{val}|i = 1, \cdots, K\}$, we solve the following bilevel optimization,

$$\min_{x} \frac{1}{K} \sum_{i=1}^{K} \frac{1}{|\mathcal{D}_i^{val}|} \sum_{\xi \in \mathcal{D}_i^{val}} \mathcal{L}(x, y^*(x); \xi)$$

$$\text{s.t. } y^*(x) = \arg\min_{y} \frac{1}{K} \sum_{i=1}^{K} \frac{1}{|\mathcal{D}_i^{tr}|} \sum_{\zeta \in \mathcal{D}_i^{tr}} \mathcal{L}(x, y_i; \zeta),$$

where $y = (y_1, y_2, \cdots, y_K)$. In the experiment, a fully-connected 3-layer used as a classifier for the LL, and a neural network consisting of one convolutional layer and three residual layers used as the UL.

We construct $K = 5$, where $\mathcal{D}_i^{tr}$ and $\mathcal{D}_i^{val}$ randomly sample disjoint categories from the CIFAR10 dataset (Krizhevsky et al., 2009), respectively. We use Resnet-18 (He et al., 2016) as task-shared model at the UL problem, and use a 2-layer neural network as task-specific model at the LL problem. Clearly, both UL and LL problems are non-convex. To ensure fairness, we adopt runtime as the metric. We run each method for 1800 seconds (CPU time) with minibatch size 64 for both training and validation set. For our algorithm, we set $\eta_t = 0.05$, $\alpha_t = \frac{0.085}{(t+1)^{0.5}}$, $\beta_t = \frac{0.5}{(t+1)^{0.5}}$ in CIFAR10 dataset. The learning rate setting of other algorithms are shown in the following Table 5.

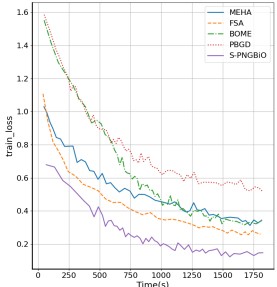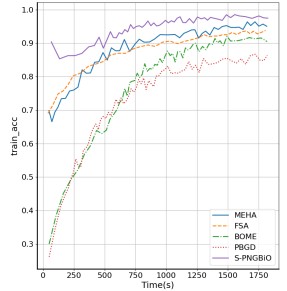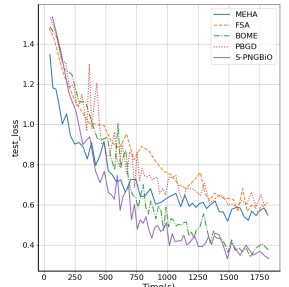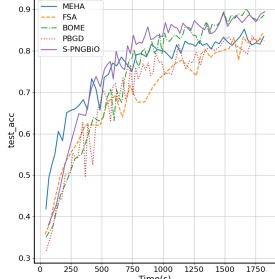

*Figure 6.* Experimental results of meta learning on CIFAR10 dataset.

Figure 6 illustrates that under the same time, our S-PNGBiO algorithm achieves a faster convergence rate than these baselines, and achieves higher accuracy and lower loss values on the test set. In other words, our algorithm S-PNGBiO consistently yields higher test accuracy and lower loss values while maintaining performance stability throughout the evaluation, thereby showcasing a robust generalization capability.

## 7. Conclusion

In this paper, we studied the generalized smooth bilevel optimization with the nonconvex lower-level objective, and proposed an efficient penalty normalized gradient (i.e., PNG-BiO) method to solve the deterministic bilevel problem (1). Moreover, we proved that our PNGBiO method has a fast convergence rate of $O\big(\frac{1}{T^{\frac{1}{4}}}\big)$ to find a stationary solution of the problem (1). Meanwhile, we proposed a stochastic version of PNGBiO (i.e.,S-PNGBiO) method, and proved that our S-PNGBiO method has a fast convergence rate of $O\big(\frac{1}{T^{\frac{1}{6}}}\big)$ to obtain a stationary solution of the stochastic bilevel problem (7). In particular, our methods do not compute expensive Hessian/Jacobian matrices and also do not require any conditions on Hessian/Jacobian matrices, and can simultaneously deal with the generalized smooth upper-level and lower-level objectives in bilevel optimization.

## Acknowledgements

We thank the anonymous reviewers for their helpful suggestions. This paper was partially supported by NSFC under Grant No. 62376125. It was also partially supported by the Fundamental Research Funds for the Central Universities NO.NJ2023032.

## Impact Statement

This paper presents work whose goal is to advance the field of Machine Learning. There are many potential societal consequences of our work, none which we feel must be specifically highlighted here.

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

# A. Appendix

In this section, we provide the detailed convergence analysis of our algorithm. We first review some useful lemmas and give some new lemmas.

**Lemma A.1.** *(Lemma A.7 of (Liu et al., 2024)) Let $\gamma \in (0, \frac{1}{2\kappa_{g_2}})$, $(x', y') \in \mathbb{R}^d \times \mathbb{R}^p$. Then for any $\kappa_{v_1} \geq \kappa_{g_1}$, $\kappa_{v_2} \geq \frac{1}{\gamma}$ and $(x, y)$ on $\mathbb{R}^d \times \mathbb{R}^p$, the following inequality holds:*

$$-v_\gamma(x, y) \leq -v_\gamma(x', y') - \langle \nabla v_\gamma(x', y'), (x, y) - (x', y') \rangle + \frac{\kappa}{2} \|(x, y) - (x', y')\|^2, \tag{18}$$

*where $\kappa := \max\{\kappa_{v_1}, \kappa_{v_2}\}$.*

**Lemma A.2.** *(Lemma 2 of (Chen et al., 2023)) $f$ is generalized smooth if and only if for any $u, u' \in \mathbb{R}^d \times \mathbb{R}^p$,*

$$\|\nabla_1 f(u') - \nabla_1 f(u)\| \leq (L_{fx,0} + L_{fx,1} \int_0^1 \|\nabla_1 f(u_\mu)\|^\rho d\mu) \|u' - u\|, \tag{19}$$

*and*

$$\|\nabla_2 f(u') - \nabla_2 f(u)\| \leq (L_{fx,0} + L_{fx,1} \int_0^1 \|\nabla_2 f(u_\mu)\|^\rho d\mu) \|u' - u\|, \tag{20}$$

*where $u_\mu = \mu u' + (1 - \mu)u$, $\mu \in [0, 1]$.*

**Lemma A.3.** *(Lemma 5 of (Chen et al., 2023)) For any $x \geq 0$, $C \in [0, 1]$, $\Delta > 0$ and $0 \leq \omega \leq \omega'$, such that $\Delta \geq \omega' - \omega$, the following inequality holds*

$$Cx^\omega \leq x^{\omega'} + C^{\frac{\omega'}{\Delta}}. \tag{21}$$

**Proposition A.4.** *Under Assumptions 5.2 and 5.3, we have*

*(i) For $\rho \in [0, 1)$, then for any $u, u' \in \mathbb{R}^d \times \mathbb{R}^p$,*

$$\frac{1}{c_t} f(u') + g(u') \leq \frac{1}{c_t} f(u) + g(u) + \langle \nabla_1 \frac{1}{c_t} f(u) + \nabla_1 g(u), u' - u \rangle + \frac{1}{2} \big( K_0 + K_1(\|\nabla_1 \frac{1}{c_t} f(u)\|^\rho + \|\nabla_1 g(u)\|^\rho)$$
$$+ 2K_2 \|u' - u\|^{\frac{\rho}{1-\rho}} \big) \|u' - u\|^2, \tag{22}$$

*and*

$$\frac{1}{c_t} f(u') + g(u') \leq \frac{1}{c_t} f(u) + g(u) + \langle \nabla_2(\frac{1}{c_t} f(u) + g(u)), u' - u \rangle + \frac{1}{2} \big( K_3 + K_4(\|\frac{1}{c_t} \nabla_2 f(u)\|^\rho + \|\nabla_2 g(u)\|^\rho)$$
$$+ 2K_5 \|u' - u\|^{\frac{\rho}{1-\rho}} \big) \|u' - u\|^2, \tag{23}$$

*where $K_0 := (\frac{1}{c_t} L_{fx,0} + L_{gx,0})(2^{\frac{\rho^2}{1-\rho}} + 1)$, $K_1 := \max\{L_{fx,1}, L_{gx,1}\} \cdot 2^{\frac{\rho^2}{1-\rho}} \cdot 3^\rho$, $K_2 := (L_{fx,1}^{\frac{1}{1-\rho}} + L_{gx,1}^{\frac{1}{1-\rho}}) \cdot 2^{\frac{\rho^2}{1-\rho}} \cdot 3^\rho (1-\rho)^{\frac{\rho}{1-\rho}}$, $K_3 := (\frac{1}{c_t} L_{fy,0} + L_{gy,0})(2^{\frac{\rho^2}{1-\rho}} + 1)$, $K_4 := \max\{L_{fy,1}, L_{gy,1}\} \cdot 2^{\frac{\rho^2}{1-\rho}} \cdot 3^\rho$ and $K_5 := (L_{fy,1}^{\frac{1}{1-\rho}} + L_{gy,1}^{\frac{1}{1-\rho}}) \cdot 2^{\frac{\rho^2}{1-\rho}} \cdot 3^\rho (1-\rho)^{\frac{\rho}{1-\rho}}$.*

*(ii) For $\rho = 1$, then for any $u, u' \in \mathbb{R}^d \times \mathbb{R}^p$,*

$$\frac{1}{c_t} f(u') + g(u') \leq \frac{1}{c_t} f(u) + g(u) + \langle \frac{1}{c_t} \nabla_1 f(u) + \nabla_1 g(u), u' - u \rangle + \frac{1}{2} \big( L_0 + L_1(\|\frac{1}{c_t} \nabla_1 f(u)\| + \|\nabla_1 g(u)\|) \big)$$
$$\cdot \|u' - u\|^2 \exp(L_1 \|u' - u\|), \tag{24}$$

*and*

$$\frac{1}{c_t} f(u') + g(u') \leq \frac{1}{c_t} f(u) + g(u) + \langle \frac{1}{c_t} \nabla_2 f(u) + \nabla_2 g(u), u' - u \rangle + \frac{1}{2} \big( L_2 + L_3(\|\frac{1}{c_t} \nabla_2 f(u)\| + \|\nabla_2 g(u)\|) \big)$$
$$\cdot \|u' - u\|^2 \exp(L_3 \|u' - u\|), \tag{25}$$

*where $L_0 := \frac{1}{c_t} L_{fx,0} + L_{gx,0}$, $L_1 := \max\{L_{fx,1}, L_{gx,1}\}$, $L_2 := \frac{1}{c_t} L_{fy,0} + L_{gy,0}$, $L_3 := \max\{L_{fy,1}, L_{gy,1}\}$.*

*Proof.* According to Assumptions 5.2 and 5.3,

(i) for $\rho \in [0, 1)$, we have

$$\frac{1}{c_t}f(u') + g(u') - \frac{1}{c_t}f(u) - g(u) - \langle \nabla_1 \frac{1}{c_t}f(u) + \nabla_1 g(u), u' - u\rangle$$

$$= \int_0^1 \left(\nabla_1 \frac{1}{c_t}f(u_\mu) + \nabla_1 g(u_\mu) - \nabla_1 \frac{1}{c_t}f(u) - \nabla_1 g(u)\right)^\mathsf{T} (u' - u)d\mu$$

$$\leq \int_0^1 \|\nabla_1 \frac{1}{c_t}f(u_\mu) + \nabla_1 g(u_\mu) - \nabla_1 \frac{1}{c_t}f(u) - \nabla_1 g(u)\|\|u' - u\|d\mu.$$

For $\mu' \in [0, 1]$, we have $u_{\mu'} = \mu'u' + (1 - \mu')u$, such that $\mu'u_\mu + (1 - \mu')u = \mu'\mu u' + (1 - \mu'\mu)u = u_{\mu'\mu}$. Therefore, we obtain

$$\|\frac{1}{c_t}\nabla_1 f(u_\mu) - \frac{1}{c_t}\nabla_1 f(u)\|$$

$$\leq \left(\frac{1}{c_t}L_{fx,0} + L_{fx,1}\int_0^1 \|\frac{1}{c_t}\nabla_1 f(u_{\mu'\mu})\|^\rho d\mu\right)\|u_\mu - u\|$$

$$= \left(\frac{1}{c_t}L_{fx,0}\mu + L_{fx,1}\int_0^1 \|\frac{1}{c_t}\nabla_1 f(u_{\mu'\mu})\|^\rho \mu d\mu\right)\|u' - u\|. \tag{26}$$

Let $H(\mu') := \frac{1}{c_t}L_{fx,0}\mu' + L_{fx,1}\int_0^1 \|\frac{1}{c_t}\nabla_1 f(u_{\mu'\mu})\|^\rho \mu' d\mu = \frac{1}{c_t}L_{fx,0}\mu' + L_{fx,1}\int_0^{\mu'} \|\frac{1}{c_t}\nabla_1 f(u_z)\|^\rho dz$, we have the derivative formula

$$\partial H(\mu') \leq \frac{1}{c_t}L_{fx,0} + L_{fx,1}\|\frac{1}{c_t}\nabla_1 f(u_{\mu'}) - \frac{1}{c_t}\nabla_1 f(u)\|^\rho + L_{fx,1}\|\frac{1}{c_t}\nabla_1 f(u)\|^\rho$$

$$\leq \frac{1}{c_t}L_{fx,0} + L_{fx,1}\|u' - u\|^\rho H(\mu')^\rho + L_{fx,1}\|\frac{1}{c_t}\nabla_1 f(u)\|^\rho$$

$$\leq 3L_{fx,1}\left(\frac{1}{3}\|u' - u\|H(\mu') + \frac{1}{3}\|\frac{1}{c_t}\nabla_1 f(u)\| + \frac{(\frac{1}{c_t}L_{fx,0})^{\frac{1}{\rho}}}{3L_{fx,1}^{\frac{1}{\rho}}}\right)^\rho, \tag{27}$$

where the last inequality applies Jensen's inequality to the concave function $h(x) = x^\rho$. Rearranging the above inequality yields that

$$3^{1-\rho}L_{fx,1}(1 - \rho)\|u' - u\|$$

$$\geq (1 - \rho)\|u' - u\|\left(\|u' - u\|H(\mu') + \|\frac{1}{c_t}\nabla_1 f(u)\| + \frac{(\frac{1}{c_t}L_{fx,0})^{\frac{1}{\rho}}}{L_{fx,1}^{\frac{1}{\rho}}}\right)^{-\rho}\partial H(\mu')$$

$$= \frac{d}{d\mu'}\left(\|u' - u\|H(\mu') + \|\frac{1}{c_t}\nabla_1 f(u)\| + \frac{(\frac{1}{c_t}L_{fx,0})^{\frac{1}{\rho}}}{L_{fx,1}^{\frac{1}{\rho}}}\right)^{1-\rho}.$$

Integrating the above inequality over $\mu' \in [0, \mu]$ yields that

$$\left(\|u' - u\|H(\mu') + \|\frac{1}{c_t}\nabla_1 f(u)\| + \frac{(\frac{1}{c_t}L_{fx,0})^{\frac{1}{\rho}}}{3L_{fx,1}^{\frac{1}{\rho}}}\right)^{1-\rho}$$

$$\leq 3^{1-\rho}L_{fx,1}(1 - \rho)\|u' - u\|\mu + \left(\|u' - u\|H(0) + \|\frac{1}{c_t}\nabla_1 f(u)\| + \frac{(\frac{1}{c_t}L_{fx,0})^{\frac{1}{\rho}}}{L_{fx,1}^{\frac{1}{\rho}}}\right)^{1-\rho}$$

$$\leq 2^\rho\left(3(L_{fx,1}(1 - \rho)\|u' - u\|\mu)^{\frac{1}{1-\rho}} + \|\frac{1}{c_t}\nabla_1 f(u)\| + \frac{(\frac{1}{c_t}L_{fx,0})^{\frac{1}{\rho}}}{L_{fx,1}^{\frac{1}{\rho}}}\right)^{1-\rho},$$

where the last inequality follows from $H(0) = 0$ and Jensen's inequality to the concave function $h(x) = x^{1-\rho}$. Thus, we can obtain

$$\|u' - u\| H(\mu)$$

$$\leq 2^{\frac{\rho}{1-\rho}} \left( 3(L_{fx,1}(1-\rho)\|u'-u\|\mu)^{\frac{1}{1-\rho}} + \|\frac{1}{c_t}\nabla_1 f(u)\| + \frac{(\frac{1}{c_t}L_{fx,0})^{\frac{1}{\rho}}}{L_{fx,1}^{\frac{1}{\rho}}} \right) - \|\frac{1}{c_t}\nabla_1 f(u)\| - \frac{L_{fx,0}^{\frac{1}{\rho}}}{(c_t L_{fx,1})^{\frac{1}{\rho}}},$$

such that

$$\|\frac{1}{c_t}\nabla_1 f(u_\mu)\| \leq \|\frac{1}{c_t}\nabla_1 f(u)\| + \|\frac{1}{c_t}\nabla_1 f(u_\mu) - \frac{1}{c_t}\nabla_1 f(u)\|$$

$$\leq \|\frac{1}{c_t}\nabla_1 f(u)\| + \|u' - u\| H(\mu)$$

$$\leq 2^{\frac{\rho}{1-\rho}} \left( 3(L_{fx,1}(1-\rho)\|u'-u\|\mu)^{\frac{1}{1-\rho}} + \|\frac{1}{c_t}\nabla_1 f(u)\| + \frac{L_{fx,0}^{\frac{1}{\rho}}}{(c_t L_{fx,1})^{\frac{1}{\rho}}} \right).$$

Then we have

$$\|\frac{1}{c_t}\nabla_1 f(u') - \frac{1}{c_t}\nabla_1 f(u)\|$$

$$\leq (\frac{1}{c_t}L_{fx,0} + \max_{\mu\in[0,1]}\|\frac{1}{c_t}\nabla_1 f(u_\mu)\|^\rho)\|u'-u\|$$

$$\leq \left( \frac{1}{c_t}L_{fx,0} + L_{fx,1} 2^{\frac{\rho^2}{1-\rho}} \left( 3(L_{fx,1}(1-\rho)\|u'-u\|)^{\frac{1}{1-\rho}} + \|\frac{1}{c_t}\nabla_1 f(u)\| + \frac{L_{fx,0}^{\frac{1}{\rho}}}{(c_t L_{fx,1})^{\frac{1}{\rho}}} \right)^\rho \right)\|u'-u\|$$

$$\leq \left( \frac{1}{c_t}L_{fx,0} + L_{fx,1} 2^{\frac{\rho^2}{1-\rho}} \left( 3^\rho(L_{fx,1}(1-\rho)\|u'-u\|)^{\frac{\rho}{1-\rho}} + \|\frac{1}{c_t}\nabla_1 f(u)\|^\rho + \frac{L_{fx,0}}{(c_t L_{fx,1})} \right) \right)\|u'-u\|$$

$$\leq \left( \frac{1}{c_t}L_{fx,0}(1 + 2^{\frac{\rho^2}{1-\rho}}) + L_{fx,1} \cdot 2^{\frac{\rho^2}{1-\rho}} \cdot 3^\rho \|\frac{1}{c_t}\nabla_1 f(u)\|^\rho + L_{fx,1}^{\frac{1}{1-\rho}} \cdot 2^{\frac{\rho^2}{1-\rho}} \cdot 3^\rho(1-\rho)^{\frac{\rho}{1-\rho}}\|u'-u\|^{\frac{\rho}{1-\rho}} \right)\|u'-u\|.$$

Similarly, we have

$$\|\nabla_1 g(u') - \nabla_1 g(u)\|$$

$$\leq \left( L_{gx,0}(1 + 2^{\frac{\rho^2}{1-\rho}}) + L_{gx,1} \cdot 2^{\frac{\rho^2}{1-\rho}} \cdot 3^\rho \|\nabla_1 g(u)\| + L_{gx,1}^{\frac{1}{1-\rho}} \cdot 2^{\frac{\rho^2}{1-\rho}} \cdot 3^\rho(1-\rho)^{\frac{\rho}{1-\rho}}\|u'-u\|^{\frac{\rho}{1-\rho}} \right)\|u'-u\|.$$

Therefore, we can obtain

$$\int_0^1 \|\nabla_1 \frac{1}{c_t}f(u_\mu) + \nabla_1 g(u_\mu) - \nabla_1 \frac{1}{c_t}f(u) - \nabla_1 g(u)\|\|u'-u\|d\mu$$

$$\leq \int_0^1 \|u_\mu - u\|(\frac{1}{c_t}L_{fx,0}(2^{\frac{\rho^2}{1-\rho}} + 1) + L_{fx,1} \cdot 2^{\frac{\rho^2}{1-\rho}} \cdot 3^\rho \frac{1}{c_t}\|\nabla_1 f(u)\|^\rho + L_{fx,1}^{\frac{1}{1-\rho}} \cdot 2^{\frac{\rho^2}{1-\rho}} \cdot 3^\rho(1-\rho)^{\frac{\rho}{1-\rho}}\|u'-u\|^{\frac{\rho}{1-\rho}}$$

$$+ L_{gx,0}(2^{\frac{\rho^2}{1-\rho}} + 1) + L_{gx,1} \cdot 2^{\frac{\rho^2}{1-\rho}} \cdot 3^\rho \|\nabla_1 g(u)\|^\rho + L_{gx,1}^{\frac{1}{1-\rho}} \cdot 2^{\frac{\rho^2}{1-\rho}} \cdot 3^\rho(1-\rho)^{\frac{\rho}{1-\rho}}\|u'-u\|^{\frac{\rho}{1-\rho}})\|u'-u\|d\mu$$

$$= \int_0^1 \mu\|u'-u\|^2(\frac{1}{c_t}L_{fx,0}(2^{\frac{\rho^2}{1-\rho}} + 1) + L_{fx,1} \cdot 2^{\frac{\rho^2}{1-\rho}} \cdot 3^\rho \frac{1}{c_t}\|\nabla_1 f(u)\|^\rho + L_{fx,1}^{\frac{1}{1-\rho}} \cdot 2^{\frac{\rho^2}{1-\rho}} \cdot 3^\rho(1-\rho)^{\frac{\rho}{1-\rho}}\|u'-u\|^{\frac{\rho}{1-\rho}}$$

$$+ L_{gx,0}(2^{\frac{\rho^2}{1-\rho}} + 1) + L_{gx,1} \cdot 2^{\frac{\rho^2}{1-\rho}} \cdot 3^\rho \|\nabla_1 g(u)\|^\rho + L_{gx,1}^{\frac{1}{1-\rho}} \cdot 2^{\frac{\rho^2}{1-\rho}} \cdot 3^\rho(1-\rho)^{\frac{\rho}{1-\rho}}\|u'-u\|^{\frac{\rho}{1-\rho}})d\mu$$

$$= \frac{1}{2}\|u'-u\|^2 \left( (\frac{1}{c_t}L_{fx,0} + L_{gx,0})(2^{\frac{\rho^2}{1-\rho}} + 1) + \max\{L_{fx,1}, L_{gx,1}\} \cdot 2^{\frac{\rho^2}{1-\rho}} \cdot 3^\rho(\|\frac{1}{c_t}\nabla_1 f(u)\|^\rho + \|\nabla_1 g(u)\|^\rho) + \right.$$

$$\left. (L_{fx,1}^{\frac{1}{1-\rho}} + L_{gx,1}^{\frac{1}{1-\rho}}) \cdot 2^{\frac{\rho^2}{1-\rho}} \cdot 3^\rho(1-\rho)^{\frac{\rho}{1-\rho}}\|u'-u\|^{\frac{2-\rho}{1-\rho}} \right) \int_0^1 \mu^{\frac{1}{1-\rho}}d\mu$$

$$\leq \frac{1}{2}\|u'-u\|^2 \left( K_0 + K_1(\|\frac{1}{c_t}\nabla_1 f(u)\|^\rho + \|\nabla_1 g(u)\|^\rho) + 2K_2\|u'-u\|^{\frac{\rho}{1-\rho}} \right),$$

where $K_0 := (\frac{1}{c_t}L_{fx,0} + L_{gx,0})(2^{\frac{\rho^2}{1-\rho}} + 1)$, $K_1 := \max\{L_{fx,1}, L_{gx,1}\} \cdot 2^{\frac{\rho^2}{1-\rho}} \cdot 3^\rho$ and $K_2 := (L_{fx,1}^{\frac{1}{1-\rho}} + L_{gx,1}^{\frac{1}{1-\rho}}) \cdot 2^{\frac{\rho^2}{1-\rho}} \cdot 3^\rho(1-\rho)^{\frac{\rho}{1-\rho}}$.

Similarly, (23) can be proved.

(ii) For $\rho = 1$, we prove (24) as follows.

$$\frac{1}{c_t}f(u') + g(u') - \frac{1}{c_t}f(u) - g(u) - \langle\nabla_1\frac{1}{c_t}f(u) + \nabla_1 g(u), u' - u\rangle$$

$$= \int_0^1 (\nabla_1\frac{1}{c_t}f(u_\mu) + \nabla_1 g(u_\mu) - \nabla_1\frac{1}{c_t}f(u) - \nabla_1 g(u))^\mathsf{T}(u' - u)d\mu$$

$$\leq \int_0^1 \|\nabla_1\frac{1}{c_t}f(u_\mu) + \nabla_1 g(u_\mu) - \nabla_1\frac{1}{c_t}f(u) - \nabla_1 g(u)\|\|u' - u\|d\mu.$$

When $\rho = 1$, by (27), we have

$$\partial H(\mu') \leq \frac{1}{c_t}L_{fx,0} + L_{fx,1}\|u' - u\|H(\mu') + L_{fx,1}\|\frac{1}{c_t}\nabla_1 f(u)\|,$$

where $H(\mu') := \frac{1}{c_t}L_{fx,0}\mu' + L_{fx,1}\int_0^{\mu'} \|\nabla_1\frac{1}{c_t}f(u_\mu)\|d\mu$. It follows that

$$L_{fx,1}\|u' - u\| \leq \frac{L_{fx,1}\|u' - u\|\partial H(\mu')}{\frac{1}{c_t}L_{fx,0} + L_{fx,1}\|u' - u\|H(\mu') + L_{fx,1}\|\nabla_1\frac{1}{c_t}f(u)\|}$$

$$= \frac{d}{d\mu'}\ln(\frac{1}{c_t}L_{fx,0} + L_{fx,1}\|u' - u\|H(\mu') + L_{fx,1}\|\nabla_1\frac{1}{c_t}f(u)\|),$$

and integrating the inequality over $\mu' \in [0, \mu]$, we have

$$\ln(\frac{1}{c_t}L_{fx,0} + L_{fx,1}\|u' - u\|H(\mu') + L_{fx,1}\|\nabla_1\frac{1}{c_t}f(u)\|) \leq \ln(\frac{1}{c_t}L_{fx,0} + L_{fx,1}\|\nabla_1\frac{1}{c_t}f(u)\|) + L_{fx,1}\|u' - u\|,$$

which implies that

$$L_{fx,1}\|u' - u\|H(\mu) \leq (\frac{1}{c_t}L_{fx,0} + L_{fy,1}\|\nabla_1\frac{1}{c_t}f(u)\|)\exp(L_{fx,1}\|u' - u\|) - \frac{1}{c_t}L_{fx,0} - L_{fx,1}\|\nabla_1\frac{1}{c_t}f(u)\|.$$

Then, we have

$$\|\nabla_1\frac{1}{c_t}f(u_\mu)\| \leq \|\nabla_1\frac{1}{c_t}f(u)\| + \|\nabla_1\frac{1}{c_t}f(u_\mu) - \nabla_1\frac{1}{c_t}f(u)\|$$

$$\leq \|\nabla_1\frac{1}{c_t}f(u)\| + \frac{1}{L_{fx,1}}\left((\frac{1}{c_t}L_{fx,0} + L_{fx,1}\|\nabla_1\frac{1}{c_t}f(u)\|)\exp(L_{fx,1}\|u' - u\|) - \frac{L_{fx,0}}{c_t}\right.$$

$$\left. - L_{fx,1}\|\nabla_1\frac{1}{c_t}f(u)\|\right)$$

$$= \left(\frac{L_{fx,0}}{c_t L_{fx,1}} + \|\nabla_1\frac{1}{c_t}f(u)\|\right)\exp(L_{fx,1}\|u' - u\|) - \frac{L_{fx,0}}{c_t L_{fx,1}},$$

and

$$\|\nabla_1 f(u') - \nabla_1 f(u)\| \leq \left(\frac{1}{c_t}L_{fx,0} + L_{fx,1}\|\nabla_1\frac{1}{c_t}f(u)\|\right)\exp(L_{fx,1}\|u' - u\|)\|u' - u\|.$$

Similarly, we have

$$\|\nabla_1 g(u') - \nabla_1 g(u)\| \leq \left(L_{gx,0} + L_{gx,1}\|\nabla_1 g(u)\|\right)\exp(L_{gx,1}\|u' - u\|)\|u' - u\|.$$

Therefore, we can obtain

$$\int_0^1 \|\nabla_1 \frac{1}{c_t} f(u_\mu) + \nabla_1 g(u_\mu) - \nabla_1 \frac{1}{c_t} f(u) - \nabla_1 g(u)\| \|u' - u\| d\mu$$

$$\leq \int_0^1 \|u_\mu - u\| \left( \left( \frac{1}{c_t} L_{fx,0} + L_{fx,1} \|\nabla_1 \frac{1}{c_t} f(u)\| \right) \exp(L_{fx,1} \|u' - u\|) + \left( L_{gx,0} + L_{gx,1} \|\nabla_1 g(u)\| \right) \right.$$

$$\left. \exp(L_{gx,1} \|u' - u\|) \right) \|u' - u\| d\mu$$

$$\leq \int_0^1 \mu \|u' - u\|^2 \left( \left( \frac{1}{c_t} L_{fx,0} + L_{fx,1} \|\nabla_1 \frac{1}{c_t} f(u)\| \right) \exp(L_{fx,1} \|u' - u\|) + \left( L_{gx,0} + L_{gx,1} \|\nabla_1 g(u)\| \right) \right.$$

$$\left. \exp(L_{gx,1} \|u' - u\|) \right) d\mu$$

$$= \frac{1}{2} \|u' - u\|^2 \left( \left( \frac{1}{c_t} L_{fx,0} + L_{fx,1} \|\nabla_1 \frac{1}{c_t} f(u)\| \right) \exp(L_{fx,1} \|u' - u\|) + \left( L_{gx,0} + L_{gx,1} \|\nabla_1 g(u)\| \right) \right.$$

$$\left. \exp(L_{gx,1} \|u' - u\|) \right)$$

$$\leq \frac{1}{2} \|u' - u\|^2 (L_0 + L_1 (\|\nabla_1 \frac{1}{c_t} f(u)\| + \|\nabla_1 g(u)\|)) \exp(L_1 \|u' - u\|),$$

where $L_0 := \frac{1}{c_t} L_{fx,0} + L_{gx,0}$, $L_1 := \max\{L_{fx,1}, L_{gx,1}\}$.

Similarly, (25) can be proved. $\qquad\square$

**Lemma A.5.** *Let* $\gamma \in (0, \frac{1}{\kappa_{g_2}})$. *Then, there exists* $L_\theta > 0$ *such that for any* $(x,y), (x',y') \in \mathbb{R}^d \times \mathbb{R}^p$, *the following inequality holds:*

$$\|\theta_\gamma^*(x,y) - \theta_\gamma^*(x',y')\| \leq L_\theta \|(x,y) - (x',y')\|, \tag{28}$$

*where* $L_\theta = \frac{s}{1 - \sqrt{1 - s(\frac{1}{\gamma} - \kappa_{g_2})}} \max \left\{ L_{gy,0} + L_{gy,1} \max_{\mu \in [0,1]} \|\nabla_2 g(x_\mu, \theta_\gamma^*(x,y))\|^\rho, \frac{1}{\gamma} \right\}$, $x_\mu := \mu x + (1 - \mu)x'$.

*Proof.* Since $\theta_\gamma^*(x,y)$ is the optimal solution of $\min_{\theta \in \mathbb{R}^p} g(x,\theta) + \frac{1}{2\gamma} \|\theta - y\|^2$, we have for any $(x,y), (x',y') \in \mathbb{R}^d \times \mathbb{R}^p$,

$$\nabla_2 g(x, \theta_\gamma^*(x,y)) + \frac{1}{\gamma}(\theta_\gamma^*(x,y) - y) = 0,$$

$$\nabla_2 g(x', \theta_\gamma^*(x',y')) + \frac{1}{\gamma}(\theta_\gamma^*(x',y') - y') = 0.$$

Then we can obtain

$$\|\theta_\gamma^*(x,y) - \theta_\gamma^*(x',y')\|$$

$$= \|\theta_\gamma^*(x,y) - s(\nabla_2 g(x, \theta_\gamma^*(x,y)) + \frac{1}{\gamma}(\theta_\gamma^*(x,y) - y)) - \theta_\gamma^*(x',y') + s(\nabla_2 g(x', \theta_\gamma^*(x',y')) + \frac{1}{\gamma}(\theta_\gamma^*(x',y') - y'))\|$$

$$\leq \|\theta_\gamma^*(x,y) - s(\nabla_2 g(x, \theta_\gamma^*(x,y)) + \frac{1}{\gamma}(\theta_\gamma^*(x,y) - y)) - \theta_\gamma^*(x',y') + s(\nabla_2 g(x, \theta_\gamma^*(x',y')) + \frac{1}{\gamma}(\theta_\gamma^*(x',y') - y))\|$$

$$+ \|\theta_\gamma^*(x',y') - s(\nabla_2 g(x, \theta_\gamma^*(x',y')) + \frac{1}{\gamma}(\theta_\gamma^*(x',y') - y)) - \theta_\gamma^*(x',y') + s(\nabla_2 g(x', \theta_\gamma^*(x',y'))$$

$$+ \frac{1}{\gamma}(\theta_\gamma^*(x',y') - y'))\|. \tag{29}$$

Due to $\gamma \in (0, \frac{1}{\kappa_{g_2}})$, the function $g(x,\theta) + \frac{1}{2\gamma} \|\theta - y\|^2$ is $(\frac{1}{\gamma} - \kappa_{g_2})$-strongly convex respect to $\theta \in \mathbb{R}^p$, it follows that

$$\langle \nabla_2 g(x, \theta_\gamma^*(x,y)) + \frac{1}{\gamma}(\theta_\gamma^*(x,y) - y) - \nabla_2 g(x, \theta_\gamma^*(x',y')) - \frac{1}{\gamma}(\theta_\gamma^*(x',y') - y), \theta_\gamma^*(x,y) - \theta_\gamma^*(x',y') \rangle$$

$$\geq (\frac{1}{\gamma} - \kappa_{g_2}) \|\theta_\gamma^*(x,,y) - \theta_\gamma^*(x',y')\|^2.$$

Given $0 < s \leq \frac{1/\gamma - \kappa_{g_2}}{(1/\gamma + L_{gy,0} + L_{gy,1} \max_{\mu \in (0,1)} \|\nabla_2 g(x, \theta_\gamma^*(x,y)_\mu)\|^\rho)^2}$, we have

$$\|\theta_\gamma^*(x,y) - s(\nabla_2 g(x, \theta_\gamma^*(x,y)) + \frac{1}{\gamma}(\theta_\gamma^*(x,y) - y)) - \theta_\gamma^*(x',y') + s(\nabla_2 g(x, \theta_\gamma^*(x',y')) + \frac{1}{\gamma}(\theta_\gamma^*(x',y') - y))\|^2$$

$$\leq \left(1 - 2s(\frac{1}{\gamma} - \kappa_{g_2}) + s^2(\frac{1}{\gamma} + L_{gy,0} + L_{gy,1} \max_{\mu \in (0,1)} \|\nabla_2 g(x, \theta_\gamma^*(x,y)_\mu)\|^\rho)^2\right) \|\theta_\gamma^*(x,y) - \theta_\gamma^*(x',y')\|^2$$

$$\leq \left(1 - s(\frac{1}{\gamma} - \kappa_{g_2})\right) \|\theta_\gamma^*(x,y) - \theta_\gamma^*(x',y')\|^2, \tag{30}$$

and

$$\|\theta_\gamma^*(x',y') - s(\nabla_2 g(x, \theta_\gamma^*(x',y')) + \frac{1}{\gamma}(\theta_\gamma^*(x',y') - y)) - \theta_\gamma^*(x',y') + s(\nabla_2 g(x', \theta_\gamma^*(x',y')) + \frac{1}{\gamma}(\theta_\gamma^*(x',y') - y'))\|$$

$$\leq s\|x - x'\|(L_{gy,0} + L_{gy,1} \max_{\mu \in [0,1]} \|\nabla_2 g(x_\mu, \theta_\gamma^*(x',y'))\|^\rho) + \frac{s}{\gamma}\|y - y'\|, \tag{31}$$

where $x_\mu := \mu x + (1 - \mu)x'$, $\theta_\gamma^*(x,y)_\mu := \mu \theta_\gamma^*(x,y) + (1 - \mu)\theta_\gamma^*(x',y')$. By combining the above inequalities (29), (30) and (31), we have

$$\|\theta_\gamma^*(x,y) - \theta_\gamma^*(x',y')\|$$
$$\leq \sqrt{1 - s(\frac{1}{\gamma} - \kappa_{g_2})} \|\theta_\gamma^*(x,y) - \theta_\gamma^*(x',y')\| + s((L_{gy,0} + L_{gy,1} \max_{\mu \in [0,1]} \|\nabla_2 g(x_\mu, \theta_\gamma^*(x,y))\|^\rho)\|x - x'\| + \frac{1}{\gamma}\|y - y'\|).$$

Finally we can obtain

$$\|\theta_\gamma^*(x,y) - \theta_\gamma^*(x',y')\|$$
$$\leq \frac{s}{\left(1 - \sqrt{1 - s(\frac{1}{\gamma} - \kappa_{g_2})}\right)} ((L_{gy,0} + L_{gy,1} \max_{\mu \in [0,1]} \|\nabla_2 g(x_\mu, \theta_\gamma^*(x,y))\|^\rho)\|x - x'\| + \frac{1}{\gamma}\|y - y'\|)$$
$$\leq L_\theta \|(x,y) - (x',y')\|,$$

where $L_\theta = \frac{s}{1 - \sqrt{1 - s(\frac{1}{\gamma} - \kappa_{g_2})}} \max\left\{L_{gy,0} + L_{gy,1} \max_{\mu \in [0,1]} \|\nabla_2 g(x_\mu, \theta_\gamma^*(x,y))\|^\rho, \frac{1}{\gamma}\right\}$. It implies that the solution $\theta_\gamma^*$ is generally smooth with $(x, y)$.

$\square$

**Lemma A.6.** *Under Assumptions 5.2 and 5.3, let $\Psi_{c_t}(x,y) = \frac{1}{c_t} f(x,y) + g(x,y) - v_\gamma(x,y)$ with $c_{t+1} \geq c_t > 0$ for all $t \geq 0$, we have*

$$\|\nabla_1 \Psi_{c_t}(x^{t+1}, y^{t+1}) - \nabla_1 \Psi_{c_t}(x^t, y^t)\| \leq L_{\Psi_1} \|(x^{t+1}, y^{t+1}) - (x^t, y^t)\|, \tag{32}$$

$$\|\nabla_2 \Psi_{c_t}(x^{t+1}, y^{t+1}) - \nabla_2 \Psi_{c_t}(x^t, y^t)\| \leq L_{\Psi_2} \|(x^{t+1}, y^{t+1}) - (x^t, y^t)\|. \tag{33}$$

*where $L_{\Psi_1} = \frac{1}{c_t} L_{fx,0} + L_{fx,1} \max_{\mu \in [0,1]} \|\frac{1}{c_t} \nabla_1 f(x_\mu, y_\mu)\|^\rho + L_{gx,0} + L_{gx,1} \max_{\mu \in [0,1]} \|\nabla_1 g(x_\mu, y_\mu)\|^\rho + (L_{gx,0} + L_{gx,1} \max_{\mu \in [0,1]} \|\nabla_1 g(x_\mu, \theta_\gamma^*(x_\mu, y_\mu))\|^\rho (1 + L_\theta))$ and $L_{\Psi_2} = \frac{1}{c_t} L_{fy,0} + L_{fy,1} \max_{\mu \in [0,1]} \|\frac{1}{c_t} \nabla_2 f(x_\mu, y_\mu)\|^\rho + L_{gy,0} + L_{gy,1} \max_{\mu \in [0,1]} \|\nabla_2 g(x_\mu, y_\mu)\|^\rho + \frac{1 + L_\theta}{\gamma}.*

*Proof.* When $\rho \in [0, 1]$, we have

$$
\begin{aligned}
&\|\nabla_1 \Psi_{c_t}(x^{t+1}, y^{t+1}) - \nabla_1 \Psi_{c_t}(x^t, y^t)\| \\
=&\|\frac{1}{c_t}\nabla_1 f(x^{t+1}, y^{t+1}) - \frac{1}{c_t}\nabla_1 f(x^t, y^t) + \nabla_1 g(x^{t+1}, y^{t+1}) - \nabla_1 g(x^t, y^t) - \nabla_1 g(x^{t+1}, \theta_\gamma^*(x^{t+1}, y^{t+1})) \\
&+ \nabla_1 g(x^t, \theta_\gamma^*(x^t, y^t))\| \\
\leq&\|\frac{1}{c_t}\nabla_1 f(x^{t+1}, , y^{t+1}) - \frac{1}{c_t}\nabla_1 f(x^t, y^t)\| + \|\nabla_1 g(x^{t+1}, y^{t+1}) - \nabla_1 g(x^t, y^t)\| + \|\nabla_1 g(x^{t+1}, \theta_\gamma^*(x^{t+1}, y^{t+1})) \\
&- \nabla_1 g(x^t, \theta_\gamma^*(x^t, y^t))\| \\
\leq&\left(\frac{1}{c_t}L_{fx,0} + L_{fx,1}\max_{\mu\in[0,1]}\|\frac{1}{c_t}\nabla_1 f(x_\mu, y_\mu)\|^\rho) + L_{gx,0} + L_{gx,1}\max_{\mu\in[0,1]}\|\nabla_1 g(x_\mu, y_\mu)\|^\rho\right)\|(x^{t+1}, y^{t+1}) - (x^t, y^t)\| \\
&+ (L_{gx,0} + L_{gx,1}\max_{\mu\in[0,1]}\|\nabla_1 g(x_\mu, \theta_\gamma^*(x_\mu, y_\mu))\|^\rho(1 + L_\theta))\|(x^{t+1}, y^{t+1}) - (x^t, y^t)\| \\
=&\left(\frac{1}{c_t}L_{fx,0} + L_{fx,1}\max_{\mu\in[0,1]}\|\frac{1}{c_t}\nabla_1 f(x_\mu, y_\mu)\|^\rho + L_{gx,0} + L_{gx,1}\max_{\mu\in[0,1]}\|\nabla_1 g(x_\mu, y_\mu)\|^\rho + (L_{gx,0}\right. \\
&\left.+ L_{gx,1}\max_{\mu\in[0,1]}\|\nabla_1 g(x_\mu, \theta_\gamma^*(x_\mu, y_\mu))\|^\rho(1 + L_\theta))\right)\|(x^{t+1}, y^{t+1}) - (x^t, y^t)\|,
\end{aligned}
\tag{34}
$$

where the first inequality follows from triangle inequality and the second inequality follows from Assumptions 5.2, 5.3 and Lemma A.5. Similarly, we have

$$
\begin{aligned}
&\|\nabla_2 \Psi_{c_t}(x^{t+1}, y^{t+1}) - \nabla_2 \Psi_{c_t}(x^t, y^t)\| \\
=&\left\|\frac{1}{c_t}\nabla_2 f(x^{t+1}, , y^{t+1}) - \frac{1}{c_t}\nabla_2 f(x^t, y^t) + \nabla_2 g(x^{t+1}, y^{t+1}) - \nabla_2 g(x^t, y^t) + \frac{\theta_\gamma^*(x^{t+1}, y^{t+1}) - y^{t+1}}{\gamma} - \frac{\theta_\gamma^*(x^t, y^t) - y^t}{\gamma}\right\| \\
\leq&\left(\frac{1}{c_t}L_{fy,0} + L_{fy,1}\max_{\mu\in[0,1]}\|\frac{1}{c_t}\nabla_2 f(x_\mu, y_\mu)\|^\rho + L_{gy,0} + L_{gy,1}\max_{\mu\in[0,1]}\|\nabla_2 g(x_\mu, y_\mu)\|^\rho\right)\|(x^{t+1}, y^{t+1}) - (x^t, y^t)\| \\
&+ \frac{1 + L_\theta}{\gamma}\|(x^{t+1}, y^{t+1}) - (x^t, y^t)\| \\
=&\left(\frac{1}{c_t}L_{fy,0} + L_{fy,1}\max_{\mu\in[0,1]}\|\frac{1}{c_t}\nabla_2 f(x_\mu, y_\mu)\|^\rho + L_{gy,0} + L_{gy,1}\max_{\mu\in[0,1]}\|\nabla_2 g(x_\mu, y_\mu)\|^\rho + \frac{1 + L_\theta}{\gamma}\right) \\
&\|(x^{t+1}, y^{t+1}) - (x^t, y^t)\|,
\end{aligned}
\tag{35}
$$

where the first inequality follows from triangle inequality and the second inequality follows from Assumptions 5.2, 5.3 and Lemma A.5. $\square$

## A.1. Deterministic Setting

**Lemma A.7.** *Given $\gamma \in (0, \frac{1}{2\kappa_{g_2}})$ and $\eta_t \in \left(0, \frac{1/\gamma - \kappa_{g_2}}{(1/\gamma + L_{gy,0} + L_{gy,1}\max_{\mu\in(0,1)}\|\nabla_2 g(x, \theta_\gamma^*(x,y)_\mu)\|^\rho)^2}\right)$, the sequence $\{x^t, y^t, \theta^t\}$ generated by Algorithm 1 satisfies*

$$
\|\theta^{t+1} - \theta^*(x^t, y^t)\| \leq \sigma_t\|\theta^t - \theta_\gamma^*(x^t, y^t)\|,
\tag{36}
$$

*where $\sigma_t = \sqrt{1 - \eta_t(1/\gamma - \kappa_{g_2})}$.*

*Proof.* Let $\theta_\gamma^*(x^t, y^t)$ be the optimal solution of $\min_{\theta\in\mathbb{R}^p} g(x^t, \theta) + \frac{1}{2\gamma}\|\theta - y^t\|$, we have

$$
\theta_\gamma^*(x^t, y^t) = \theta_\gamma^*(x^t, y^t) - \eta_t(\nabla_2 g(x, \theta_\gamma^*(x^t, y^t)) + \frac{1}{\gamma}(\theta_\gamma^*(x^t, y^t) - y^t)).
$$

By using the update rule for $\theta^{t+1}$ in the Algorithm 1, we can obtain

$$\|\theta^{t+1} - \theta_\gamma^*(x^t, y^t)\|$$

$$\leq \|\theta^t - \eta_t(\nabla_2 g(x^t, \theta^t) + \frac{1}{\gamma}(\theta^t - y^t)) - \theta_\gamma^*(x^t, y^t) + \eta_t(\nabla_2 g(x^t, \theta_\gamma^*(x^t, y^t)) + \frac{1}{\gamma}(\theta_\gamma^*(x^t, y^t) - y^t))\|$$

$$\leq \sqrt{1 - \eta_t(1/\gamma - \kappa_{g_2})}\|\theta^t - \theta_\gamma^*(x^t, y^t)\|.$$

$\square$

**Lemma A.8.** *Under Assumptions 5.1, 5.2 and 5.3, suppose $\gamma \in (0, \frac{1}{2\kappa_{g_2}})$ the sequence of $\{x^t, y^t, \theta^t\}$ generated by Algorithm 1. We define a function $\Psi_{c_t}(x, y) = \frac{1}{c_t} f(x, y) + g(x, y) - v_\gamma(x, y)$ with $c_{t+1} \geq c_t > 0$ for all $t \geq 0$, which satisfies*

*(i) when $\rho \in [0, 1)$,*

$$\Psi_{c_t}(x^{t+1}, y^{t+1})$$

$$\leq \Psi_{c_t}(x^t, y^t) + \left(\frac{\alpha_t}{\|d_x^t\|}\left(L_{gx,0} + L_{gx,1}M^\rho\right) + \frac{2\beta_t}{\gamma^2\|d_y^t\|}\right)\|\theta^{t+1} - \theta_\gamma^*(x^t, y^t)\|^2$$

$$- \frac{\alpha_t}{4}\left\|\frac{1}{c_t}\nabla_1 f(x^t, y^t) + \nabla_1 g(x^t, y^t)\right\| + \frac{3}{4}\alpha_t\|\nabla_1 g(x^t, \theta^{t+1})\| + \frac{\alpha_t^2}{2}\left(K_0 + \kappa + K_1 + 2K_2 + \frac{4\beta_t}{\gamma^2\|d_y^t\|}\right)$$

$$- \frac{\beta_t}{4}\left\|\frac{1}{c_t}\nabla_2 f(x^{t+1}, y^t) + \nabla_2 g(x^{t+1}, y^t)\right\| + \frac{\beta_t^2}{2}(K_3 + \kappa + K_4 + 2K_5),$$

*(ii) when $\rho = 1$,*

$$\Psi_{c_t}(x^{t+1}, y^{t+1})$$

$$\leq \Psi_{c_t}(x^t, y^t) + \left(\frac{\alpha_t}{\|d_x^t\|}(L_{gx,0} + L_{gx,1}M^\rho + \frac{2\beta_t}{\gamma^2\|d_y^t\|}\right)\|\theta^{t+1} - \theta_\gamma^*(x^t, y^t)\|^2$$

$$- \frac{\alpha_t}{4}\left(\left\|\frac{1}{c_t}\nabla_1 f(x^t, y^t)\right\| + \|\nabla_1 g(x^t, y^t)\|\right) + \frac{3}{4}\alpha_t\|\nabla_1 g(x^t, \theta^{t+1})\| + \frac{\alpha_t^2}{2}\left(L_0 + \kappa + L_1 + \frac{4\beta_t}{\gamma^2\|d_y^t\|}\right)$$

$$- \frac{\beta_t}{4}\left(\|\frac{1}{c_t}\nabla_2 f(x^{t+1}, y^t)\| + \|\nabla_2 g(x^{t+1}, y^t)\|\right) + \frac{\beta_t^2}{2}(L_2 + \kappa + L_3).$$

*Proof.* By the definition of function $\Psi_{c_t}(x, y)$, we have

(i) when $\rho \in [0, 1)$,

$$\Psi_{c_t}(x^{t+1}, y^t) \leq \frac{1}{c_t}f(x^t, y^t) + g(x^t, y^t) - v_\gamma(x^{t+1}, y^t) + \left\langle\frac{1}{c_t}\nabla_1 f(x^t, y^t) + \nabla_1 g(x^t, y^t), x^{t+1} - x^t\right\rangle$$

$$+ \frac{1}{2}\left(K_0 + K_1(\|\frac{1}{c_t}\nabla_1 f(x^t, y^t)\|^\rho + \|\nabla_1 g(x^t, y^t)\|^\rho) + 2K_2\left\|x^{t+1} - x^t\right\|^{\frac{\rho}{1-\rho}}\right)\|x^{t+1} - x^t\|^2$$

$$\leq \Psi_{c_t}(x^t, y^t) + \langle\nabla_1\Psi_{c_t}(x^t, y^t), x^{t+1} - x^t\rangle + \frac{1}{2}\left(K_0 + K_1(\|\frac{1}{c_t}\nabla_1 f(x^t, y^t)\|^\rho + \|\nabla_1 g(x^t, y^t)\|^\rho)\right.$$

$$\left. + 2K_2\left\|x^{t+1} - x^t\right\|^{\frac{\rho}{1-\rho}} + \kappa\right)\|x^{t+1} - x^t\|^2$$

$$= \Psi_{c_t}(x^t, y^t) + \langle\nabla_1\Psi_{c_t}(x^t, y^t) - d_x^t, x^{t+1} - x^t\rangle + \langle d_x^t, x^{t+1} - x^t\rangle$$

$$+ \frac{1}{2}\left(K_0 + K_1(\|\frac{1}{c_t}\nabla_1 f(x^t, y^t)\|^\rho + \|\nabla_1 g(x^t, y^t)\|^\rho) + 2K_2\left\|x^{t+1} - x^t\right\|^{\frac{\rho}{1-\rho}} + \kappa\right)\|x^{t+1} - x^t\|^2,$$

$$(37)$$

where the first inequality holds by Proposition A.4, and the second inequality holds by the Lemma A.1.

Considering the update rule for the variable $x$ in Algorithm 1, it follows that

$$\langle d_x^t, x^{t+1} - x^t \rangle = -\alpha_t \|d_x^t\|. \tag{38}$$

And we have

$$
\begin{aligned}
&\langle \nabla_1 \Psi_{c_t}(x^t, y^t) - d_x^t, x^{t+1} - x^t \rangle \\
&= \langle \nabla_1 g(x^t, \theta^{t+1}) - \nabla_1 g(x^t, \theta_\gamma^*(x^t, y^t)), x^{t+1} - x^t \rangle \\
&\leq \|\nabla_1 g(x^t, \theta^{t+1}) - \nabla_1 g(x^t, \theta_\gamma^*(x^t, y^t))\| \|x^{t+1} - x^t\| \\
&\leq \left( L_{gx,0} + L_{gx,1} \max_{\mu \in [0,1]} \|\nabla_1 g(x^t, \mu\theta^{t+1} + (1-\mu)\theta_\gamma^*(x^t, y^t))\|^\rho \right) \|\theta_\gamma^*(x^t, y^t) - \theta^{t+1}\| \|x^{t+1} - x^t\| \\
&\leq \frac{\|d_x^t\|}{4\alpha_t} \|x^{t+1} - x^t\|^2 + \frac{\alpha_t}{\|d_x^t\|} \left( L_{gx,0} + L_{gx,1} \max_{\mu \in [0,1]} \|\nabla_1 g(x^t, \mu\theta^{t+1} + (1-\mu)\theta_\gamma^*(x^t, y^t))\|^\rho \right) \|\theta_\gamma^*(x^t, y^t) - \theta^{t+1}\|^2,
\end{aligned}
\tag{39}
$$

where the last inequality follows from Young's inequality.

For the sake of simplicity in presentation, let $M = \max_{\mu \in [0,1]} \|\nabla_1 g(x^t, \mu\theta^{t+1} + (1-\mu)\theta_\gamma^*(x^t, y^t))\|$, by combining (38) with (39), we have

$$
\begin{aligned}
&\Psi_{c_t}(x^{t+1}, y^t) \\
&\leq \Psi_{c_t}(x^t, y^t) + \frac{\alpha_t}{\|d_x^t\|} \left( L_{gx,0} + L_{gx,1} M^\rho \right) \|\theta^{t+1} - \theta_\gamma^*(x^t, y^t)\|^2 + \frac{\|d_x^t\|}{4\alpha_t} \|x^{t+1} - x^t\|^2 - \frac{\|d_x^t\|}{\alpha_t} \|x^{t+1} - x^t\|^2 \\
&\quad + \frac{1}{2} \left( K_0 + K_1(\|\frac{1}{c_t}\nabla_1 f(x^t, y^t)\|^\rho + \|\nabla_1 g(x^t, y^t)\|^\rho) + 2K_2 \|x^{t+1} - x^t\|^{\frac{\rho}{1-\rho}} + \kappa \right) \|x^{t+1} - x^t\|^2 \\
&\leq \Psi_{c_t}(x^t, y^t) + \frac{\alpha_t}{\|d_x^t\|} \left( L_{gx,0} + L_{gx,1} M^\rho \right) \|\theta^{t+1} - \theta_\gamma^*(x^t, y^t)\|^2 - \frac{3}{4}\alpha_t \|d_x^t\| \\
&\quad + \frac{\alpha_t}{6} \left[ 3(K_0 + \kappa)\alpha_t + 3(\|\frac{1}{c_t}\nabla_1 f(x^t, y^t)\| + \|\nabla_1 g(x^t, y^t)\|) + 6K_1\alpha_t + 6K_2\alpha_t \right] \\
&\leq \Psi_{c_t}(x^t, y^t) + \frac{\alpha_t}{\|d_x^t\|} \left( L_{gx,0} + L_{gx,1} M^\rho \right) \|\theta^{t+1} - \theta_\gamma^*(x^t, y^t)\|^2 - \frac{\alpha_t}{4}(\|\frac{1}{c_t}\nabla_1 f(x^t, y^t)\| + \|\nabla_1 g(x^t, y^t)\|) \\
&\quad + \frac{3\alpha_t}{4} \|\nabla_1 g(x^t, \theta^{t+1})\| + \frac{\alpha_t^2}{2}(K_0 + \kappa + K_1 + 2K_2),
\end{aligned}
\tag{40}
$$

where the second inequality follows from the above Lemma A.3.

Similarly, we have

$$
\begin{aligned}
&\Psi_{c_t}(x^{t+1}, y^{t+1}) \\
&\leq \Psi_{c_t}(x^{t+1}, y^t) + \langle \nabla_2 \Psi_{c_t}(x^{t+1}, y^t), y^{t+1} - y^t \rangle + \frac{1}{2} \left( K_3 + K_4(\|\frac{1}{c_t}\nabla_2 f(x^{t+1}, y^t)\|^\rho + \|\nabla_2 g(x^{t+1}, y^t)\|^\rho) \right. \\
&\quad \left. + 2K_5 \|y^{t+1} - y^t\|^{\frac{\rho}{1-\rho}} + \kappa \right) \|y^{t+1} - y^t\|^2 \\
&\leq \Psi_{c_t}(x^{t+1}, y^t) + \frac{2\beta_t}{\gamma^2 \|d_y^t\|} \left( \|\theta^{t+1} - \theta_\gamma^*(x^t, y^t)\|^2 + \|x^{t+1} - x^t\|^2 \right) - \frac{3\|d_y^t\|}{4\beta_t} \|y^{t+1} - y^t\|^2 + \frac{1}{2} \left( K_3 \right. \\
&\quad \left. + K_4(\|\frac{1}{c_t}\nabla_2 f(x^{t+1}, y^t)\|^\rho + \|\nabla_2 g(x^{t+1}, y^t)\|^\rho) + 2K_5 \|y^{t+1} - y^t\|^{\frac{\rho}{1-\rho}} + \kappa \right) \|y^{t+1} - y^t\|^2 \\
&\leq \Psi_{c_t}(x^{t+1}, y^t) + \frac{2\beta_t}{\gamma^2 \|d_y^t\|} \left( \|\theta^{t+1} - \theta_\gamma^*(x^t, y^t)\|^2 + \|x^{t+1} - x^t\|^2 \right) - \frac{\beta_t}{4} \left( \left\| \frac{1}{c_t}\nabla_2 f(x^{t+1}, y^t) \right\| \right. \\
&\quad \left. + \|\nabla_2 g(x^{t+1}, y^t)\| \right) + \frac{\beta_t^2}{2}(K_3 + \kappa + K_4 + 2K_5).
\end{aligned}
\tag{41}
$$

By combining the above inequalities (41) with (40), we can obtain

$$
\begin{aligned}
\Psi_{c_t}&(x^{t+1}, y^{t+1}) \\
&\leq \Psi_{c_t}(x^t, y^t) + \left( \frac{\alpha_t}{\|d_x^t\|}(L_{gx,0} + L_{gx,1}M^\rho) + \frac{2\beta_t}{\gamma^2 \|d_y^t\|} \right) \|\theta^{t+1} - \theta_\gamma^*(x^t, y^t)\|^2 \\
&\quad - \frac{\alpha_t}{4}(\|\frac{1}{c_t}\nabla_1 f(x^t, y^t)\| + \|\nabla_1 g(x^t, y^t)\|) + \frac{3}{4}\alpha_t \|\nabla_1 g(x^t, \theta^{t+1})\| + \frac{\alpha_t^2}{2}\left( K_0 + \kappa + K_1 + 2K_2 + \frac{4\beta_t}{\gamma^2 \|d_y^t\|} \right) \\
&\quad - \frac{\beta_t}{4}(\|\frac{1}{c_t}\nabla_2 f(x^{t+1}, y^t)\| + \|\nabla_2 g(x^{t+1}, y^t)\|) + \frac{\beta_t^2}{2}(K_3 + \kappa + K_4 + 2K_5).
\end{aligned}
$$

(ii) When $\rho = 1$, similarly, we have

$$
\begin{aligned}
\Psi_{c_t}&(x^{t+1}, y^{t+1}) - \Psi_{c_t}(x^t, y^t) \\
&\leq \left( \frac{\alpha_t}{\|d_x^t\|}(L_{gx,0} + L_{gx,1}M) + \frac{2\beta_t}{\gamma^2 \|d_y^t\|} \right) \|\theta^{t+1} - \theta_\gamma^*(x^t, y^t)\|^2 - \frac{\alpha_t}{4}\left( \left\|\frac{1}{c_t}\nabla_1 f(x^t, y^t)\right\| + \|\nabla_1 g(x^t, y^t)\| \right) \\
&\quad + \frac{3}{4}\alpha_t \|\nabla_1 g(x^t, \theta^{t+1})\| + \frac{\alpha_t^2}{2}\left( L_0 + \kappa + L_1 + \frac{4\beta_t}{\gamma^2 \|d_y^t\|} \right) - \frac{\beta_t}{4}\left( \left\|\frac{1}{c_t}\nabla_2 f(x^{t+1}, y^t)\right\| + \|\nabla_2 g(x^{t+1}, y^t)\| \right) \\
&\quad + \frac{\beta_t^2}{2}(L_2 + \kappa + L_3).
\end{aligned}
\tag{42}
$$

The proof have completed. □

**Lemma A.9.** *Under Assumptions 5.2 and 5.3, suppose $\gamma \in (0, \frac{1}{2\kappa_{g_2}})$, $c_{t+1} \geq c_t > 0$ and $\eta_t$ satisfies $\eta_t >$*

$$
\frac{1 + \sqrt{1 + 4(L_{gx,0} + L_{gx,1}M^\rho + \frac{2}{\gamma^2}))\left( (L_{gx,0} + L_{gx,1}M^\rho)\frac{\alpha_t}{\|d_x^t\|} + \frac{\beta_t}{\gamma^2 \|d_y^t\|} \right)}}{2\kappa_{g_2}(L_{gx,0} + L_{gx,1}M^\rho + \frac{2}{\gamma^2})}, \text{ we have}
$$

$$
\begin{aligned}
\Omega_{t+1}& - \Omega_t \\
&\leq -\frac{\alpha_t}{8}(\|\frac{1}{c_t}\nabla_1 f(x^t, y^t)\| + \|\nabla_1 g(x^t, y^t)\|) - \frac{\beta_t}{8}(\|\frac{1}{c_t}\nabla_2 f(x^t, y^t)\| + \|\nabla_2 g(x^t, y^t)\|) - \eta_t \kappa_{g_2} \|\theta^{t+1} - \theta_\gamma^*(x^t, y^t)\|.
\end{aligned}
\tag{43}
$$

*where $\Omega_{t+1} = \Psi_{c_{t+1}}(x^{t+1}, y^{t+1}) + \left( L_{gx,0} + L_{gx,1}M^\rho + \frac{2}{\gamma^2} \right)\left\|\theta^{t+1} - \theta_\gamma^*(x^{t+1}, y^{t+1})\right\|^2$.*

*Proof.* We first define a useful Lyapunov function,

$$
\Omega_t = \Psi_{c_t}(x^t, y^t) + \left( L_{gx,0} + L_{gx,1}M^\rho + \frac{2}{\gamma^2} \right)\left\|\theta^t - \theta_\gamma^*(x^t, y^t)\right\|^2,
\tag{44}
$$

where $\Psi_{c_t}(x, y) = \frac{1}{c_t}f(x, y) + g(x, y) - v_\gamma(x, y)$ and $c_{t+1} \geq c_t > 0$ for all $t \geq 0$.

By the above definition of $\Omega_t$, when $\rho \in [0, 1)$, we have

$$
\begin{aligned}
&\Omega_{t+1} - \Omega_t \\
=&\Psi_{c_{t+1}}(x^{t+1}, y^{t+1}) - \Psi_{c_t}(x^t, y^t) + \left(L_{gx,0} + L_{gx,1}M^\rho + \frac{2}{\gamma^2}\right)\left\|\theta^{t+1} - \theta_\gamma^*(x^{t+1}, y^{t+1})\right\|^2 \\
&- \left(L_{gx,0} + L_{gx,1}M^\rho + \frac{2}{\gamma^2}\right)\left\|\theta^t - \theta_\gamma^*(x^t, y^t)\right\|^2 \\
\leq&\Psi_{c_t}(x^{t+1}, y^{t+1}) - \Psi_{c_t}(x^t, y^t) + \left(L_{gx,0} + L_{gx,1}M^\rho + \frac{2}{\gamma^2}\right)\left\|\theta^{t+1} - \theta_\gamma^*(x^{t+1}, y^{t+1})\right\|^2 \\
&- \left(L_{gx,0} + L_{gx,1}M^\rho + \frac{2}{\gamma^2}\right)\left\|\theta^t - \theta_\gamma^*(x^t, y^t)\right\|^2 \\
\leq&\left(\frac{\alpha_t}{\|d_x^t\|}(L_{gx,0} + L_{gx,1}M^\rho) + \frac{2\beta_t}{\gamma^2\|d_y^t\|}\right)\|\theta^{t+1} - \theta_\gamma^*(x^t, y^t)\|^2 \\
&- \frac{\alpha_t}{4}(\|\frac{1}{c_t}\nabla_1 f(x^t, y^t)\| + \|\nabla_1 g(x^t, y^t)\| + \frac{3}{4}\alpha_t\|\nabla_1 g(x^t, \theta^{t+1})\| + \frac{\alpha_t^2}{2}\left(K_0 + \kappa + K_1 + 2K_2 + \frac{4\beta_t}{\gamma^2\|d_y^t\|}\right) \\
&- \frac{\beta_t}{4}(\|\frac{1}{c_t}\nabla_2 f(x^{t+1}, y^t)\| + \|\nabla_2 g(x^{t+1}, y^t)\|) + \frac{\beta_t^2}{2}(K_3 + \kappa + K_4 + 2K_5) \\
&+ \left(L_{gx,0} + L_{gx,1}M^\rho + \frac{2}{\gamma^2}\right)\left\|\theta^{t+1} - \theta_\gamma^*(x^{t+1}, y^{t+1})\right\|^2 - \left(L_{gx,0} + L_{gx,1}M^\rho + \frac{2}{\gamma^2}\right)\left\|\theta^t - \theta_\gamma^*(x^t, y^t)\right\|^2 \\
\leq&-\frac{\alpha_t}{4}\left(\|\frac{1}{c_t}\nabla_1 f(x^t, y^t)\| + \|\nabla_1 g(x^t, y^t)\| - 3\|\nabla_1 g(x^t, \theta^{t+1})\| - 2\alpha_t\left(K_0 + \kappa + K_1 + 2K_2 + \frac{4\beta_t}{\gamma^2\|d_y^t\|}\right)\right) \\
&- \frac{\beta_t}{4}\left(\|\frac{1}{c_t}\nabla_2 f(x^{t+1}, y^t)\| + \|\nabla_2 g(x^{t+1}, y^t)\| - 2\beta_t(K_3 + \kappa + K_4 + 2K_5)\right) \\
&+ \left(L_{gx,0} + L_{gx,1}M^\rho + \frac{2}{\gamma^2}\right)\left\|\theta^{t+1} - \theta_\gamma^*(x^{t+1}, y^{t+1})\right\|^2 - \left(L_{gx,0} + L_{gx,1}M^\rho + \frac{2}{\gamma^2}\right)\left\|\theta^t - \theta_\gamma^*(x^t, y^t)\right\|^2 \\
&+ \left(\frac{\alpha_t}{\|d_x^t\|}\left(L_{gx,0} + L_{gx,1}M^\rho\right) + \frac{2\beta_t}{\gamma^2\|d_y^t\|}\right)\|\theta^{t+1} - \theta_\gamma^*(x^t, y^t)\|^2,
\end{aligned}
\tag{45}
$$

where the second inequality follows from Lemma A.7 and the last inequality by rearrangement.

Next, we demonstrate that

$$
\begin{aligned}
&\|\theta^{t+1} - \theta_\gamma^*(x^{t+1}, y^{t+1})\|^2 - \|\theta^t - \theta_\gamma^*(x^t, y^t)\|^2 + \frac{\alpha_t}{\|d_x^t\|}\|\theta^{t+1} - \theta_\gamma^*(x^t, y^t)\|^2 \\
\leq&\left(1 + \nu_t + \frac{\alpha_t}{\|d_x^t\|}\right)\|\theta^{t+1} - \theta_\gamma^*(x^t, y^t)\|^2 - \|\theta^t - \theta_\gamma^*(x^t, y^t)\|^2 + \left(1 + \frac{1}{\nu_t}\right)\|\theta_\gamma^*(x^{t+1}, y^{t+1}) - \theta_\gamma^*(x^t, y^t)\|^2 \\
\leq&\left(1 + \nu_t + \frac{\alpha_t}{\|d_x^t\|}\right)\sigma_t^2\|\theta^t - \theta_\gamma^*(x^t, y^t)\|^2 - \|\theta^t - \theta_\gamma^*(x^t, y^t)\|^2 + \left(1 + \frac{1}{\nu_t}\right)L_\theta^2\|(x^{t+1}, y^{t+1}) - (x^t, y^t)\|^2,
\end{aligned}
\tag{46}
$$

where the first inequality follows from Young's inequality with parameter $\nu_t$ and the last inequality follows from Lemma A.7.

Given $\nu_t = \eta_t \kappa_{g_2}$, we have

$$
\begin{aligned}
&\|\theta^{t+1} - \theta_\gamma^*(x^{t+1}, y^{t+1})\|^2 - \|\theta^t - \theta_\gamma^*(x^t, y^t)\|^2 + \frac{\alpha_t}{\|d_x^t\|}\|\theta^{t+1} - \theta_\gamma^*(x^t, y^t)\|^2 \\
\leq&-\left(\eta_t^2 \kappa_{g_2}^2 - \frac{\alpha_t}{\|d_x^t\|}\right)\|\theta^t - \theta_\gamma^*(x^t, y^t)\|^2 + \left(1 + \frac{1}{\eta_t \kappa_{g_2}}\right)L_\theta^2\|(x^{t+1}, y^{t+1}) - (x^t, y^t)\|^2.
\end{aligned}
\tag{47}
$$

Similarly, we have

$$
\|\theta^{t+1} - \theta_\gamma^*(x^{t+1}, y^{t+1})\|^2 - \|\theta^t - \theta_\gamma^*(x^t, y^t)\|^2 + \frac{2\beta_t}{\|d_y^t\|}\|\theta^{t+1} - \theta_\gamma^*(x^t, y^t)\|^2
$$

$$
\leq \left(1 + \nu_t + \frac{2\beta_t}{\|d_y^t\|}\right)\sigma_t^2\|\theta^t - \theta_\gamma^*(x^t, y^t)\|^2 - \|\theta^t - \theta_\gamma^*(x^t, y^t)\|^2 + \left(1 + \frac{1}{\nu_t}\right)L_\theta^2\|(x^{t+1}, y^{t+1}) - (x^t, y^t)\|^2
$$

$$
\leq -\left(\eta_t^2\kappa_{g_2}^2 - \frac{2\beta_t}{\|d_y^t\|}\right)\|\theta^t - \theta_\gamma^*(x^t, y^t)\|^2 + \left(1 + \frac{1}{\eta_t\kappa_{g_2}}\right)L_\theta^2\|(x^{t+1}, y^{t+1}) - (x^t, y^t)\|^2. \tag{48}
$$

Putting the above inequalities (45), (47) into (48), we have

$$
\Omega_{t+1} - \Omega_t
$$

$$
\leq -\frac{\alpha_t}{4}\left(\|\frac{1}{c_t}\nabla_1 f(x^t, y^t)\| + \|\nabla_1 g(x^t, y^t)\| - 3\|\nabla_1 g(x^t, \theta^{t+1})\| - 2\alpha_t\left(K_0 + \kappa + K_1 + 2K_2 + \frac{4\beta_t}{\gamma^2\|d_y^t\|}\right)\right.
$$

$$
- 4\alpha_t\left(1 + \frac{1}{\eta_t\kappa_{g_2}}\right)L_\theta^2\left(L_{gx,0} + L_{gx,1}M^\rho + \frac{2}{\gamma^2}\right)\right) - \frac{\beta_t}{4}\left(\|\frac{1}{c_t}\nabla_2 f(x^{t+1}, y^t)\| + \|\nabla_2 g(x^{t+1}, y^t)\|\right.
$$

$$
- 2\beta_t(K_3 + \kappa + K_4 + 2K_5) - 4\beta_t\left(1 + \frac{1}{\eta_t\kappa_{g_2}}\right)L_\theta^2(L_{gx,0} + L_{gx,1}M^\rho + \frac{2}{\gamma^2})\right) - \left((L_{gx,0} + L_{gx,1}M^\rho + \frac{2}{\gamma})\eta_t^2\kappa_{g_2}^2\right.
$$

$$
- (L_{gx,0} + L_{gx,1}M^\rho)\frac{\alpha_t}{\|d_x^t\|} - \frac{\beta_t}{\gamma^2\|d_y^t\|}\right)\|\theta^t - \theta_\gamma^*(x^t, y^t)\|^2.
$$

Given $0 < \alpha_t \leq \frac{\|\frac{1}{c_t}\nabla_1 f(x^t, y^t)\| + \|\nabla_1 g(x^t, y^t)\|}{4(K_0 + \kappa + K_1 + 2K_2) + 8(1 + \frac{1}{\eta_t\kappa_{g_2}})L_\theta^2(L_{gx,0} + L_{gx,1}M^\rho + \frac{2}{\gamma^2}) + 6\|\nabla_1 g(x^t, \theta^{t+1})\|}$,

$0 < \beta_t \leq \frac{\|\frac{1}{c_t}\nabla_2 f(x^t, y^t)\| + \|\nabla_2 g(x^t, y^t)\|}{4(K_3 + \kappa + K_4 + 2K_5) + 8(1 + \frac{1}{\eta_t\kappa_{g_2}})L_\theta^2(L_{gx,0} + L_{gx,1}M^\rho + \frac{2}{\gamma^2})}$, and $\eta_t >$

$\frac{1 + \sqrt{1 + 4(L_{gx,0} + L_{gx,1}M^\rho + \frac{2}{\gamma^2}))\left((L_{gx,0} + L_{gx,1}M^\rho)\frac{\alpha_t}{\|d_x^t\|} + \frac{\beta_t}{\gamma^2\|d_y^t\|}\right)}}{2\kappa_{g_2}(L_{gx,0} + L_{gx,1}M^\rho + \frac{2}{\gamma^2})}$, we can obtain

$$
\Omega_{t+1} - \Omega_t
$$

$$
\leq -\frac{\alpha_t}{8}(\|\frac{1}{c_t}\nabla_1 f(x^t, y^t)\| + \|\nabla_1 g(x^t, y^t)\|) - \frac{\beta_t}{8}(\|\frac{1}{c_t}\nabla_2 f(x^t, y^t)\| + \|\nabla_2 g(x^t, y^t)\|) - \eta_t\kappa_{g_2}\|\theta^{t+1} - \theta_\gamma^*(x^t, y^t)\|^2. \tag{49}
$$

When $\rho = 1$, according to Lemma A.2, we have

$$
\Omega_{t+1} - \Omega_t
$$

$$
\leq \left(\frac{\alpha_t}{\|d_x^t\|}(L_{gx,0} + L_{gx,1}M) + \frac{2\beta_t}{\gamma^2\|d_y^t\|}\right)\|\theta^{t+1} - \theta_\gamma^*(x^t, y^t)\|^2 - \frac{\alpha_t}{4}\left(\left\|\frac{1}{c_t}\nabla_1 f(x^t, y^t)\right\| + \|\nabla_1 g(x^t, y^t)\|\right)
$$

$$
+ \frac{3}{4}\alpha_t\|\nabla_1 g(x^t, \theta^{t+1})\| + \frac{\alpha_t^2}{2}\left(L_0 + \kappa + L_1 + \frac{4\beta_t}{\gamma^2\|d_y^t\|}\right) - \frac{\beta_t}{4}\left(\left\|\frac{1}{c_t}\nabla_2 f(x^{t+1}, y^t)\right\| + \|\nabla_2 g(x^{t+1}, y^t)\|\right)
$$

$$
+ \frac{\beta_t^2}{2}(L_2 + \kappa + L_3) + \left(L_{gx,0} + L_{gx,1}M + \frac{2}{\gamma^2}\right)\|\theta^t - \theta_\gamma^*(x^{t+1}, y^{t+1})\|^2 - \left(L_{gx,0} + L_{gx,1}M + \frac{2}{\gamma^2}\right)\|\theta^t - \theta_\gamma^*(x^t, y^t)\|^2
$$

$$
\leq -\frac{\alpha_t}{4}\left(\left\|\frac{1}{c_t}\nabla_1 f(x^t, y^t)\right\| + \|\nabla_1 g(x^t, y^t)\| - 3\|\nabla_1 g(x^t, \theta^{t+1})\| - 2\alpha_t(L_0 + \kappa + L_1) - 4\alpha_t(1 + \frac{1}{\eta_t\kappa_{g_2}})L_\theta^2(L_{gx,0}\right.
$$

$$
+ L_{gx,1}M + \frac{2}{\gamma^2})\right) - \frac{\beta_t}{4}\left(\left\|\frac{1}{c_t}\nabla_2 f(x^{t+1}, y^t)\right\| + \|\nabla_2 g(x^{t+1}, y^t)\| - 2\beta_t(L_2 + L_3 + \kappa) - 4\beta_t(1 + \frac{1}{\eta_t\kappa_{g_2}})L_\theta^2(L_{gx,0}\right.
$$

$$
+ L_{gx,1}M + \frac{2}{\gamma^2})\right) - \left((L_{gx,0} + L_{gx,1}M + \frac{2}{\gamma})^2\eta_t^2\kappa_t^2 - (L_{gx,0} + L_{gx,1}M)\frac{\alpha_t}{\|d_x^t\|} - \frac{\beta_t}{\gamma^2\|d_y^t\|}\right)\|\theta^t - \theta_\gamma^*(x^t, y^t)\|^2,
$$

where the last inequality follows from Young's inequality with parameter $\eta_t\kappa_{g_2}$ and rearrangement.

Given $0 < \alpha_t \leq \dfrac{\|\frac{1}{c_t}\nabla_1 f(x^t,y^t)\|+\|\nabla_1 g(x^t,y^t)\|}{4(L_0+\kappa+L_1)+8(1+\frac{1}{\eta_t\kappa_{g_2}})L_\theta^2(L_{gx,0}+L_{gx,1}M^\rho+\frac{2}{\gamma^2})+6\|\nabla_1 g(x^t,\theta^{t+1})\|}$, $0 < \beta_t \leq$

$\dfrac{\|\frac{1}{c_t}\nabla_2 f(x^t,y^t)\|+\|\nabla_2 g(x^t,y^t)\|}{4(L_2+\kappa+L_3)+8(1+\frac{1}{\eta_t\kappa_{g_2}})L_\theta^2(L_{gx,0}+L_{gx,1}M^\rho+\frac{2}{\gamma^2})}$, and $\eta_t > \dfrac{1+\sqrt{1+4(L_{gx,0}+L_{gx,1}M^\rho+\frac{2}{\gamma^2}))\left((L_{gx,0}+L_{gx,1}M^\rho)\frac{\alpha_t}{\|d_x^t\|}+\frac{\beta_t}{\gamma^2\|d_y^t\|}\right)}}{2\kappa_{g_2}(L_{gx,0}+L_{gx,1}M^\rho+\frac{2}{\gamma^2})}$,

we can also obtain

$$\Omega_{t+1} - \Omega_t$$
$$\leq -\frac{\alpha_t}{8}(\|\frac{1}{c_t}\nabla_1 f(x^t,y^t)\| + \|\nabla_1 g(x^t,y^t)\|) - \frac{\beta_t}{8}(\|\frac{1}{c_t}\nabla_2 f(x^t,y^t)\| + \|\nabla_2 g(x^t,y^t)\|) - \eta_t\kappa_{g_2}\|\theta^{t+1} - \theta_\gamma^*(x^t,y^t)\|^2. \tag{50}$$

Thus the function $\Omega_t$ is descent. □

Based on the Lyapunov function, the non-asymptotic convergence rate can be shown as:

**Theorem A.10.** *(Restatement of Theorem 5.5) Under Assumptions 5.1, 5.2, 5.3 and 5.4, given $\gamma \in (0, \frac{1}{2\kappa_{g_2}})$, $c_t = \underline{c}(t + 1)^{1/4}$ with $\underline{c} > 0$, and when $\rho \in [0,1)$ let $0 < \alpha_t \leq$*
$\dfrac{\|\frac{1}{c_t}\nabla_1 f(x^t,y^t)\|+\|\nabla_1 g(x^t,y^t)\|}{4(K_0+\kappa+K_1+2K_2)+8(1+\frac{1}{\eta_t\kappa_{g_2}})L_\theta^2 C+6\|\nabla_1 g(x^t,\theta^{t+1})\|}$, $0 < \beta_t \leq \dfrac{\|\frac{1}{c_t}\nabla_2 f(x^t,y^t)\|+\|\nabla_2 g(x^t,y^t)\|}{4(K_3+\kappa+K_4+2K_5)+8(1+\frac{1}{\eta_t\kappa_{g_2}})L_\theta^2 C}$, *when $\rho = 1$*
*let $0 < \alpha_t \leq \dfrac{\|\frac{1}{c_t}\nabla_1 f(x^t,y^t)\|+\|\nabla_1 g(x^t,y^t)\|}{4(L_0+\kappa+L_1)+8(1+\frac{1}{\eta_t\kappa_{g_2}})L_\theta^2 C+6\|\nabla_1 g(x^t,\theta^{t+1})\|}$, $0 < \beta_t \leq \dfrac{\|\frac{1}{c_t}\nabla_2 f(x^t,y^t)\|+\|\nabla_2 g(x^t,y^t)\|}{4(L_2+\kappa+L_3)+8(1+\frac{1}{\eta_t\kappa_{g_2}})L_\theta^2 C}$, and*

$\eta_t \in \left(\dfrac{1+\sqrt{1+4C\left((L_{gx,0}+L_{gx,1}M)\frac{\alpha_t}{\|d_x^t\|}+\frac{\beta_t}{\gamma^2\|d_y^t\|}\right)}}{2\kappa_{g_2}C}, \dfrac{1/\gamma-\kappa_{g_2}}{(1/\gamma+L_{gy,0}+L_{gy,1}\max_{\mu\in(0,1)}\|\nabla_2 g(x,\theta_\gamma^*(x,y)_\mu)\|^\rho)^2}\right)$, *where*

$C = L_{gx,0} + L_{gx,1}M^\rho + \frac{2}{\gamma^2}$. *The the sequence of $\{x^t, y^t, \theta^t\}_{t=0}^T$ generated by Algorithm 1 satisfies*

$$\min_{0\leq t\leq T}\|\theta^t - \theta_\gamma^*(x^t,y^t)\| = O\left(\frac{1}{T^{1/2}}\right),$$

$$\min_{0\leq t\leq T} R_t(x^{t+1},y^{t+1}) = O\left(\frac{1}{T^{\frac{1}{4}}}\right),$$

$$g(x^T,y^T) - v_\gamma(x^T,y^T) = O\left(\frac{1}{T^{\frac{1}{4}}}\right).$$

*Proof.* By using the above Lemma A.9, telescoping the above inequality (43) over the range $t = 0, 1, \cdots, T-1$, given $\underline{\eta} = \min_{0\leq t\leq T-1}\eta_t$, we have

$$\sum_{t=0}^{T-1}\left(\frac{\alpha_t}{8}(\|\frac{1}{c_t}\nabla_1 f(x^t,y^t)\| + \|\nabla_1 g(x^t,y^t)\|) + \frac{\beta_t}{8}(\|\frac{1}{c_t}\nabla_2 f(x^t,y^t)\| + \|\nabla_2 g(x^t,y^t)\|) + \underline{\eta}\kappa_{g_2}\|\theta^t - \theta_\gamma^*(x^t,y^t)\|^2\right)$$
$$\leq \Omega_0 - \Omega_T$$
$$\leq \Omega_0 - \frac{1}{c_T}f^* \leq \Omega_0, \tag{51}$$

where the last second inequality holds by Assumption 5.1. We obtain

$$\sum_{t=0}^{\infty}\|\theta^t - \theta_\gamma^*(x^t,y^t)\|^2 < \infty,$$

and then

$$\min_{0\leq t\leq T}\|\theta^t - \theta_\gamma^*(x^t,y^t)\| = O\left(\frac{1}{T^{1/2}}\right).$$

According to the update rule of variables $(x, y)$ in Algorithm 1, we have

$$c_t \frac{d_x^t}{\|d_x^t\|} + \frac{c_t}{\alpha_t}(x^{t+1} - x^t) = 0, \tag{52}$$

$$c_t \frac{d_y^t}{\|d_y^t\|} + \frac{c_t}{\beta_t}(y^{t+1} - x^t) = 0. \tag{53}$$

Let

$$(e_x^t, e_y^t) = \nabla f(x^{t+1}, y^{t+1}) + c_t(\nabla g(x^{t+1}, y^{t+1}) - v_\gamma(x^{t+1}, y^{t+1})),$$

with

$$e_x^t := c_t \nabla_1 \Psi_{c_t}(x^{t+1}, y^{t+1}) - c_t d_x^t - \frac{c_t \|d_x^t\|}{\alpha_t}(x^{t+1} - x^t), \tag{54}$$

$$e_y^t := c_t \nabla_2 \Psi_{c_t}(x^{t+1}, y^{t+1}) - c_t d_y^t - \frac{c_t \|d_y^t\|}{\beta_t}(y^{t+1} - y^t). \tag{55}$$

Considering the term $\|e_x^t\|$, we have

$$\|e_x^t\| \leq c_t \|\nabla_1 \Psi_{c_t}(x^{t+1}, y^{t+1}) - \nabla_1 \Psi_{c_t}(x^t, y^t)\| + \|c_t \nabla_1 \Psi_{c_t}(x^t, y^t) - c_t d_x^t\| + \frac{c_t \|d_x^t\|}{\alpha_t} \|x^{t+1} - x^t\|$$

$$\overset{(i)}{\leq} L_{\Psi_1} \|(x^{t+1}, y^{t+1}) - (x^t, y^t)\| + c_t \|\nabla_1 \Psi_{c_t}(x^t, y^t) - d_x^t\| + \frac{c_t \|d_x^t\|}{\alpha_t} \|x^{t+1} - x^t\|$$

$$= c_t L_{\Psi_1} \|(x^{t+1}, y^{t+1}) - (x^t, y^t)\| + \frac{c_t \|d_x^t\|}{\alpha_t} \|x^{t+1} - x^t\| + \|\nabla_1 g(x^t, \theta^t) - \nabla_1 g(x^t, \theta_\gamma^*(x^t, y^t))\|$$

$$\overset{(ii)}{\leq} L_{\Psi_1} \|(x^{t+1}, y^{t+1}) - (x^t, y^t)\| + \frac{c_t \|d_x^t\|}{\alpha_t} \|x^{t+1} - x^t\| + (L_{gx,0} + L_{gx,1} M^\rho) \|\theta^t - \theta_\gamma^*(x^t, y^t)\|,$$

where the above inequality (i) holds by the Lemma A.6, and the above inequality (ii) is due to Assumption 5.3. Thus, we can obtain

$$\|e_x^t\| \leq c_t L_{\Psi_1} \|(x^{t+1}, y^{t+1}) - (x^t, y^t)\| + \frac{c_t \|d_x^t\|}{\alpha_t} \|x^{t+1} - x^t\| + c_t (L_{gx,0} + L_{gx,1} M^\rho) \|\theta^t - \theta_\gamma^*(x^t, y^t)\|.$$

Similarity, based on Lemma A.6, we can obtain

$$\|e_y^t\| \leq c_t L_{\Psi_2} \|(x^{t+1}, y^{t+1}) - (x^t, y^t)\| + \frac{c_t \|d_y^t\|}{\beta_t} \|y^{t+1} - y^t\| + \frac{c_t}{\gamma}(\|\theta^t - \theta_\gamma^*(x^t, y^t)\| + L_\theta \|x^{t+1} - x^t\|),$$

where $L_{\Psi_2} = \frac{1}{c_t}(L_{fy,0} + L_{fy,1} \max_{\mu \in [0,1]} \|\nabla_2 f(x_\mu, y_\mu)\|^\rho) + L_{gy,0} + L_{gy,1} \max_{\mu \in [0,1]} \|\nabla_2 g(x_\mu, y_\mu)\|^\rho + \frac{1 + L_\theta}{\gamma}$.

Since $R_t(x, y) = \mathrm{dist}(0, \nabla f(x, y) + c_t(\nabla g(x, y) - \nabla v_\gamma(x, y)))$, we have

$$R_t(x^{t+1}, y^{t+1}) \leq c_t L_\Psi \|(x^{t+1}, y^{t+1}) - (x^t, y^t)\| + \left( \frac{c_t \|d_x^t\|}{\alpha_t} + \frac{c_t L_\theta}{\gamma} \right) \|x^{t+1} - x^t\| + \frac{c_t \|d_y^t\|}{\beta_t} \|y^{t+1} - y^t\|$$

$$+ c_t (L_{gx,0} + L_{gx,1} M^\rho + \frac{1}{\gamma}) \|\theta^t - \theta_\gamma^*(x^t, y^t)\|.$$

Given $C_R \leq \frac{\min\left\{ \|\nabla_1 f(x^t, y^t) + \nabla_1 g(x^t, y^t)\|, \|\nabla_2 f(x^t, y^t) + \nabla_2 g(x^t, y^t)\| \right\}}{\underline{\eta} \kappa_{g_2} \max\{\|d_x^t\|, \|d_y^t\|\}}$, we have

$$\frac{1}{c_t^2} R_t(x^{t+1}, y^{t+1})^2$$

$$\leq C_R \left( \frac{\alpha_t}{8}(\|\frac{1}{c_t} \nabla_1 f(x^t, y^t)\| + \|\nabla_1 g(x^t, y^t)\|) + \frac{\beta_t}{8}(\|\frac{1}{c_t} \nabla_2 f(x^t, y^t)\| + \|\nabla_2 g(x^t, y^t)\|) + \underline{\eta} \kappa_{g_2} \|\theta^t - \theta_\gamma^*(x^t, y^t)\|^2 \right).$$

By using the above inequality (51), then we have

$$\sum_{t=0}^{T-1} \frac{1}{c_t^2} R_t(x^{t+1}, y^{t+1})^2 < \infty. \tag{56}$$

Since $c_t = \underline{c}(t+1)^{\frac{1}{4}}$ with $\underline{c} > 0$, we have

$$\sum_{t=0}^{T-1} \frac{1}{c_t^2} = \frac{1}{\underline{c}^2} \sum_{t=0}^{T-1} \left(\frac{1}{t+1}\right)^{1/2} \geq \frac{(T+2)^{-1/2}}{1/2\underline{c}^2}. \tag{57}$$

Based on the above inequalities (56) and (57), we can obtain

$$\min_{0 \leq t \leq T} R_t(x^{t+1}, y^{t+1}) = O\left(\frac{1}{T^{1/4}}\right).$$

Since $\{x^t, y^t\}_{0 \leq t \leq T}$ is a sequence generated from Algorithm 1, we can find $m = \max_{0 \leq t \leq T} \Psi_{c_t}(x^t, y^t) = \frac{1}{c_t} f(x^t, y^t) + g(x^t, y^t) - v_\gamma(x^t, y^t)$, and $f(x^t, y^t) \geq f^*$ for any $t$. Then we have

$$c_t(g(x^t, y^t) - v_\gamma(x^t, y^t)) \leq m - f^*, \quad \forall t > 0.$$

Based on $c_t = \underline{c}(t+1)^{1/4}$, we can obtain

$$g(x^T, y^T) - v_\gamma(x^T, y^T) = O(\frac{1}{T^{1/4}}).$$

$\qquad\qquad\qquad\qquad\qquad\qquad\qquad\qquad\qquad\qquad\qquad\qquad\qquad\qquad\qquad\qquad\qquad\qquad\qquad\qquad\square$

## A.2. Stochastic Setting

**Lemma A.11.** *Given $\gamma \in (0, \frac{1}{2\kappa_{g_2}})$ and $\eta_t \in \left(0, \frac{1/\gamma - \kappa_{g_2}}{(1/\gamma + L_{gy,0} + L_{gy,1} \max_{\mu \in (0,1)} \|\nabla_2 g(x, \theta_\gamma^*(x,y)_\mu)\|^\rho)^2}\right)$, the sequence $\{x^t, y^t, \theta^t\}$ generated by Algorithm 2 satisfies*

$$\mathbb{E}[\|\theta^{t+1} - \theta^*(x^t, y^t)\|^2 | \mathcal{F}_t] \leq \sigma_t^2 \|\theta^t - \theta_\gamma^*(x^t, y^t)\|^2 + \frac{\eta_t^2 \delta_g^2}{B_t}. \tag{58}$$

*where $\sigma_t = \sqrt{1 - \eta_t(1/\gamma - \kappa_{g_2})}$ and $\theta_\gamma^*(x, y) = \arg\min_\theta \{\mathbb{E}_{\zeta \sim \mathcal{O}}[G(x, \theta; \zeta)] + \frac{1}{2\gamma} \|y - \theta\|^2\}$.*

*Proof.* By the update rule of $\theta$ in Algorithm 2 in line 3, we have

$$\|\theta^{t+1} - \theta_\gamma^*(x^t, y^t)\|^2$$

$$= \left\|\theta^t - \eta_t\left(\frac{1}{B_t}\sum_{i=1}^{B_t} \nabla_2 G(x^t, \theta^t; \hat{\zeta}_i^t) + \frac{1}{\gamma}(\theta^t - y^t)\right) - \theta_\gamma^*(x^t, y^t)\right\|^2$$

$$= \left\|\theta^t - \eta_t(\nabla_2 g(x^t, \theta^t) + \frac{1}{\gamma}(\theta^t - y^t)) - \theta_\gamma^*(x^t, y^t) - \eta_t\left(\frac{1}{B_t}\sum_{i=1}^{B_t} \nabla_2 G(x^t, \theta^t; \hat{\zeta}_i^t) + \frac{1}{\gamma}(\theta^t - y^t)\right)\right.$$

$$\left. - \nabla_2 g(x^t, \theta^t) + \frac{1}{\gamma}(\theta^t - y^t)\right)\right\|^2$$

$$= \|\theta^t - \eta_t(\nabla_2 g(x^t, \theta^t) + \frac{1}{\gamma}(\theta^t - y^t)) - \theta_\gamma^*(x^t, y^t)\|^2 + \eta_t^2 \left\|\frac{1}{B_t}\sum_{i=1}^{B_t} \nabla_2 G(x^t, \theta^t; \hat{\zeta}_i^t) - \nabla_2 g(x^t, \theta^t)\right\|^2$$

$$- 2\eta_t \langle \theta^t - \eta_t(\nabla_2 g(x^t, \theta^t) + \frac{1}{\gamma}(\theta^t - y^t)) - \theta_\gamma^*(x^t, y^t), \frac{1}{B_t}\sum_{i=1}^{B_t} \nabla_2 G(x^t, \theta^t; \hat{\zeta}_i^t) - \nabla_2 g(x^t, \theta^t)\rangle.$$

Then,

$$\mathbb{E}[\|\theta^{t+1} - \theta_\gamma^*(x^t, y^t)\|^2 | \mathcal{F}_t]$$

$$= \|\theta^t - \eta_t(\nabla_2 g(x^t, \theta^t) + \frac{1}{\gamma}(\theta^t - y^t)) - \theta_\gamma^*(x^t, y^t)\|^2 + \eta_t^2 \mathbb{E}\left[\left\|\frac{1}{B_t}\sum_{i=1}^{B_t} \nabla_2 G(x^t, \theta^t; \hat{\zeta}_i^t) - \nabla_2 g(x^t, \theta^t)\right\|^2 \Big| \mathcal{F}_t\right]$$

$$\leq \|\theta^t - \eta_t(\nabla_2 g(x^t, \theta^t) + \frac{1}{\gamma}(\theta^t - y^t)) - \theta_\gamma^*(x^t, y^t)\|^2 + \frac{\eta_t^2 \delta_g^2}{B_t}$$

$$\leq (1 - \eta_t(1/\gamma - \kappa_{g_2}))\|\theta^t - \theta_\gamma^*(x^t, y^t)\|^2 + \frac{\eta_t^2 \delta_g^2}{B_t},$$

where the first equality and first inequality follows from Assumption 5.7 and the last inequality follows from Lemma A.7. $\qquad\square$

**Lemma A.12.** *Under Assumptions 5.1, 5.2, 5.3,and 5.7, suppose $\gamma \in (0, \frac{1}{2\kappa_{g_2}})$ the sequence of $\{x^t, y^t, \theta^t\}$ generated by Algorithm 2. We define a function $\Phi_{c_t}(x, y) = \mathbb{E}[\frac{1}{c_t}F(x, y; \xi) + G(x, y; \zeta) - v_\gamma(x, y)]$ with $c_{t+1} \geq c_t > 0$ for all $t \geq 0$, and $\delta = \frac{1}{c_0}\delta_f^2 + 2\delta_g^2$, which satisfies*

*(i) when $\rho \in [0, 1)$,*

$$\mathbb{E}[\Phi_{c_t}(x^{t+1}, y^{t+1}) | \mathcal{F}_t]$$

$$\leq \Phi_{c_t}(x^t, y^t) + \mathbb{E}\left[\left(\frac{\alpha_t}{\|\tilde{d}_x^t\|}(L_{gx,0} + L_{gx,1}M^\rho) + \frac{2\beta_t}{\gamma^2\|\tilde{d}_y^t\|}\right)\|\theta^{t+1} - \theta_\gamma^*(x^t, y^t)\|^2 \Big| \mathcal{F}_t\right] +$$

$$- \frac{\alpha_t}{4}(\|\frac{1}{c_t}\nabla_1 f(x^t, y^t)\| + \|\nabla_1 g(x^t, y^t)\|) + \frac{3}{4}\alpha_t\|\nabla_1 g(x^t, \theta^{t+1})\| + \frac{\alpha_t^2}{2}\left(K_0 + \kappa + K_1 + 2K_2 + 2 + \frac{4\beta_t}{\gamma^2\|\tilde{d}_y^t\|}\right)$$

$$- \frac{\beta_t}{4}(\|\frac{1}{c_t}\nabla_2 f(x^{t+1}, y^t)\| + \|\nabla_2 g(x^{t+1}, y^t)\|) + \frac{\beta_t^2}{2}(K_3 + \kappa + K_4 + 2K_5 + 2) + \frac{\delta}{2B_t}(\alpha_t + \beta_t).$$

*(ii) when $\rho = 1$,*

$$\mathbb{E}[\Phi_{c_t}(x^{t+1}, y^{t+1}) | \mathcal{F}_t]$$

$$\leq \Phi_{c_t}(x^t, y^t) + \mathbb{E}\left[\left(\frac{\alpha_t}{\|\tilde{d}_x^t\|}(L_{gx,0} + L_{gx,1}M) + \frac{2\beta_t}{\gamma^2\|\tilde{d}_y^t\|}\right)\|\theta^{t+1} - \theta_\gamma^*(x^t, y^t)\|^2 \Big| \mathcal{F}_t\right] - \frac{\alpha_t}{4}\left(\left\|\frac{1}{c_t}\nabla_1 f(x^t, y^t)\right\|\right.$$

$$\left. + \|\nabla_1 g(x^t, y^t)\|\right) + \frac{3}{4}\alpha_t\|\nabla_1 g(x^t, \theta^{t+1})\| + \frac{\alpha_t^2}{2}\left(L_0 + \kappa + L_1 + 2 + \frac{4\beta_t}{\gamma^2\|\tilde{d}_y^t\|}\right) - \frac{\beta_t}{4}\left(\left\|\frac{1}{c_t}\nabla_2 f(x^{t+1}, y^t)\right\|\right.$$

$$\left. + \|\nabla_2 g(x^{t+1}, y^t)\|\right) + \frac{\beta_t^2}{2}(L_2 + \kappa + L_3 + 2) + \frac{\delta(\alpha_t + \beta_t)}{2B_t}.$$

*Proof.* By the definition of function $\Phi_{c_t}(x, y)$, we have

(i) when $\rho \in [0, 1)$,

$$\mathbb{E}[\Phi_{c_t}(x^{t+1}, y^t) | \mathcal{F}_t]$$

$$\leq \Phi_{c_t}(x^t, y^t) + \langle \nabla_1 \Phi_{c_t}(x^t, y^t) - d_x^t, x^{t+1} - x^t \rangle + \langle d_x^t, x^{t+1} - x^t \rangle$$

$$+ \frac{1}{2}\left(K_0 + K_1(\|\frac{1}{c_t}\nabla_1 f(x^t, y^t)\|^\rho + \|\nabla_1 g(x^t, y^t)\|^\rho) + 2K_2\|x^{t+1} - x^t\|^{\frac{\rho}{1-\rho}} + \kappa\right)\|x^{t+1} - x^t\|^2, \quad (59)$$

Considering the update rule for the variable $x$ in Algorithm 2, it follows that

$$\langle d_x^t, x^{t+1} - x^t \rangle \leq \langle d_x^t - \tilde{d}_x^t + \tilde{d}_x^t, x^{t+1} - x^t \rangle$$

$$\leq \frac{1}{4}\|d_x^t - \tilde{d}_x^t\|^2 + \|x^{t+1} - x^t\|^2 - \frac{\|\tilde{d}_x^t\|}{\alpha_t}\|x^{t+1} - x^t\|^2, \quad (60)$$

where the first inequality follows from Young's inequality and the update rule for variable $x$ in Algorithm 2. And similar as (39), we have

$$\langle \nabla_1 \Phi_{c_t}(x^t, y^t) - d_x^t, x^{t+1} - x^t \rangle \le \frac{\|\tilde{d}_x^t\|}{4\alpha_t} \|x^{t+1} - x^t\|^2 + \frac{\alpha_t}{\|\tilde{d}_x^t\|} \left( L_{gx,0} + L_{gx,1} M^\rho \right) \|\theta_\gamma^*(x^t, y^t) - \theta^{t+1}\|^2, \quad (61)$$

where $M = \max_{\mu \in [0,1]} \|\nabla_1 g(x^t, \mu\theta^{t+1} + (1-\mu)\theta_\gamma^*(x^t, y^t))\|$.

By combining (59) with (61), we have

$$\mathbb{E}[\Phi_{c_t}(x^{t+1}, y^t)|\mathcal{F}_t]$$
$$\le \Phi_{c_t}(x^t, y^t) + \mathbb{E}\left[ \frac{\alpha_t}{\|\tilde{d}_x^t\|} \left( L_{gx,0} + L_{gx,1} M^\rho \right) \|\theta^{t+1} - \theta_\gamma^*(x^t, y^t)\|^2 + \frac{1}{4}\|d_x^t - \tilde{d}_x^t\|^2 \Big| \mathcal{F}_t \right]$$
$$- \frac{3}{4}\alpha_t \mathbb{E}[\|\tilde{d}_x^t\| | \mathcal{F}_t] + \frac{\alpha_t}{6}\left[ 3(K_0 + \kappa)\alpha_t + 3(\|\frac{1}{c_t}\nabla_1 f(x^t, y^t)\| + \|\nabla_1 g(x^t, y^t)\|) + 6K_1\alpha_t + 6(K_2+1)\alpha_t \right]$$
$$\le \Phi_{c_t}(x^t, y^t) + \mathbb{E}\left[ \frac{\alpha_t}{\|\tilde{d}_x^t\|} \left( L_{gx,0} + L_{gx,1} M^\rho \right) \|\theta^{t+1} - \theta_\gamma^*(x^t, y^t)\|^2 + \frac{1}{4}\|d_x^t - \tilde{d}_x^t\|^2 \Big| \mathcal{F}_t \right]$$
$$- \frac{3}{4}\alpha_t \|d_x^t\| + \frac{\alpha_t}{6}\left[ 3(K_0 + \kappa)\alpha_t + 3(\|\frac{1}{c_t}\nabla_1 f(x^t, y^t)\| + \|\nabla_1 g(x^t, y^t)\|) + 6K_1\alpha_t + 6(K_2+1)\alpha_t \right]$$
$$\le \Phi_{c_t}(x^t, y^t) + \mathbb{E}\left[ \frac{\alpha_t}{\|\tilde{d}_x^t\|} \left( L_{gx,0} + L_{gx,1} M^\rho \right) \|\theta^{t+1} - \theta_\gamma^*(x^t, y^t)\|^2 + \frac{1}{4}\|d_x^t - \tilde{d}_x^t\|^2 \Big| \mathcal{F}_t \right]$$
$$- \frac{\alpha_t}{4}(\|\frac{1}{c_t}\nabla_1 f(x^t, y^t)\| + \|\nabla_1 g(x^t, y^t)\|) + \frac{3\alpha_t}{4}\|\nabla_1 g(x^t, \theta^{t+1})\| + \frac{\alpha_t^2}{2}(K_0 + \kappa + K_1 + 2K_2 + 2), \quad (62)$$

where the first inequality follows from the above Lemma A.3 and Assumption 5.7, the second inequality follows from Jensen's inequity $\mathbb{E}[\|\tilde{d}_x^t\| | \mathcal{F}_t] \ge \|d_x^t\|$ and the last inequality follows from $\|a - b\| \ge \|a\| - \|b\|$.

Similarly, we have

$$\mathbb{E}[\Phi_{c_t}(x^{t+1}, y^{t+1})|\mathcal{F}_t]$$
$$\le \Phi_{c_t}(x^{t+1}, y^t) + \langle \nabla_2 \Phi_{c_t}(x^{t+1}, y^t), y^{t+1} - y^t \rangle + \frac{1}{2}\bigg( K_3 + K_4(\|\frac{1}{c_t}\nabla_2 f(x^{t+1}, y^t)\|^\rho + \|\nabla_2 g(x^{t+1}, y^t)\|^\rho)$$
$$+ 2K_5\|y^{t+1} - y^t\|^{\frac{\rho}{1-\rho}} + \kappa \bigg) \|y^{t+1} - y^t\|^2$$
$$\le \Phi_{c_t}(x^{t+1}, y^t) + \mathbb{E}\left[ \frac{2\beta_t}{\gamma^2\|\tilde{d}_y^t\|} \left( \|\theta^{t+1} - \theta_\gamma^*(x^t, y^t)\|^2 + \|x^{t+1} - x^t\|^2 \right) \Big| \mathcal{F}_t \right] - \mathbb{E}\left[ \frac{3\|\tilde{d}_y^t\|}{4\beta_t}\|y^{t+1} - y^t\|^2 \Big| \mathcal{F}_t \right]$$
$$+ \frac{1}{2}\bigg( K_3 + K_4(\|\frac{1}{c_t}\nabla_2 f(x^{t+1}, y^t)\|^\rho + \|\nabla_2 g(x^{t+1}, y^t)\|^\rho) + 2K_5\|y^{t+1} - y^t\|^{\frac{\rho}{1-\rho}} + \kappa + 2 \bigg) \|y^{t+1} - y^t\|^2$$
$$+ \mathbb{E}\left[ \frac{1}{4}\|d_y^t - \tilde{d}_y^t\|^2 \Big| \mathcal{F}_t \right]$$
$$\le \Phi_{c_t}(x^{t+1}, y^t) + \mathbb{E}\left[ \frac{2\beta_t}{\gamma^2\|\tilde{d}_y^t\|} \left( \|\theta^{t+1} - \theta_\gamma^*(x^t, y^t)\|^2 + \|x^{t+1} - x^t\|^2 \right) + \frac{1}{4}\|d_y^t - \tilde{d}_y^t\|^2 \Big| \mathcal{F}_t \right]$$
$$- \frac{\beta_t}{4}(\|\frac{1}{c_t}\nabla_2 f(x^{t+1}, y^t)\| + \|\nabla_2 g(x^{t+1}, y^t)\|) + \frac{\beta_t^2}{2}(K_3 + \kappa + K_4 + 2K_5 + 2). \quad (63)$$

By combining the above inequalities (63) with (62), we can obtain

$$
\mathbb{E}[\Phi_{c_t}(x^{t+1}, y^{t+1})|\mathcal{F}_t]
$$

$$
\leq \Phi_{c_t}(x^t, y^t) + \mathbb{E}\left[\left(\frac{\alpha_t}{\|\tilde{d}_x^t\|}(L_{gx,0} + L_{gx,1}M^\rho) + \frac{2\beta_t}{\gamma^2\|\tilde{d}_y^t\|}\right)\|\theta^{t+1} - \theta_\gamma^*(x^t, y^t)\|^2 \Big| \mathcal{F}_t\right] +
$$

$$
- \frac{\alpha_t}{4}(\|\frac{1}{c_t}\nabla_1 f(x^t, y^t)\| + \|\nabla_1 g(x^t, y^t)\|) + \frac{3}{4}\alpha_t\|\nabla_1 g(x^t, \theta^{t+1})\| + \frac{\alpha_t^2}{2}\left(K_0 + \kappa + K_1 + 2K_2 + 2 + \frac{4\beta_t}{\gamma^2\|d_y^t\|}\right)
$$

$$
- \frac{\beta_t}{4}(\|\frac{1}{c_t}\nabla_2 f(x^{t+1}, y^t)\| + \|\nabla_2 g(x^{t+1}, y^t)\|) + \frac{\beta_t^2}{2}(K_3 + \kappa + K_4 + 2K_5 + 2) + \frac{\delta}{2B_t}(\alpha_t + \beta_t). \tag{64}
$$

(ii) When $\rho = 1$, similarly, we have

$$
\mathbb{E}[\Phi_{c_t}(x^{t+1}, y^{t+1})|\mathcal{F}_t]
$$

$$
\leq \Phi_{c_t}(x^t, y^t) + \mathbb{E}\left[\left(\frac{\alpha_t}{\|\tilde{d}_x^t\|}(L_{gx,0} + L_{gx,1}M) + \frac{2\beta_t}{\gamma^2\|\tilde{d}_y^t\|}\right)\|\theta^{t+1} - \theta_\gamma^*(x^t, y^t)\|^2 \Big| \mathcal{F}_t\right] - \frac{\alpha_t}{4}\left(\left\|\frac{1}{c_t}\nabla_1 f(x^t, y^t)\right\|\right.
$$

$$
+ \|\nabla_1 g(x^t, y^t)\|\Big) + \frac{3}{4}\alpha_t\|\nabla_1 g(x^t, \theta^{t+1})\| + \frac{\alpha_t^2}{2}\left(L_0 + \kappa + L_1 + 2 + \frac{4\beta_t}{\gamma^2\|\tilde{d}_y^t\|}\right) - \frac{\beta_t}{4}\left(\left\|\frac{1}{c_t}\nabla_2 f(x^{t+1}, y^t)\right\|\right.
$$

$$
+ \|\nabla_2 g(x^{t+1}, y^t)\|\Big) + \frac{\beta_t^2}{2}(L_2 + \kappa + L_3 + 2) + \frac{\delta}{2B_t}(\alpha_t + \beta_t). \tag{65}
$$

$\square$

Consider

$$
\Gamma_t = \left(L_{gx,0} + L_{gx,1}M^\rho + \frac{2}{\gamma^2}\right)\|\theta^{t+1} - \theta_\gamma^*(x^{t+1}, y^{t+1})\|^2 + \Phi_{c_t}(x^t, y^t). \tag{66}
$$

Thus, we have the following Lemma.

**Lemma A.13.** *Under Assumptions 5.2, 5.3 and 5.7, suppose* $\gamma \in (0, \frac{1}{2\kappa_{g_2}})$, $c_{t+1} \geq c_t$ *and* $\eta_t$ *satisfies* $\eta_t >$

$$
\frac{1 + \sqrt{1 + 4(L_{gx,0} + L_{gx,1}M^\rho + \frac{2}{\gamma^2}))\left((L_{gx,0} + L_{gx,1}M^\rho)\frac{\alpha_t}{\|\tilde{d}_x^t\|} + \frac{\beta_t}{\gamma^2\|\tilde{d}_y^t\|}\right)}}{2\kappa_{g_2}(L_{gx,0} + L_{gx,1}M^\rho + \frac{2}{\gamma^2})}, \textit{ we have}
$$

$$
\mathbb{E}[\Gamma_{t+1} - \Gamma_t|\mathcal{F}_t]
$$

$$
\leq -\frac{\alpha_t}{8}(\|\frac{1}{c_t}\nabla_1 f(x^t, y^t)\| + \|\nabla_1 g(x^t, y^t)\|) - \frac{\beta_t}{8}(\|\frac{1}{c_t}\nabla_2 f(x^t, y^t)\| + \|\nabla_2 g(x^t, y^t)\|) - \eta_t\kappa_{g_2}(\|\theta^t - \theta_\gamma^*(x^t, y^t)\|^2
$$

$$
+ \frac{\eta_t^2\delta_g^2}{B_t}) + \frac{2\delta}{B_t}(\alpha_t + \beta_t) \tag{67}
$$

*Proof.* By the definition of $\Gamma_t$. when $\rho \in [0, 1)$, we have

$$\mathbb{E}[\Gamma_{t+1} - \Gamma_t | \mathcal{F}_t]$$

$$\leq \mathbb{E}\left[\left(\frac{\alpha_t}{\|\tilde{d}_x^t\|}(L_{gx,0} + L_{gx,1}M^\rho) + \frac{2\beta_t}{\gamma^2\|\tilde{d}_y^t\|}\right)\|\theta^{t+1} - \theta_\gamma^*(x^t, y^t)\|^2\Big|\mathcal{F}_t\right]$$

$$- \frac{\alpha_t}{4}(\|\frac{1}{c_t}\nabla_1 f(x^t, y^t)\| + \|\nabla_1 g(x^t, y^t)\|) + \frac{3}{4}\alpha_t\|\nabla_1 g(x^t, \theta^{t+1})\| + \frac{\alpha_t^2}{2}\left(K_0 + \kappa + K_1 + 2K_2 + 2 + \frac{4\beta_t}{\gamma^2\|d_y^t\|}\right)$$

$$- \frac{\beta_t}{4}(\|\frac{1}{c_t}\nabla_2 f(x^{t+1}, y^t)\| + \|\nabla_2 g(x^{t+1}, y^t)\|) + \frac{\beta_t^2}{2}(K_3 + \kappa + K_4 + 2K_5 + 2) + \frac{\delta}{2B_t}(\alpha_t + \beta_t)$$

$$+ \left(L_{gx,0} + L_{gx,1}M^\rho + \frac{2}{\gamma^2}\right)\mathbb{E}[\|\theta^{t+1} - \theta_\gamma^*(x^{t+1}, y^{t+1})\|^2|\mathcal{F}_t] - \left(L_{gx,0} + L_{gx,1}M^\rho + \frac{2}{\gamma^2}\right)\|\theta^t - \theta_\gamma^*(x^t, y^t)\|^2$$

$$\leq -\frac{\alpha_t}{4}\left(\|\frac{1}{c_t}\nabla_1 f(x^t, y^t)\| + \|\nabla_1 g(x^t, y^t)\| - 3\|\nabla_1 g(x^t, \theta^{t+1})\| - 2\alpha_t\left(K_0 + \kappa + K_1 + 2K_2 + 2 + \frac{4\beta_t}{\gamma^2\|d_y^t\|}\right)\right)$$

$$- \frac{\beta_t}{4}\left(\|\frac{1}{c_t}\nabla_2 f(x^{t+1}, y^t)\| + \|\nabla_2 g(x^{t+1}, y^t)\| - 2\beta_t(K_3 + \kappa + K_4 + 2K_5 + 2)\right) + \frac{\delta}{2B_t}(\alpha_t + \beta_t)$$

$$+ \left(L_{gx,0} + L_{gx,1}M^\rho + \frac{2}{\gamma^2}\right)\mathbb{E}[\|\theta^{t+1} - \theta_\gamma^*(x^{t+1}, y^{t+1})\|^2|\mathcal{F}_t] - \left(L_{gx,0} + L_{gx,1}M^\rho + \frac{2}{\gamma^2}\right)\|\theta^t - \theta_\gamma^*(x^t, y^t)\|^2$$

$$+ \mathbb{E}\left[\left(\frac{\alpha_t}{\|\tilde{d}_x^t\|}(L_{gx,0} + L_{gx,1}M^\rho) + \frac{2\beta_t}{\gamma^2\|\tilde{d}_y^t\|}\right)\|\theta^{t+1} - \theta_\gamma^*(x^t, y^t)\|^2\Big|\mathcal{F}_t\right]. \tag{68}$$

Next, we demonstrate that

$$\mathbb{E}[\|\theta^{t+1} - \theta_\gamma^*(x^{t+1}, y^{t+1})\|^2 - \|\theta^t - \theta_\gamma^*(x^t, y^t)\|^2 + \frac{\alpha_t}{\|\tilde{d}_x^t\|}\|\theta^{t+1} - \theta_\gamma^*(x^t, y^t)\|^2|\mathcal{F}_t]$$

$$\leq \mathbb{E}\left[\left(1 + \nu_t + \frac{\alpha_t}{\|\tilde{d}_x^t\|}\right)\sigma_t^2(\|\theta^t - \theta_\gamma^*(x^t, y^t)\|^2 + \frac{\eta_t^2\delta_g^2}{B_t}) - \|\theta^t - \theta_\gamma^*(x^t, y^t)\|^2\right.$$

$$\left. + \left(1 + \frac{1}{\nu_t}\right)L_\theta^2\|(x^{t+1}, y^{t+1}) - (x^t, y^t)\|^2|\mathcal{F}_t\right], \tag{69}$$

where the last inequality follows from Lemma A.11.

Given $\nu_t = \eta_t\kappa_{g_2}$, we have

$$\mathbb{E}[\|\theta^{t+1} - \theta_\gamma^*(x^{t+1}, y^{t+1})\|^2 - \|\theta^t - \theta_\gamma^*(x^t, y^t)\|^2 + \frac{\alpha_t}{\|\tilde{d}_x^t\|}\|\theta^{t+1} - \theta_\gamma^*(x^t, y^t)\|^2|\mathcal{F}_t]$$

$$\leq \mathbb{E}\left[-\left(\eta_t^2\kappa_{g_2}^2 - \frac{\alpha_t}{\|\tilde{d}_x^t\|}\right)(\|\theta^t - \theta_\gamma^*(x^t, y^t)\|^2 + \frac{\eta_t^2\delta_g^2}{B_t}) + \left(1 + \frac{1}{\eta_t\kappa_{g_2}}\right)L_\theta^2\|(x^{t+1}, y^{t+1}) - (x^t, y^t)\|^2|\mathcal{F}_t\right]. \tag{70}$$

Similarly, we have

$$\mathbb{E}[\|\theta^{t+1} - \theta_\gamma^*(x^{t+1}, y^{t+1})\|^2 - \|\theta^t - \theta_\gamma^*(x^t, y^t)\|^2 + \frac{2\beta_t}{\|\tilde{d}_y^t\|}\|\theta^{t+1} - \theta_\gamma^*(x^t, y^t)\|^2|\mathcal{F}_t]$$

$$\leq \mathbb{E}\left[\left(1 + \nu_t + \frac{2\beta_t}{\|\tilde{d}_y^t\|}\right)\delta_t^2(\|\theta^t - \theta_\gamma^*(x^t, y^t)\|^2 + \frac{\eta_t^2\delta_g^2}{B_t}) - \|\theta^t - \theta_\gamma^*(x^t, y^t)\|^2\right.$$

$$\left. + \left(1 + \frac{1}{\nu_t}\right)L_\theta^2\|(x^{t+1}, y^{t+1}) - (x^t, y^t)\|^2|\mathcal{F}_t\right]$$

$$\leq \mathbb{E}\left[-\left(\eta_t^2\kappa_{g_2}^2 - \frac{2\beta_t}{\|\tilde{d}_y^t\|}\right)(\|\theta^t - \theta_\gamma^*(x^t, y^t)\|^2 + \frac{\eta_t^2\delta_g^2}{B_t}) + \left(1 + \frac{1}{\eta_t\kappa_{g_2}}\right)L_\theta^2\|(x^{t+1}, y^{t+1}) - (x^t, y^t)\|^2|\mathcal{F}_t\right]. \tag{71}$$

Putting the above inequalities (70), (71) into (69), we have

$$
\mathbb{E}[\Gamma_{t+1} - \Gamma_t | \mathcal{F}_t]
$$
$$
\leq - \frac{\alpha_t}{4}\left( \|\frac{1}{c_t}\nabla_1 f(x^t, y^t)\| + \|\nabla_1 g(x^t, y^t)\| - 3\|\nabla_1 g(x^t, \theta^{t+1})\| - 2\alpha_t\Big(K_0 + \kappa + K_1 + 2K_2 + 2 + \frac{4\beta_t}{\gamma^2\|d_y^t\|}\Big) \right.
$$
$$
\left. - 4\alpha_t\big(1 + \frac{1}{\eta_t\kappa_{g_2}}\big)L_\theta^2\Big(L_{gx,0} + L_{gx,1}M^\rho + \frac{2}{\gamma^2}\Big)\right) - \frac{\beta_t}{4}\left(\|\frac{1}{c_t}\nabla_2 f(x^{t+1}, y^t)\| + \|\nabla_2 g(x^{t+1}, y^t)\|\right.
$$
$$
\left. - 2\beta_t(K_3 + \kappa + K_4 + 2K_5 + 2) - 4\beta_t\big(1 + \frac{1}{\eta_t\kappa_{g_2}}\big)L_\theta^2(L_{gx,0} + L_{gx,1}M^\rho + \frac{2}{\gamma^2})\right) - \mathbb{E}\Big[\Big((L_{gx,0} + L_{gx,1}M^\rho + \frac{2}{\gamma^2})
$$
$$
\eta_t^2\kappa_{g_2}^2 - (L_{gx,0} + L_{gx,1}M^\rho)\frac{\alpha_t}{\|\tilde{d}_x^t\|} - \frac{\beta_t}{\gamma^2\|\tilde{d}_y^t\|}\Big)(\|\theta^t - \theta_\gamma^*(x^t, y^t)\|^2 + \frac{\eta_t^2\delta_g^2}{B_t})\Big|\mathcal{F}_t\Big] + \frac{\delta(\alpha_t + \beta_t)}{2B_t}. \tag{72}
$$

Given $0 < \alpha_t \leq \frac{\|\frac{1}{c_t}\nabla_1 f(x^t,y^t)\| + \|\nabla_1 g(x^t,y^t)\|}{4(K_0 + \kappa + K_1 + 2K_2 + 2) + 8(1 + \frac{1}{\eta_t\kappa_{g_2}})L_\theta^2(L_{gx,0} + L_{gx,1}M^\rho + \frac{2}{\gamma^2}) + 6\|\nabla_1 g(x^t, \theta^{t+1})\|}$,

$0 < \beta_t \leq \frac{\|\frac{1}{c_t}\nabla_2 f(x^t,y^t)\| + \|\nabla_2 g(x^t,y^t)\|}{4(K_3 + \kappa + K_4 + 2K_5 + 2) + 8(1 + \frac{1}{\eta_t\kappa_{g_2}})L_\theta^2(L_{gx,0} + L_{gx,1}M^\rho + \frac{2}{\gamma^2})}$, and $\eta_t >$

$\frac{1 + \sqrt{1 + 4(L_{gx,0} + L_{gx,1}M^\rho + \frac{2}{\gamma^2}))\Big((L_{gx,0} + L_{gx,1}M^\rho)\frac{\alpha_t}{\|\tilde{d}_x^t\|} + \frac{\beta_t}{\gamma^2\|\tilde{d}_y^t\|}\Big)}}{2\kappa_{g_2}(L_{gx,0} + L_{gx,1}M^\rho + \frac{2}{\gamma^2})}$, we can obtain

$$
\mathbb{E}[\Gamma_{t+1} - \Gamma_t | \mathcal{F}_t]
$$
$$
\leq -\frac{\alpha_t}{8}(\|\frac{1}{c_t}\nabla_1 f(x^t, y^t)\| + \|\nabla_1 g(x^t, y^t)\|) - \frac{\beta_t}{8}(\|\frac{1}{c_t}\nabla_2 f(x^t, y^t)\| + \|\nabla_2 g(x^t, y^t)\|) - \eta_t\kappa_{g_2}(\|\theta^t - \theta_\gamma^*(x^t, y^t)\|^2
$$
$$
+ \frac{\eta_t^2\delta_g^2}{B_t}) + \frac{2\delta(\alpha_t + \beta_t)}{B_t}. \tag{73}
$$

When $\rho = 1$, according to Lemma A.12, we have

$$
\mathbb{E}[\Gamma_{t+1} - \Gamma_t | \mathcal{F}_t]
$$
$$
\leq \mathbb{E}\left[\left(\Big(\frac{\alpha_t}{\|\tilde{d}_x^t\|}(L_{gx,0} + L_{gx,1}M) + \frac{2\beta_t}{\gamma^2\|\tilde{d}_y^t\|}\Big)\|\theta^{t+1} - \theta_\gamma^*(x^t, y^t)\|^2 - \frac{\alpha_t}{4}\left(\Big\|\frac{1}{c_t}\nabla_1 f(x^t, y^t)\Big\| + \|\nabla_1 g(x^t, y^t)\|\right)\right.\right.
$$
$$
+ \frac{3}{4}\alpha_t\|\nabla_1 g(x^t, \theta^{t+1})\| + \frac{\alpha_t^2}{2}\Big(L_0 + \kappa + L_1 + 2 + \frac{4\beta_t}{\gamma^2\|d_y^t\|}\Big) - \frac{\beta_t}{4}\left(\Big\|\frac{1}{c_t}\nabla_2 f(x^{t+1}, y^t)\Big\| + \|\nabla_2 g(x^{t+1}, y^t)\|\right)
$$
$$
+ \frac{\beta_t^2}{2}(L_2 + \kappa + L_3 + 2) + \big(L_{gx,0} + L_{gx,1}M + \frac{2}{\gamma^2}\big)\big\|\theta^t - \theta_\gamma^*(x^{t+1}, y^{t+1})\big\|^2 - \big(L_{gx,0} + L_{gx,1}M + \frac{2}{\gamma^2}\big)
$$
$$
\left.\left.\big\|\theta^t - \theta_\gamma^*(x^t, y^t)\big\|^2\right|\mathcal{F}_t\right]
$$
$$
\leq -\frac{\alpha_t}{4}\left(\Big\|\frac{1}{c_t}\nabla_1 f(x^t, y^t)\Big\| + \|\nabla_1 g(x^t, y^t)\| - 3\|\nabla_1 g(x^t, \theta^{t+1})\| - 2\alpha_t(L_0 + \kappa + L_1 + 2 + \frac{4\beta_t}{\gamma^2\|d_y^t\|})\right.
$$
$$
\left. - 2\alpha_t(1 + \frac{1}{\eta_t\kappa_{g_2}})L_\theta^2(L_{gx,0} + L_{gx,1}M + \frac{2}{\gamma^2})\right) - \frac{\beta_t}{4}\left(\Big\|\frac{1}{c_t}\nabla_2 f(x^{t+1}, y^t)\Big\| + \|\nabla_2 g(x^{t+1}, y^t)\|\right.
$$
$$
\left. - 2\beta_t(L_2 + L_3 + 2 + \kappa) - 2\beta_t(1 + \frac{1}{\eta_t\kappa_{g_2}})L_\theta^2(L_{gx,0} + L_{gx,1}M + \frac{2}{\gamma^2})\right) - \mathbb{E}\Big[\Big((L_{gx,0} + L_{gx,1}M + \frac{2}{\gamma})\eta_t^2\kappa_{g_2}^2
$$
$$
- (L_{gx,0} + L_{gx,1}M)\frac{\alpha_t}{\|\tilde{d}_x^t\|} - \frac{\beta_t}{\gamma^2\|\tilde{d}_y^t\|}\Big)(\|\theta^t - \theta_\gamma^*(x^t, y^t)\|^2 + \frac{\eta_t^2\delta_g^2}{B_t})\Big|\mathcal{F}_t\Big] + \frac{2\delta(\alpha_t + \beta_t)}{B_t},
$$

where the last inequality follows from Young's inequality with parameter $\eta_t\kappa_{g_2}$ and rearrangement.

Given $0 < \alpha_t \leq \frac{\|\frac{1}{c_t}\nabla_1 f(x^t,y^t)\| + \|\nabla_1 g(x^t,y^t)\|}{4(L_0 + \kappa + L_1 + 2) + 8(1 + \frac{1}{\eta_t\kappa_{g_2}})L_\theta^2(L_{gx,0} + L_{gx,1}M + \frac{2}{\gamma^2}) + 6\|\nabla_1 g(x^t, \theta^{t+1})\|}$, $0 < \beta_t \leq$

$$\frac{\|\frac{1}{c_t}\nabla_2 f(x^t,y^t)\|+\|\nabla_2 g(x^t,y^t)\|}{4(L_2+\kappa+L_3+2)+8(1+\frac{1}{\eta_t\kappa_{g_2}})L_\theta^2(L_{gx,0}+L_{gx,1}M+\frac{2}{\gamma^2})}, \text{ and } \eta_t > \frac{1+\sqrt{1+4(L_{gx,0}+L_{gx,1}M+\frac{2}{\gamma^2}))\left((L_{gx,0}+L_{gx,1}M)\frac{\alpha_t}{\|\bar{d}_x^t\|}+\frac{\beta_t}{\gamma^2\|\bar{d}_y^t\|}\right)}}{2\kappa_{g_2}(L_{gx,0}+L_{gx,1}M+\frac{2}{\gamma^2})},$$

we can also obtain

$$\mathbb{E}[\Gamma_{t+1}-\Gamma_t|\mathcal{F}_t]$$
$$\leq -\frac{\alpha_t}{8}(\|\frac{1}{c_t}\nabla_1 f(x^t,y^t)\|+\|\nabla_1 g(x^t,y^t)\|)-\frac{\beta_t}{8}(\|\frac{1}{c_t}\nabla_2 f(x^t,y^t)\|+\|\nabla_2 g(x^t,y^t)\|)-\eta_t\kappa_{g_2}(\|\theta^t-\theta_\gamma^*(x^t,y^t)\|^2$$
$$+\frac{\eta_t^2\delta_g^2}{B_t})\Big|\mathcal{F}_t\Big]+\frac{2\delta(\alpha_t+\beta_t)}{B_t}.$$

$\square$

**Theorem A.14.** (*Restatement of Theorem 5.8*) *Under Assumptions 5.1, 5.2, 5.3, 5.4 and 5.7, given $\gamma \in (0,\frac{1}{2\kappa_{g_2}})$, $c_t = \underline{c}(t+1)^{1/4}$ with $\underline{c} > 0$ and $B_t = O(T^{1/2})$, and when $\rho \in [0,1)$ let $0 < \alpha_t \leq \frac{\|\frac{1}{c_t}\nabla_1 f(x^t,y^t)\|+\|\nabla_1 g(x^t,y^t)\|}{4(K_0+\kappa+K_1+2K_2+2)+8(1+\frac{1}{\eta_t\kappa_{g_2}})L_\theta^2 C+6\|\nabla_1 g(x^t,\theta^{t+1})\|}$, $0 < \beta_t \leq \frac{\|\frac{1}{c_t}\nabla_2 f(x^t,y^t)\|+\|\nabla_2 g(x^t,y^t)\|}{4(K_3+\kappa+K_4+2K_5+2)+8(1+\frac{1}{\eta_t\kappa_{g_2}})L_\theta^2 C}$, when $\rho = 1$ let $0 < \alpha_t \leq \frac{\|\frac{1}{c_t}\nabla_1 f(x^t,y^t)\|+\|\nabla_1 g(x^t,y^t)\|}{4(L_0+\kappa+L_1+2)+8(1+\frac{1}{\eta_t\kappa_{g_2}})L_\theta^2 C+6\|\nabla_1 g(x^t,\theta^{t+1})\|}$, $0 < \beta_t \leq \frac{\|\frac{1}{c_t}\nabla_2 f(x^t,y^t)\|+\|\nabla_2 g(x^t,y^t)\|}{4(L_2+\kappa+L_3+2)+8(1+\frac{1}{\eta_t\kappa_{g_2}})L_\theta^2 C}$, and $\eta_t \in \left(\frac{1+\sqrt{1+4C\left((L_{gx,0}+L_{gx,1}M)\frac{\alpha_t}{\|\bar{d}_x^t\|}+\frac{\beta_t}{\gamma^2\|\bar{d}_y^t\|}\right)}}{2\kappa_{g_2}C},\frac{1/\gamma-\kappa_{g_2}}{(1/\gamma+L_{gy,0}+L_{gy,1}\max_{\mu\in(0,1)}\|\nabla_2 g(x,\theta_\gamma^*(x,y)_\mu)\|^\rho)^2}\right)$, where $C = L_{gx,0} + L_{gx,1}M^\rho + \frac{2}{\gamma^2}$. Then the sequence of $(x^t,y^t,\theta^t)$ generated by Algorithm 2 satisfies*

$$\min_{0\leq t\leq T}\mathbb{E}[\|\theta^t-\theta_\gamma^*(x^t,y^t)\|]=O(\frac{1}{T^{1/2}}),$$
$$\min_{0\leq t\leq T}\mathbb{E}[R_t(x^{t+1},y^{t+1})]=O(\frac{1}{T^{1/6}}),$$
$$\mathbb{E}[g(x^T,y^T)-v_\gamma(x^T,y^T)]=O(\frac{1}{T^{1/4}}).$$

*Proof.* By using the above Lemma A.13, telescoping the above inequality (67) over the range $t = 0, 1, \cdots, T-1$, given $\underline{\eta}=\min_{0\leq t\leq T-1}\eta_t$, and take the total expectation, we have

$$\mathbb{E}[\Gamma_T]-\mathbb{E}[\Gamma_0]$$
$$\leq \sum_{t=0}^{T-1}\mathbb{E}\Big[-\frac{\alpha_t}{8}(\|\frac{1}{c_t}\nabla_1 f(x^t,y^t)\|+\|\nabla_1 g(x^t,y^t)\|)-\frac{\beta_t}{8}(\|\frac{1}{c_t}\nabla_2 f(x^t,y^t)\|+\|\nabla_2 g(x^t,y^t)\|)$$
$$-\eta_t\kappa_{g_2}(\|\theta^t-\theta_\gamma^*(x^t,y^t)\|^2+\frac{\eta_t^2\delta_g^2}{B_t})+\frac{2\delta(\alpha_t+\beta_t)}{B_t}\Big]$$
$$\leq \sum_{t=0}^{T-1}\mathbb{E}\Big[\Big(-\frac{\alpha_t}{8}(\|\frac{1}{c_t}\nabla_1 f(x^t,y^t)\|+\|\nabla_1 g(x^t,y^t)\|)-\frac{\beta_t}{8}(\|\frac{1}{c_t}\nabla_2 f(x^t,y^t)\|+\|\nabla_2 g(x^t,y^t)\|)$$
$$-\underline{\eta}\kappa_{g_2}(\|\theta^t-\theta_\gamma^*(x^t,y^t)\|^2+\frac{\underline{\eta}^2\delta_g^2}{B_t})\Big)+\frac{2\delta(\alpha_t+\beta_t)}{B_t}\Big]. \tag{74}$$

It follows that

$$\underline{\eta}\kappa_{g_2}\sum_{t=0}^{T-1}\mathbb{E}[\|\theta^t-\theta_\gamma^*(x^t,y^t)\|^2]\leq\mathbb{E}[\Gamma_0]-\mathbb{E}[\Gamma_T]+\sum_{t=0}^{T-1}\mathbb{E}\Big[\Big(-\frac{\alpha_t}{8}(\|\frac{1}{c_t}\nabla_1 f(x^t,y^t)\|+\|\nabla_1 g(x^t,y^t)\|)$$
$$-\frac{\beta_t}{8}(\|\frac{1}{c_t}\nabla_2 f(x^t,y^t)\|+\|\nabla_2 g(x^t,y^t)\|)-\underline{\eta}\kappa_{g_2}\frac{\underline{\eta}^2\delta_g^2}{B_t}\Big)+\frac{2\delta(\alpha_t+\beta_t)}{B_t}\Big)\Big]$$

and let $B_t = O(T^{1/2})$,

$$\sum_{i=0}^{T-1} \frac{1}{B_t} = O(T^{1/2}).$$

Consequently,

$$\min_{0 \le t \le T} \mathbb{E}[\|\theta^t - \theta_\gamma^*(x^t, y^t)\|] = O(\frac{1}{T^{1/2}}).$$

According to the update rule of variables $(x, y)$ in Algorithm 2, we have

$$c_t \frac{d_x^t}{\|\tilde{d}_x^t\|} + \frac{c_t}{\alpha_t}(x^{t+1} - x^t) = 0, \tag{75}$$

$$c_t \frac{d_y^t}{\|\tilde{d}_y^t\|} + \frac{c_t}{\beta_t}(y^{t+1} - x^t) = 0. \tag{76}$$

Let

$$(e_x^t, e_y^t) = \nabla f(x^{t+1}, y^{t+1}) + c_t(\nabla g(x^{t+1}, y^{t+1}) - v_\gamma(x^{t+1}, y^{t+1})),$$

with

$$e_x^t := c_t \nabla_1 \Phi_{c_t}(x^{t+1}, y^{t+1}) - c_t \tilde{d}_x^t - \frac{c_t \|\tilde{d}_x^t\|}{\alpha_t}(x^{t+1} - x^t), \tag{77}$$

$$e_y^t := c_t \nabla_2 \Phi_{c_t}(x^{t+1}, y^{t+1}) - c_t \tilde{d}_y^t - \frac{c_t \|\tilde{d}_y^t\|}{\beta_t}(y^{t+1} - y^t). \tag{78}$$

Considering the term $\|e_x^t\|$, we have

$$\begin{aligned}
\mathbb{E}[\|e_x^t\| | \mathcal{F}_t] \le & \mathbb{E}[c_t \|\nabla_1 \Phi_{c_t}(x^{t+1}, y^{t+1}) - \nabla_1 \Phi_{c_t}(x^t, y^t)\| + \|c_t \nabla_1 \Phi_{c_t}(x^t, y^t) - c_t d_x^t\| + \frac{c_t \|\tilde{d}_x^t\|}{\alpha_t}\|x^{t+1} - x^t\| \\
& + \|c_t d_x^t - c_t \tilde{d}_x^t\| | \mathcal{F}_t] \\
\le & \mathbb{E}[L_{\Phi_1}\|(x^{t+1}, y^{t+1}) - (x^t, y^t)\| + c_t \|\nabla_1 \Phi_{c_t}(x^t, y^t) - d_x^t\| + \frac{c_t \|\tilde{d}_x^t\|}{\alpha_t}\|x^{t+1} - x^t\| + c_t \sqrt{\frac{\delta}{B_t}} | \mathcal{F}_t] \\
= & \mathbb{E}[c_t L_{\Phi_1}\|(x^{t+1}, y^{t+1}) - (x^t, y^t)\| + \frac{c_t \|\tilde{d}_x^t\|}{\alpha_t}\|x^{t+1} - x^t\| + c_t \|\nabla_1 g(x^t, \theta^t) - \nabla_1 g(x^t, \theta_\gamma^*(x^t, y^t))\| | \mathcal{F}_t] \\
& + c_t \sqrt{\frac{\delta}{B_t}} \\
\le & \mathbb{E}[c_t L_{\Phi_1}\|(x^{t+1}, y^{t+1}) - (x^t, y^t)\| + \frac{c_t \|\tilde{d}_x^t\|}{\alpha_t}\|x^{t+1} - x^t\| + c_t(L_{gx,0} + L_{gx,1}M^\rho)\|\theta^t - \theta_\gamma^*(x^t, y^t)\| | \mathcal{F}_t] \\
& + c_t \sqrt{\frac{\delta}{B_t}}.
\end{aligned}$$

Thus, we can obtain

$$\mathbb{E}[\|e_x^t\| | \mathcal{F}_t]$$
$$\le \mathbb{E}[c_t L_{\Phi_1}\|(x^{t+1}, y^{t+1}) - (x^t, y^t)\| + \frac{c_t \|\tilde{d}_x^t\|}{\alpha_t}\|x^{t+1} - x^t\| + c_t(L_{gx,0} + L_{gx,1}M^\rho)\|\theta^t - \theta_\gamma^*(x^t, y^t)\| | \mathcal{F}_t] + c_t \sqrt{\frac{\delta}{B_t}}.$$

Similarity, based on Lemma A.6, we can obtain

$$\mathbb{E}[\|e_y^t\| | \mathcal{F}_t]$$
$$\le \mathbb{E}[c_t L_{\Phi_2}\|(x^{t+1}, y^{t+1}) - (x^t, y^t)\| + \frac{c_t \|\tilde{d}_y^t\|}{\beta_t}\|y^{t+1} - y^t\| + \frac{c_t}{\gamma}(\|\theta^t - \theta_\gamma^*(x^t, y^t)\| + L_\theta\|x^{t+1} - x^t\|) | \mathcal{F}_t] + c_t \sqrt{\frac{\delta}{B_t}},$$

where $L_{\Phi_2} = \frac{1}{c_t}(L_{fy,0} + L_{fy,1} \max_{\mu \in [0,1]} \|\nabla_2 f(x_\mu, y_\mu)\|^\rho) + L_{gy,0} + L_{gy,1} \max_{\mu \in [0,1]} \|\nabla_2 g(x_\mu, y_\mu)\|^\rho + \frac{1+L_\theta}{\gamma}$. Since $R_t(x,y) = \text{dist}(0, \nabla f(x,y) + c_t(\nabla g(x,y) - \nabla v_\gamma(x,y)))$, we have

$$\mathbb{E}[R_t(x^{t+1}, y^{t+1})] \leq \mathbb{E}[c_t L_\Phi \|(x^{t+1}, y^{t+1}) - (x^t, y^t)\| + \left(\frac{c_t\|\tilde{d}_x^t\|}{\alpha_t} + \frac{c_t L_\theta}{\gamma}\right)\|x^{t+1} - x^t\| + \frac{c_t\|\tilde{d}_y^t\|}{\beta_t}\|y^{t+1} - y^t\|$$

$$+ c_t(L_{gx,0} + L_{gx,1}M^\rho + \frac{1}{\gamma})\|\theta^t - \theta_\gamma^*(x^t, y^t)\|] + 2c_t\sqrt{\frac{\delta}{B_t}}.$$

Given $C_R' \leq \frac{\min\left\{\|\frac{1}{c_t}\nabla_1 f(x^t,y^t)\| + \|\nabla_1 g(x^t,y^t)\|, \|\frac{1}{c_t}\nabla_2 f(x^t,y^t)\| + \|\nabla_2 g(x^t,y^t)\|\right\}}{\underline{\eta}\kappa_{g_2} \max\{\|\tilde{d}_x^t\|, \|\tilde{d}_y^t\|\}}$, we have

$$\frac{1}{c_t^2}\mathbb{E}[R_t(x^{t+1}, y^{t+1})^2]$$

$$\leq C_R'\left(\frac{\alpha_t}{8}(\|\frac{1}{c_t}\nabla_1 f(x^t,y^t)\| + \|\nabla_1 g(x^t,y^t)\|) + \frac{\beta_t}{8}(\|\frac{1}{c_t}\nabla_2 f(x^t,y^t)\| + \|\nabla_2 g(x^t,y^t)\|) + \underline{\eta}\kappa_{g_2}\|\theta^t - \theta_\gamma^*(x^t,y^t)\|^2\right)$$

$$+ \frac{4\delta}{B_t}.$$

By using the above inequality (74), then we have

$$\sum_{t=0}^{T-1} \frac{1}{c_t^2}\mathbb{E}[R_t(x^{t+1}, y^{t+1})^2] \leq \mathbb{E}[\Gamma_0] - \mathbb{E}[\Gamma_T] + \sum_{i=0}^{T-1}\frac{2\delta}{B_t}(2 + \alpha_t + \beta_t)$$

$$\leq \mathbb{E}[\Gamma_0] - \mathbb{E}[\Gamma_{inf}] + \sum_{i=0}^{T-1}\frac{2\delta}{B_t}(2 + \alpha_t + \beta_t)$$

$$\leq l_1 + 4\delta l_2 \sum_{i=0}^{T-1}\frac{1}{B_t}, \tag{79}$$

where $\mathbb{E}[\Gamma_{inf}]$ is the lower bound of $\mathbb{E}[\Gamma_i]$, $l_2$ is the upper bound of $\alpha_t + \beta_t$ from $i \in \{0, \cdots, T\}$ and $l_1, l_2$ are positive constants.

Based on the above inequalities (79), let $c_t = \underline{c}(t+1)^{1/4}$, we can obtain

$$\min_{0 \leq t \leq T}\mathbb{E}[R_t(x^{t+1}, y^{t+1})] = O(\frac{1}{T^{1/6}}).$$

Since $\{x^t, y^t\}_{0 \leq t \leq T}$ is a sequence generated from Algorithm 2, we also can find $m' = \max_{0 \leq t \leq T}\mathbb{E}[\Phi_{c_t}(x^t, y^t)] = \frac{1}{c_t}f(x^t, y^t) + g(x^t, y^t) - v_\gamma(x^t, y^t)$, and $f(x^t, y^t) \geq f^*$ for any $t$. Then we have

$$c_t\mathbb{E}[(g(x^t, y^t) - v_\gamma(x^t, y^t))] \leq m' - f^*, \quad \forall t > 0.$$

Based on $c_t = \underline{c}(t+1)^{1/4}$, we can obtain

$$\mathbb{E}[g(x^T, y^T) - v_\gamma(x^T, y^T)] = O(\frac{1}{T^{1/4}}).$$

$\square$

