# OpenReview forum: "Generalized Smooth Bilevel Optimization with Nonconvex Lower-Level"
_ICML.cc/2025/Conference — ICML 2025 poster_

### Official Review · Reviewer_RaRE · 2025-03-10

**Overall Recommendation:** 2

**Summary:**

The paper investigates bilevel optimization problems where the inner function is nonconvex, and both inner and outer functions satisfy a generalized smoothness assumption. To address this problem, the single-level constrained formulation of bilevel optimization is adopted, replacing the inner problem with the Moreau envelope of the inner function. A novel algorithm, PNGBiO, is introduced, which involves performing normalized gradient steps on the Lagrangian function of this constrained problem. The algorithm demonstrates a convergence rate of O(T−1/4)O(T−1/4). Additionally, the paper presents experimental results for two bilevel problems: hyperparameter tuning and data hypercleaning.

**Claims And Evidence:**

I review the theoretical claims in the section **Theoretical Claims** and the experimental evidences in the section **Experimental Designs or analyses**.

**Essential References Not Discussed:**

N/A

**Experimental Designs Or Analyses:**

### Reproducibility
The numerical part presents several reproducibility issues:
* **Code**: The code of the experiments is not provided, which does not foster the reproducibility of the results. This is an important concern.
* **Step size choice**: How are set the step sizes? Is it a grid search? In this case, the parameters of the grid should be precised. Why some algorithms use decreasing step sizes and other not?
* **Constraint**: For the contraint on $\lambda$ in the data hyper-cleaning task, how is it handled in the different algorithms which are originally designed for unconstrained optimization? How is chosen $R$?
* **Inner loop**: Some algorithms use inner loops (F2SA, BOME), what is the size of the inner loop?
* **Batch size for stochastic algorithms**: F2SA and SLIP are stochastic algorithms. What are the batch sizes used?
* **Number of runs**: How many runs are performed for each algorithm? Since some algorithms are stochastic, a single run is not enough to draw conclusions and error bars should be provided.

### Presentation/clarity
* **x-axis of the figures**: In figures 2 to 5, the x-axis is labeled as *"Epoch"*. What does the word "Epoch" refers to? In the context of bilevel optimization, the notion of "epoch" is fuzzy since there are two datasets (training and validation). Moreover, some algorithms use an inner loop or not, some others use second order information of the lower level problem or not, thus to be perfectly fair, curves in terms of wall-clock time would be more appropriate.

* **Data hypercleaining**: It would be convinient to be more precise when talking about *"contaminated training data"*. Reader ununfamiliar with data hyper-cleaning should understand by reading the paper that *"contaminated training data"* means some labels of the training data are wrong. Moreover, it should be clearly to write explicitly the bilevel problem considered, as done in the hyperparameter tuning problem.

* **Hyperparameter learning**: what is the regularizer $\mathcal{R_{\omega, \lambda}}$?

### Design
* **No nonconvex lower-level**. If I understand appropriately, in both cases, the lower level problem is a multinomial logistic regression problem. But this problem is convex and thus this is a concern that there are no experiments with nonconvex lower-level problem when the title of the paper suggests treating this case.

**Methods And Evaluation Criteria:**

See the section **Experimental Designs or analyses** for the evaluation criteria.

**Other Comments Or Suggestions:**

### Typos/Unclear sentences
* **Abstract**: *"we stydy the convergence analysis"* -> *"we study the convergence property"*
* **Line 032**: *"When the lower-level objective function $g(x, y)$ on the variable y is strongly convex and twice differential"* -> *"When the lower-level objective $g$ is strongly convex and twice differentiable with respect to $y$"*
* **Line 043**: *"by using the following approximated gradient"* -> *"by using the following approximate gradient"*
* **Line 049**: *"where $\hat y$ is an approximate of the solution $y^*(x)$"* -> *"where $\hat y$ is an approximation of the inner solution $y^*(x)$"* or *"where $\hat y$ is an approximate solution of the inner problem $y^*(x)$"*
* **Line 82**: *"can not"* -> *"cannot"*
* **Lines 95-98**: *"More recently, (Liu et al., 2024) solve a variant of the single-level constrained optimization problem (5), where uses its Moreau envelope instead of the problem $\min_{y\in\mathbb{R}^p} g(x, y)$ in the problem (5), defined as"* -> *"More recently, Liu et al. (2024) solve a variant of the single-level constrained optimization problem (5), where the Moreau envelope of the inner  function is used as a surrogate of the inner problem $\min_{y\in\mathbb{R}^p} g(x, y)$ in (5). This yields the following problem formulation"*
* **Lines 109, 66-67**: *"More recently, thus, some methods (Hao et al., 2024; Gong et al., 2024) have begun to study generalized smooth bilevel optimization."* -> *"More recently, some authors (Hao et al., 2024; Gong et al., 2024) started to study bilevel optimization under generalized smoothness assumption."*
* **Lines 106**: *"The generalized smoothness condition firstly was studied"* -> *"The generalized smoothness condition was first studied"*
* **Lines 115-120**: *"In the problem (1), since lower-level objective $g(x, y)$ is
nonconvex on variable y, we can not easily get the minimum of lower-level problem $\min_{y\in\mathbb{R}^p} g(x, y)$. Thus, we reformulate the lower-level problem by using a value function defined as: $v(x) = \min_{y\in\mathbb{R}^p} g(x, y)$."* The logical sequence of these sentences is unclear. It textually tells that, when the inner function in non-convex, solving $\min_{y\in\mathbb{R}^p} g(x, y)$ is too difficult, and thus it is replaced by $v(x)$ which is actually the same thing.
* **Assumption 5.2 and 5.3**: *"there exists"* -> *"there exist"*
* **Assumption 5.2 and 5.3**: comma missing before *"and"*: *"$L_{fy,0}$ and $L_{fy,1}$"* ->  *"$L_{fy,0}$, and $L_{fy,1}$"*; *"$L_{gy,0}$ and $L_{gy,1}$"* ->  *"$L_{gy,0}$, and $L_{gy,1}$"*
* **Line 320**: *"the above three terms shows"* -> *"the above three terms show"*
* **Lines 417 to 425**: there are several issues in the definition of the bilevel problem:
    * $\sum_{x_i \in S_{val}}$ -> $\sum_{i \in S_{val}}$
    * $\omega^*$ -> $\omega^*(\lambda)$
    * $\omega^* = \mathcal{L}_{S_{tr}}(\omega, \lambda)$ -> $\omega^* = \mathcal{L}_{S_{tr}}(\omega^*, \lambda^*)$ -> $\omega^*(\lambda) = \underset{\omega}{\mathrm{arg\,\min}}\mathcal{L}_{S_{tr}}(\omega, \lambda)$
    * the third line of the equation should also be corrected.

### Notation consistency
* **Equations (2) and (3)**: The partial gradients are denoted with number in subscript while the partial Hessians are denoted with letter in subscript. It would be more consistent to use the same notation for both.
* **Numeratical experiments**: In the data hypercleaning problem, the individual losses are denoted $\ell$ while in the hyperparameter tuning problem, they are denoted $\mathcal{L}$. Moreover, the full loss function are denoted $L$ in the data hypercleaning problem and $\mathcal{L}$ in the hyperparameter tuning problem. Moreover, in the bilevel problem between lines 417 and 425, the individual train losses depend on $\lambda$ while the individual validation losses do not.

### Miscellanous
* **Line 024**: *"and distributionally robust optimization"*: there should be a citation for this task.

* **Line 157**: *"Based on Theorem A.4 of (Liu et al., 2024), it is known that any limit point $(\bar x, \bar y)$ of sequence $(x^t, y^t)$ is a solution to the problem (9)."* The sequence $(x^t, y^t)$ is not defined.

* **Correct citation formating**: when the authors' name of a paper is part of the sentence, they should not be in parenthesis (in LaTeX it consists in using `\cite`or `\citet` instead of `\citep` I think). For instance:
    * line 50: *"(Ghadimi & Wang, 2018)
proposed a class of approximated gradient methods based
on approximate implicit differentiation. (Ji et al., 2021) proposed
an effective approximated gradient methods based on the iterative differentiation. More recently, (Huang, 2023; 2024) studied the nonconvex bilevel optimization"* -> *"Ghadimi & Wang (2018) proposed a class of approximated gradient methods based on approximate implicit differentiation. Ji et al. (2021) proposed an effective approximated gradient methods based on the iterative differentiation. More recently, Huang (2023; 2024) studied the nonconvex bilevel optimization"*
    * Line 156: *"Based on Theorem A.4 of (Liu et al., 2024)"* -> *"Based on Theorem A.4 of Liu et al. (2024)"*

* **Precision on AID-based methods**: Equation (3) makes think that the linear system that appears in the hypergradient is solved exactly. This is not the case. As the solution of the inner problem is approximated, the solution linear system is also approximated.

* **Page header**: page header should be changed. Now it is still *"Submission and Formatting Instructions for ICML 2025"*.


* **Notation definition**: In lemma 5.6, $\theta^*_\gamma$ is used without being defined. Actually, it is defined in the appendix. It should be defined before being used.

### Mathematical writting
* **Assumptions 5.2. and 5.3.**: Are the constants $L_{fx,0}
$, $L_{fx,1}$, $L_{fy,0}$, $L_{fy,1}$, $L_{gx,0}$, $L_{gx,1}$, $L_{gy,0}$, and $L_{gy,1}$, independent from $\rho$? In this case, it is necessary to revers the order of the quantifiers by writting *"There exist constants $L_{fx,0}$,..., and $L_{fy,1}$ such that for any $\rho>0$"*. Otherwise, it means that these constants depend on $\rho$ and in this case, this dependency should be made explicit.

**Other Strengths And Weaknesses:**

### Strengths
* To my knowledge, this is the first time that a fully fisrt-order algorithm is proposed for bilevel optimization under the generalized smoothness assumption.

* The paper comes with non asymptotical convergence rates.

### Weaknesses
* **Presentation/writing**: The presentation and the writing of the paper need to be significantly improved. There are many typos, grammar issues and sentences that need to be rephrased. I provide a non-exhaustive list of these issues in the section "Other Comments Or Suggestions".

* **Section 5, page 4, 5**: the presentation of the theoretical results is really hard to parse. There is no transition between the different proposition/lemmas and they are not discussed/explained. Moreover, letting all the raw constants in the main text blurs the message. I think this section requires in profund clean up by moving some lemmas in the appendix, only keeping the most important quantities in the main text, and providing some highlevel explanations of the results.

* The algorithms comes with many hyperparameters (penalty parameter $c_t$, step sizes $\alpha_t$, $\beta_t$, and $\eta_t$, proximal parameter $\gamma$). This can make it difficult to use in practice.

**Questions For Authors:**

* In many machine learning problems, having stochastic algorithms is crucial to handle large datasets. Is it easy to extend PNGBiO to the stochastic setting?

* **Lines 205-208**: The papers claims that the generalized smoothness assumption used in Assumptions 5.1 and 5.2 is milder than the one usually used. But, I don't really see why since the proposed smoothness condition is supposed to hold for any $\rho\in[0,1]$ and not only at $\rho = 1$. Could the authors clarify this point? Moreover, is there any example of such function in machine learning?

* Can the convergence result provided by the paper be translated in terms of convergence of the norm the hypergradient (equation (2) in the paper) when the inner function is strongly convex?

**Relation To Broader Scientific Literature:**

On the one hand, this paper extends the algorithm proposed in [1] to scenarios where both the inner and outer functions satisfy a generalized smoothness assumption. On the other hand, [2, 3] consider bilevel algorithm  where only the outer function meets this assumption, while the inner function is strongly convex. These algorithms are based on approximate implicit differentiation. Thus, the method proposed in this paper complement this line of work by considering the generalized smoothness assumption on both functions and providing a fully first-order algorithm for this case.

[1] Liu, R., Liu, Z., Yao, W., Zeng, S., and Zhang, J. *Moreau envelope for nonconvex bi-level optimization: A single- loop and hessian-free solution strategy*. arXiv preprint arXiv:2405.09927, 2024.

[2] Hao, J., Gong, X., and Liu, M. *Bilevel optimization under unbounded smoothness: A new algorithm and convergence analysis*. ICLR 2024.

[3] Gong, X., Hao, J., and Liu, M. *A nearly optimal single-loop algorithm for stochastic bilevel optimization under unbounded smoothness*. ICML, 2024.

**Theoretical Claims:**

See the supplementary material section for a review of the proof of the theoretical results.
Besides the proofs, I have the following remark:

* **Stationnary point of problem (1)**: Several times (lines 130, 322, 324), the paper refers to stationary points of problem (1).  What is the meaning of *stationary point* in this context since the lower level problem is not assumed to be strongly convex?

---

> ### Author Rebuttal · Authors · 2025-04-01
>
> Thanks so much for your comments and suggestions. We response to your questions one by one as follows:
>
> **Q1:** What is the meaning of stationary point in this context since the lower level problem is not assumed to be strongly convex?
>
> **A1:** In our paper, we use the same definition of stationary point as in [1]. From [1], the stationary point of problem (1) is equivalent to that of the relaxed problem $\min_{(x,y)\in \mathbb{R}^d \times \mathbb{R}^q} f(x,y) s.t. \nabla_y g(x,y)=0$. When the lower-level of problem (1) is strongly convex, this stationary point is equivalent to the stationary point used in [2,3].
>
> [1] Liu et al., Moreau envelope for nonconvex bi-level optimization: A single-loop and hessian-free solution strategy. arXiv preprint arXiv:2405.09927, 2024.
>
> [2] Ghadimi, S. and Wang, M. Approximation methods for bilevel programming. arXiv preprint arXiv:1802.02246, 2018.
>
> [3] Ji, et al., Bilevel optimization: Convergence analysis and enhanced design. ICML, 4882–4892, 2021.
>
> **Q2:** ... Is it easy to extend PNGBiO to the stochastic setting?
>
> **A2:** Our PNBiO can be generalized to the stochastic setting. We have initially proved that our stochastic PNGBiO has a gradient complexity of $O(\epsilon^{-6})$. In the final version of our paper, we will add this conclusion.
> Meanwhile, we add some experimental results on our stochastic PNGBiO (**S-PNGBiO**) method.  Specifically,  we compare our S-PNBiO algorithm with other baselines including BO-REF, F^2SA, SLIP, BOME. In the experiment, we conduct the meta-learning task at CIFAR10 dataset,  and use Resnet18 as task-shared model at the upper-level (UL) problem, and use a 2-layer neural network as task-specific model at the lower-level (LL) problem. Clearly, both UL and LL problems are non-convex. We run every method for 500 seconds (CPU time)  with minibatch size 64 for both training and validation set. We set learning rates as 0.01 at F^2SA and BOME algorithms,  and $\alpha=0.0001$, $\beta=0.01$ in SLIP and BO-REF. Our algorithm uses a vanishing step size, which is $\alpha_t=\beta_t=0.3/(k+1)^{0.5}$,$\eta=0.01$, the rest of the parameter settings are the same as in the paper, and the experimental results are shown in the following table (we have recorded the loss):
> |  | S-PNBiO| F$^2$SA |BOME|BO-REF|SLIP|
> |-------|-------|-------|-------|-------|-------|
> |100s|1.591| 1.822| 1.831|1.691|1.948|
> |200s|1.494 |1.655|1.692|1.585|1.839|
> |300s|1.441|1.568|1.615|1.464|1.775|
> |400s|1.392|1.485|1.542|1.402|1.723|
> |500s|1.371|1.425|1.467|1.382|1.601|
>
> In the final version of our paper, we will add these experimental results.
>
> **Q3:** Lines 205-208:  The papers claims that the generalized smoothness assumption…, is there any example of such function in machine learning?
>
> **A3:** 1) When $\rho=0$, our generalized smoothness condition degenerates to standard Lipschitz smoothness condition. 2) When $\rho\in(0,1]$, this smoothness condition depends on the current gradient. For example, the polynomial function $f(x)=|x|^\frac{2-\rho}{1-\rho}$, where $x\in\mathbb{R}$ and $\rho\in(0,1)$, satisfies this generalized smoothness condition. In this case, it does not apply when $\rho=1$.
>
> **Q4:** Can the convergence result provided by the paper be translated in terms of convergence of the norm the hyper-gradient (equation (2) in the paper) when the inner function is strongly convex?
>
> **A4:** When inner function is strongly convex, we have proved that our method has the same convergence rate $O(\frac{1}{T^{1/4}})$ based on the hyper-gradient metric as in [1], which shows  our algorithm has the same rate of convergence under the generalized smooth condition as  the  standard smooth condition. In the final version, we will add these results.
>
> [1] Liu et al., Moreau envelope for nonconvex bi-level optimization: A single-loop and hessian-free solution strategy. arXiv preprint arXiv:2405.09927, 2024.
>
> **Q5:** Some questions of experimental designs.
>
> **A5:** 1. In the experiments, the tuning parameters of all algorithms are selected to be optimal through the grid.
>
> 2. In the data cleaning experiment, the problem is based on unconstrained conditions, so here $R$ should be infinite, and here is a typo.
>
> 3. In the algorithms that require inner loops, we set 10 times.
>
>
> **Q6:** The partial gradients are denoted with number in subscript while the partial Hessians are denoted with letter in subscript...
>
> **A6:** In the paper, we have a different definition for the subscripts: 1 and 2 represent the derivatives with respect to the first and second parts, respectively. We will clarify this in the final version of our paper.
>
> **Q7:** Questions about essay structure, mathematical writing and typos.
>
> **A7:**  Thanks so much for your suggestions. In the final version of our paper, we will correct these typos and mathematical writing, and will reorganize the section 5 of our paper, explaining the main results to make them clearer. Meanwhile, some theoretical results will be added.

---

> > ### Comment · Reviewer_RaRE · 2025-04-06
> >
> > Dear authors,
> >
> > I thank you for answering my review. I have the following remark:
> >
> >
> > **A3**: I am not sure I understand. According to the generalized smoothness condition provided in the paper (*e.g.*, in Assumption 5.2), a function $f$ verifies the generalized smoothness assumption if it verifies for any $u, u'$
> > $$
> > \lVert\nabla f(u) - \nabla f(u')\rVert \leq (L_{f, 0} + L_{f,1}\max_{\mu\in[0, 1]}\lVert f(u_{\mu})\rVert^{\rho})\lVert u - u'\rVert
> > $$
> > for **any** $\rho\in[0,1]$ and **not for a particular** $\rho\in[0,1]$. Thus in Assumptions 5.2 and 5.3, either the expression *"For any $\rho\in[0,1]$"* should be replaced by *"There exists $\rho\in[0, 1]$"*, either these assumptions are more restrictive than the usual smoothness.

---

> > > ### Author Response · Authors · 2025-04-07
> > >
> > > Dear Reviewer RaRE,
> > >
> > > Thanks so much for your suggestion. It is more accurate to articulate the condition in Assumptions 5.2 and 5.3 as “there exists $\rho \in [0,1]$.” Our intention was to highlight the applicability of the algorithm across the entire range of $\rho \in [0,1]$ . Considering that it may cause readers to misunderstand, we will make the necessary corrections in the final version.
> > >
> > > Best wishes,
> > >
> > > Authors

---

### Official Review · Reviewer_5XnN · 2025-03-12

**Overall Recommendation:** 4

**Summary:**

This paper proposes a gradient-based first-order algorithm called PNGBiO for generalized-smooth nonconvex-weakly-concave bilevel optimization. The authors provide the convergence analysis and claim that this algorithm achieve a convergence rate of O(ϵ^(-4)) for finding an approximation stationary point. The authors conduct some experiments to show the effectiveness of the proposed algorithm.

**Claims And Evidence:**

This article is technically sound. The theory seems to be correct

**Essential References Not Discussed:**

no

**Experimental Designs Or Analyses:**

Regarding the experiments, the authors did not conduct the comparison with BO-REP because it consumes a significant amount of memory. But in [1] a similar Hyper-representation task shows that BO-REP have a good performance compared to other baseline algorithms. Also, how do the authors choose the parameters of these algorithms? I’m not sure whether these parameters setting are fair or not. Moreover, the computational gain compared to other algorithms seems limited as shown in the experiment results.

[1] Gong X, Hao J, Liu M. A nearly optimal single loop algorithm for stochastic bilevel optimization under unbounded smoothness[C]//Proceedings of the 41st International Conference on Machine Learning. 2024: 15854-15892.

**Methods And Evaluation Criteria:**

I think the main strength is that the proposed algorithm obtains an ϵ-stationary solution with pure gradient calls and extend the smoothness condition to generalized smoothness for both upper and lower level for nonconvex-weakly-concave bilevel optimization.

The overall presentation is not clear. Specifically, in section 5 the authors give some proposition and lemmas without further description. What are the roles of these lemmas in the convergence analysis? Also, which lemmas are main novel ones? The description is very dense and difficult to parse.

**Other Comments Or Suggestions:**

no

**Other Strengths And Weaknesses:**

no

**Questions For Authors:**

The authors claim that the proposed algorithm solve the generalized-smooth bilevel optimization with nonconvex condition for both upper and lower level but only give an assumption that the lower-level objective function is weakly-convex. Did I miss something?

	About Theorem 5.10. Seems that the authors give some unusual settings for the stepsize α, β and η, with some complex form of the gradient norm in the iteration t and a variety of parameters instead of the commonly used constant stepsize in other works. Could the authors futher explain that what do these parameters stand for and how can these settings make the gradient update stable and efficient as descripted in the end of Section 4?

Regarding the experiments, the authors did not conduct the comparison with BO-REP because it consumes a significant amount of memory. But in [1] a similar Hyper-representation task shows that BO-REP have a good performance compared to other baseline algorithms. Also, how do the authors choose the parameters of these algorithms? I’m not sure whether these parameters setting are fair or not. Moreover, the computational gain compared to other algorithms seems limited as shown in the experiment results.

[1] Gong X, Hao J, Liu M. A nearly optimal single loop algorithm for stochastic bilevel optimization under unbounded smoothness[C]//Proceedings of the 41st International Conference on Machine Learning. 2024: 15854-15892.

**Relation To Broader Scientific Literature:**

No

**Theoretical Claims:**

The authors claim that the proposed algorithm solve the generalized-smooth bilevel optimization with nonconvex condition for both upper and lower level but only give an assumption that the lower-level objective function is weakly-convex. Did I miss something?

	About Theorem 5.10. Seems that the authors give some unusual settings for the stepsize α, β and η, with some complex form of the gradient norm in the iteration t and a variety of parameters instead of the commonly used constant stepsize in other works. Could the authors futher explain that what do these parameters stand for and how can these settings make the gradient update stable and efficient as descripted in the end of Section 4?

---

> ### Author Rebuttal · Authors · 2025-03-31
>
> **Q1**: The authors claim that the proposed algorithm solve the generalized-smooth bilevel optimization with nonconvex condition for both upper and lower level but only give an assumption that the lower-level objective function is weakly-convex. Did I miss something?
>
> **A1**: Thanks for your comment. At the **line 19 ( in introduction part)** of our paper, we have pointed out that the upper-level problem belongs to the category of general non-convex problems.
>
> **Q2**: Regarding the experiments, the authors did not conduct the comparison with BO-REP because it consumes a significant amount of memory. But in [1] a similar Hyper-representation task shows that BO-REP have a good ...
>
> **A2**:  Thanks for your comment. Our PNBiO can be generalized to the stochastic setting. We have initially proved that our stochastic PNGBiO has a gradient complexity of $O(\epsilon^{-6})$. In the final version of our paper, we will add this conclusion.
> Meanwhile, we have added some experimental results on stochastic version of our PNGBiO method.  Specifically,  we compare our stochastic  PNBiO algorithm with other baselines including BO-REF, F^2SA, SLIP, BOME. In the experiment, we conduct the meta-learning task at CIFAR10 dataset,  and use Resnet18 as task-shared model at the upper-level (UL) problem, and use a 2-layer neural network as task-specific model at the lower-level (LL) problem. Clearly, both UL and LL problems are non-convex. We run each method for 500 seconds (CPU time)  with minibatch size 64 for both training and validation set. We set learning rates as 0.01 at F^2SA and BOME algorithms,  and $\alpha=0.0001$, $\beta=0.01$ at SLIP and BO-REF. Our algorithm uses a vanishing step size, which is $\alpha_t=\beta_t=0.3/(k+1)^{0.5}$,$\eta=0.01$, the rest of the parameter settings are the same as in the paper, and the experimental results are shown in the following table (we have recorded the loss):
> |  | PNBiO| F$^2$SA |BOME|BO-REF|SLIP|
> |-------|-------|-------|-------|-------|-------|
> |100s|1.591| 1.822| 1.831|1.691|1.948|
> |200s|1.494 |1.655|1.692|1.585|1.839|
> |300s|1.441|1.568|1.615|1.464|1.775|
> |400s|1.392|1.485|1.542|1.402|1.723|
> |500s|1.371|1.425|1.467|1.382|1.601|
>
> In the final version of our paper, we will add these experimental results.
>
> **Q3**: About Theorem 5.10. Seems that the authors give some unusual settings for the stepsize α, β and η, with some complex form of the gradient norm in the iteration t and a variety of parameters instead of the commonly used constant stepsize in other works. Could the authors further explain that what do these parameters stand for and how can these settings make the gradient update stable and efficient as descripted in the end of Section 4?
>
> **A3**: Thanks for your comment. In our paper, we set the step size as follows:
> $\alpha_t\in\left(0,\ \frac{||\frac{1}{c_t}\nabla_1\ f(x^t,y^t)||+||\nabla_1\ g(x^t,y^t)||}{4\left(K_1+2K_2+2K_3+\kappa\right)+C}\right)$,
> $\beta_t\in\left(0,\ \frac{||\frac{1}{c_t}\nabla_2\ f(x^t,y^t)||+||\nabla_2\ g(x^t,y^t)||}{4\left(K_4+2K_4+2K_6+\kappa\right)+C}\right)$,
> and
> $\eta_t\in\left(0,\ \frac{\min\{||\frac{1}{c_t}\nabla_1\ f(x^t,y^t)||+||\nabla_1\ g(x^t,y^t)||,||\frac{1}{c_t}\nabla_2\ f(x^t,y^t)||+||\nabla_2\ g(x^t,y^t)||\}}{\kappa_{g_2}\ \max\{4\left(K_1+2K_2+2K_3+\kappa\right),4\left(K_4+2K_4+2K_6+\kappa\right)\}+C}\right)$,
> where $C=8(1+\frac{1}{\eta_t\kappa_{g_2}})L_\theta^2(L_{gx,0}+L_{gx,1}M^\rho+\frac{2}{\gamma^2})$.
>
> These step sizes are the bounds derived during the theoretical analysis process. It can be considered that during the experimental process, the selection of the step size not only depends on the parameters of the generalized smoothness of the objective function but also relies more on the gradient at the current iteration point. Due to the special structure of the algorithm, it can be viewed in the following form:
> $x^{t+1}=x^t-\alpha'_t\ \frac{||\frac{1}{c_t}\nabla_1\ f(x^t,y^t)||+||\nabla_1\ g(x^t,y^t)||}{||d_x^t||}d_x^t$,
> $y^{t+1}=y^t-\beta'_t\frac{||\frac{1}{c_t}\nabla_2\ f(x^t,y^t)||+||\nabla_2\ g(x^t,y^t)||}{||d_y^t||}d_y^t$,
> where $\alpha'_t\ \in(0,\frac{1}{4(K_1+2K_2+2K_3+\kappa)+C})$，$\beta'_t\ \in (0,\ \frac{1}{4(K_1+2K_2+2K_3+\kappa)+C})$. When $C$ is a constant, it can be regarded as a constant step size, but this depends on the boundedness of $M$.
>
> **Q4**: Specifically, in section 5 ... these lemmas in the convergence analysis? Also, which lemmas are main novel ones? The description is .....
>
> **A4**: Thanks for your suggestion. These new lemmas are crucial to our theoretical contributions, and we believe they will provide new insights for research in related fields. In the final version, we will provide a more detailed description of these lemma.  We admit that the current description may be too dense. We will reorganize this section, adding more explanations  to make it easier for readers to comprehend our arguments and conclusions.

---

### Official Review · Reviewer_4qTz · 2025-03-16

**Overall Recommendation:** 5

**Summary:**

This paper studies the generalized smooth bilevel optimization, where the upper-level objective is nonconvex with generalized smooth, and the lower-level objective is weakly convex and generalized smooth.

It proposes an effective Hessian/Jacobian-free penalty normalized gradient (PNGBiO) method to solve these bilevel optimization problems.

It also provided the convergence analysis of the proposed PNGBiO method. Some experimental results verify efficiency of the proposed PNGBiOalgorithm.

**Claims And Evidence:**

Yes

**Essential References Not Discussed:**

No

**Experimental Designs Or Analyses:**

The experimental design is robust, free of glaring weaknesses, and effectively supports its conclusions.

**Methods And Evaluation Criteria:**

This paper studies the generalized smooth bilevel optimization with nonconvex lower-level, while the existing methods mainly focus on the generalized smooth bilevel optimization relying on the strongly convex lower-level.

It presented an effective Hessian/Jacobian-free penalty normalized gradient (PNGBiO) method to solve these bilevel optimization problems, the comparison methods are designed for generalized smooth bilevel optimization which need to compute Hessian and Jacobian matrices.

**Other Comments Or Suggestions:**

N/A

**Other Strengths And Weaknesses:**

N/A

**Questions For Authors:**

1. This paper studied the generalized smooth bilevel optimization relying on the generalized smoothness introduced in [1]. The proposed PNGBiO method could apply the generalized smooth bilevel optimization with generalized smoothness introduced in [2] ?

[1] Chen, Z., Zhou, Y., Liang, Y., and Lu, Z. Generalizedsmoothnonconvex optimization is as efficient as smooth nonconvex optimization. In International Conference on Machine Learning, pp. 5396–5427. PMLR, 2023.

[2] Li, H., Qian, J., Tian, Y., Rakhlin, A., and Jadbabaie, A. Convex and non-convex optimization under generalized smoothness. Advances in Neural Information Processing Systems, 36, 2023.


2. In the experiments, how to choose the proximal parameter $\gamma$ and penalty parameter $c_t$ in the proposed PNGBiO algorithm?
3. From Figure 1 in the paper, the proposed PNGBiO method solves the approximated problem (1). In the convergence analysis, the convergence measure used in the paper is reasonable ?

4）In the convergence analysis, the term $M=\|\nabla_1g(x,\theta^*_{\gamma}(x,y)) \|$ is bounded ?

**Relation To Broader Scientific Literature:**

This paper studied the generalized smooth bilevel optimization relying on the generalized smoothness introduced in [1]
[1] Chen, Z., Zhou, Y., Liang, Y., and Lu, Z. Generalizedsmoothnonconvex optimization is as efficient as smooth nonconvex optimization. In International Conference on Machine Learning, pp. 5396–5427. PMLR, 2023.

**Theoretical Claims:**

I have read the main proof of the paper.

The primary proof presented in the paper is well-constructed, exhibiting no apparent errors, and follows a coherent and smooth logical progression that aligns with established techniques in the field.

This paper provides a convergence analysis of the proposed PNGBiO method, and proved that the PNGBiOmethod has a low gradient complexity of $O(\epsilon^{-4}) $for finding an $ \epsilon$-stationary point.

---

> ### Author Rebuttal · Authors · 2025-03-31
>
> **Q1**: This paper studied the generalized smooth bilevel optimization relying on the generalized smoothness introduced in [1]. The proposed PNGBiO method could apply the generalized smooth bilevel optimization with generalized smoothness introduced in [2] ?
>
> **A1**: Thanks for your comment. We study generalized smooth bilevel optimization based on generalized smoothness introduced  in [1] based on the following points:
>
> 1)	The framework of [1] is more compatible with our theoretical foundations and methodology, and facilitates the integration and extension of existing results.
>
> 2)	The scope of our paper is limited, and focusing on the framework of [1] helps to explore a specific problem in depth and avoid an overly broad scope.
>
> 3)	We recognize the importance of exploring different broad definitions of smoothness including [2], which will be of interest for future research.
>
> **Q2**: In the experiments, how to choose the proximal parameter $\gamma$ and penalty parameter $c_t$ in the proposed PNGBiO algorithm?
>
> **A2:** Thanks for your comment. In the experiments, we set  $c_t=\frac{10}{(k+1)^{0.25}}$ ,  $\gamma=0.1$.
>
> **Q3**: From Figure 1 in the paper, the proposed PNGBiO method solves the approximated problem (1). In the convergence analysis, the convergence measure used in the paper is reasonable?
>
> **A3**: Thanks for your comment. The metric in the article is reasonable, as our problem uses two approximations to the original problem, in the metric in this article, we have effectively carved the deviations generated during both approximations to get back to the original problem. Since the lower problem is not (strongly) convex, we could not use the traditional hyper-gradient for the carving, but instead used the residual function for the metric.
>
> **Q4**: In the convergence analysis, the term $M=\max_{\mu\ \in[0,1]}\|\nabla_1 g(x,\theta_{\mu})\|$ is bounded ?
>
> **A4**: Thanks for your comment. As you said, in the convergence analysis, $M$ has a bounded gradient paradigm for every fixed $x$. Since our iteration can be terminated in a finite number of times, for $M$ it can take the maximum value within $t$ values. Moreover, $M$ is bounded when the Lipschitz smoothness condition is satisfied for $x$ in the lower function.

---

> > ### Comment · Reviewer_4qTz · 2025-04-06
> >
> > Thanks for the authors’ responses. These responses have dealt with my concerns.
> >
> > This paper well studied bilevel optimization with nonconvex lower-level under the complex
> > generalized smooth setting, which takes bilevel optimization a step further in machine learning applications and research.

---

> > > ### Author Response · Authors · 2025-04-07
> > >
> > > Dear Reviewer 4qTz,
> > >
> > > Thanks so much for your recognition and affirmation of our paper.
> > >
> > > Best wishes,
> > >
> > > Authors

---

### Official Review · Reviewer_LknT · 2025-03-17

**Overall Recommendation:** 4

**Summary:**

This paper introduces an efficient Hessian/Jacobian-free Penalty Normalized Gradient (PNGBiO) method for solving bilevel optimization problems, where the upper-level objective is generalized smooth and nonconvex, while the lower-level objective is generalized smooth and weakly convex. Furthermore, the authors provide a comprehensive convergence analysis of the proposed PNGBiO method. Some experimental results demonstrate the effectiveness of PNGBiO.

**Claims And Evidence:**

Yes

**Essential References Not Discussed:**

No

**Experimental Designs Or Analyses:**

Yes, experimental results demonstrate the effectiveness of the proposed method.

**Methods And Evaluation Criteria:**

Yes

**Other Comments Or Suggestions:**

Typo: In Table1, the first $g(\cdot, \cdot)$ should be $f(\cdot, \cdot)$.

**Other Strengths And Weaknesses:**

Strengths:  In addition to the strengths mentioned in the Summary. In theoretical analysis, the paper proved that the proposed PNGBiO has a low gradient complexity of $O(\epsilon^{-4})$ for finding an $ \epsilon$-stationary point under milder assumptions such as generalized smoothness condition.

Weakness: Since the normalized gradient plays a crucial role, could you provide a more detailed explanation of its purpose and necessity?

**Questions For Authors:**

Q1: In the PNGBiO algorithm, how to choose the parameters $\gamma$ and $c_t$ in the experiments ?


Q2: This paper studied the nonconvex bilevel optimization with the generalized smoothness introduced in (Chen et al. 2023 ). Why not consider the nonconvex bilevel optimization with the generalized smoothness in (Li et al., 2023) ?

**Relation To Broader Scientific Literature:**

This paper proposed an effective Hessian/Jacobian-free penalty normalized gradient (PNGBiO) method to solve the generalized-smooth bilevel optimization with weakly-convex lower-level objective, while the existing methods only studied the generalized smooth bilevel optimization with strongly convex lower-level objective.

**Theoretical Claims:**

Yes, but using the same parameter $M$ to denote two different terms in the convergence analysis (at lines 240 and 266) could indeed be a typo or an oversight in notation.

---

> ### Author Rebuttal · Authors · 2025-03-31
>
> **Q1**: Using the same parameter $M$ to denote two different terms in the convergence analysis (at lines 240 and 266) could indeed be a typo or an oversight in notation.
>
> **A1**:  Thanks for your comment. There is a typo. We will correct it in the final version of our paper.
>
> **Q2**: Since the normalized gradient plays a crucial role, could you provide a more detailed explanation of its purpose and necessity?
>
> **A2**: Thanks for your insightful comment. Inspired by the paper [1], the generalized smoothing condition implies that the smoothness is unbounded, i.e., it depends on the gradient information at a certain point. During the iteration process, too large changes may produce too large or too small gradients, making the algorithm unstable and inefficient. Normalization can effectively solve these problems, making it more robust and faster convergence. Thus, in our algorithm, we use the normalized gradient descent iteration.
>
> [1] Chen, Z., Zhou, Y., Liang, Y., and Lu, Z. Generalized-smooth nonconvex optimization is as efficient as smooth nonconvex optimization. In International Conference on Machine Learning, pp. 5396–5427. PMLR, 2023.
>
> **Q3**: In the PNGBiO algorithm, how to choose the parameters $\gamma$ and $c_t$ in the experiments?
>
> **A3**: Thanks for your comment. In the experiments, we set  $c_t=\frac{10}{(k+1)^{0.25}}$ ,  $\gamma=0.1$.
>
> **Q4**: This paper studied the nonconvex bilevel optimization with the generalized smoothness introduced in (Chen et al. 2023 ). Why not consider the nonconvex bilevel optimization with the generalized smoothness in (Li et al., 2023) ?
>
> **A4**: Thanks for your comment. We chose to study the generalized smoothness nonconvex bilayer optimization proposed in (Chen et al. 2023) based on the following points:
>
> 1)	The framework of (Chen et al. 2023) is more compatible with our theoretical foundations and methodology, and facilitates the integration and extension of existing results.
>
> 2)	The scope of our paper is limited, and focusing on the framework of (Chen et al. 2023) helps to explore a specific problem in depth and avoid an overly broad scope.
>
> 3)	We recognize the importance of exploring different broad definitions of smoothness including (Li et al., 2023), which will be of interest for future research.
>
> **Q5**: In Table1, the first $g(\cdot,\cdot)$ should be $f(\cdot,\cdot)$.
>
> **A5**:  You are right, we will correct this typo in the final version of our paper.

---

> > ### Comment · Reviewer_LknT · 2025-04-07
> >
> > Thank you for the responses. I am satisfied with the answers to my concerns. I agree that exploring generalized smoothness in bilevel optimization settings is both important and challenging.

---

> > > ### Author Response · Authors · 2025-04-08
> > >
> > > Dear Reviewer LknT,
> > >
> > > Thanks so much for your recognition and affirmation of our paper.
> > >
> > > Best wishes,
> > >
> > > Authors

---

### Decision · Program_Chairs · 2025-05-01

**Decision:**

Accept (poster)

**Comment:**

This paper addresses bilevel optimization problems under generalized smoothness conditions, when both the upper-level and lower-level objectives are nonconvex. The authors introduce PNGBiO, a fully first-order, Hessian/Jacobian-free method that leverages normalized gradients. The theoretical contribution includes non-asymptotic convergence guarantees with a provable gradient complexity, and the empirical section demonstrates competitive performance across standard bilevel tasks.

I recommend acceptance. The paper makes a clear and novel contribution by proposing the first gradient-based algorithm under this generalized setting, supported by solid theoretical analysis. While presentation could be improved, the technical content is sound, and the reviewers acknowledge the approach and its convergence guarantees.